# HPV-YAP1 oncogenic alliance drives malignant transformation of fallopian tube epithelial cells

Chunbo He[1,2,13], Xiangmin Lv[1,2,13], Jiyuan Liu [1,13], Jinpeng Ruan[1,3], Peichao Chen[1], Cong Huang[1], Peter C Angeletti [4], Guohua Hua [2,5], Madelyn Leigh Moness[1], Davie Shi [1], Anjali Dhar[1], Siyi Yang[1], Savannah Murphy[1], Isabelle Montoute[1,6], Xingcheng Chen[7], Kazi Nazrul Islam[1], Sophia George [8], Tan A Ince[9], Ronny Drapkin [10], Chittibabu Guda [11], John S Davis[2,7,12] & Cheng Wang [1,2] ✉

## Abstract

**High grade serous ovarian carcinoma (HGSOC) is the most common and aggressive ovarian malignancy. Accumulating evidence indicates that HGSOC may originate from human fallopian tube epithelial cells (FTECs), although the exact pathogen(s) and/or molecular mechanism underlying the malignant transformation of FTECs is unclear. Here we show that human papillomavirus (HPV), which could reach FTECs via retrograde menstruation or sperm-carrying, interacts with the yes-associated protein 1 (YAP1) to drive the malignant transformation of FTECs. HPV prevents FTECs from natural replicative and YAP1-induced senescence, thereby promoting YAP1-induced malignant transformation of FTECs. HPV also stimulates proliferation and drives metastasis of YAP1-transformed FTECs. YAP1, in turn, stimulates the expression of the putative HPV receptors and suppresses the innate immune system to facilitate HPV acquisition. These findings provide critical clues for developing new strategies to prevent and treat HGSOC.**

**Keywords** Hippo Pathway; Fallopian Tube; HPV Infection; Innate Immunity; HGSOC
**Subject Categories** Cancer; Microbiology, Virology & Host Pathogen Interaction; Signal Transduction

## Introduction

High grade serous ovarian carcinoma (HGSOC) accounts for three-quarters of ovarian carcinoma and contributes to the vast majority of ovarian cancer-related deaths in the US (Bowtell et al, 2015).

Although rapid progress has been made towards investigating the etiology of HGSOC in the past few decades, the pathogenesis of HGSOC is still not fully understood. The origins of HGSOC are still under debate. Accumulating evidence supported the concept that a subset of HGSOC originates from fallopian tube epithelial cells (FTECs) (Crum et al, 2012; Kurman and Shih Ie, 2016; Labidi-Galy et al, 2017; Perets et al, 2013). However, the exact molecular mechanisms underlying the malignant Transformation of FTECs, as well as the factors that drive the progression of the fallopian tube derived high grade serous carcinoma (HGSC), are not entirely clear.

The Hippo signaling pathway plays a critical role in tissue homeostasis and tumorigenesis (Yu et al, 2015). The upstream of the Hippo pathway, consisting of a kinase cascade, was considered a tumor suppressor signaling since mutation/deletion of these kinases leads to overgrowth and tumorigenesis. Activation of the Hippo pathway results in the suppression of the downstream transcription coactivators YAP1 (Yes-associated protein 1) and TAZ (transcriptional coactivator with PDZ binding motif) (Moroishi et al, 2015). YAP1 is the major effector of the Hippo signaling pathway and a well-established oncoprotein. Hyperactivation of YAP1 contributes to the development of a broad range of human cancers (Pan, 2010; Zanconato et al, 2016). Our previous studies demonstrated that hyperactivated YAP1 plays a critical role in the malignant transformation of immortalized ovarian surface epithelial (HOSE) and immortalized fallopian tube secretory epithelial cells (FTSECs) (He et al, 2015a; Hua et al, 2016). However, the exact mechanism(s) underlying YAP1 induction of malignant transformation of FTSECs is still unknown. The pathogen(s) or factors inducing abnormal YAP1 expression and activation in FTECs under pathological conditions are still unclear.

Our recent studies indicated that human papillomavirus (HPV) oncoprotein E6 stabilizes the YAP1 protein in cervical epithelial cells to drive the development of cervical cancer (He et al, 2015b), suggesting that pathogenic microorganisms may interact with

---

[1]Vincent Center for Reproductive Biology, Vincent Department of Obstetrics and Gynecology, Massachusetts General Hospital, Harvard Medical School, Boston, MA 02114, USA. [2]Olson Center for Women's Health, Department of Obstetrics & Gynecology, University of Nebraska Medical Center, Omaha, NE 68198, USA. [3]School of Life Sciences, Xiamen University, Xiamen 361005, China. [4]Nebraska Center for Virology, University of Nebraska-Lincoln, Lincoln, NE 68583, USA. [5]College of Animal Science and Technology, Huazhong Agricultural University, Wuhan 430070, China. [6]Department of Human Evolutionary Biology, Harvard University, Cambridge, MA 02138, USA. [7]Fred & Pamela Cancer Center, University of Nebraska Medical Center, Omaha, NE 68198, USA. [8]Department of Obstetrics & Gynecology, Sylvester Comprehensive Cancer Center, University of Miami, Miami, FL 33136, USA. [9]New York Presbyterian Brooklyn Methodist Hospital and Department of Pathology & Laboratory Medicine, Weill Cornell Medicine, New York, NY 10021, USA. [10]Department of Obstetrics and Gynecology, University of Pennsylvania, Philadelphia, PA 19104, USA. [11]Department of Cellular and Molecular Biology, University of Nebraska Medical Center, Omaha, NE 68198, USA. [12]Western Iowa and Nebraska Veteran's Affairs Medical Center, Omaha, NE 68105, USA. [13]These authors contributed equally: Chunbo He, Xiangmin Lv, Jiyuan Liu. ✉E-mail: cwang34@mgh.harvard.edu

YAP1 to drive the carcinogenesis of epithelial layer in the female reproductive tract. HPV is a sexually transmitted infectious agent that is frequently detected in the lower genital tract of women with a very high infection prevalence (Baseman and Koutsky, 2005). Fortunately, most HPV infections are transient, and more than 70% of individuals infected with a given HPV genotype have no clinical symptoms because HPV infections are spontaneously cleared by the immune system within 12 months (Munoz et al, 2003). However, in some cases, the immune system of infected individuals may fail to control high-risk HPV-infected cells, leading to persistent infection and subsequent development of precancerous lesions, and eventually invasive cancers (Daud et al, 2011; Frazer, 2009; Kanodia et al, 2007). Previous mechanistic studies demonstrated that high-risk HPV E6 and E7 proteins target TP53 and RB1 tumor suppressors and human TERT to increase cell viability, promote cell proliferation, and drive tumorigenesis in tissues that are frequently exposed to HPV, such as the cervix and skin (Gewin and Galloway, 2001; Jones et al, 1997; Reznikoff et al, 1994; Veldman et al, 2001; Werness et al, 1990). Albeit the link of high-risk HPV with ovarian cancer is controversial, a steady stream of studies suggests that a cohort of women diagnosed with ovarian cancer have evidence of HPV (Roos et al, 2015; Rosa et al, 2013; Svahn et al, 2014). Theoretically, HPV can reach the fallopian tube epithelium. For example, retrograde menstruation may introduce high-risk HPV virions into the fallopian tube fimbriae area of HPV-positive women under certain pathological conditions. In addition, sperm may serve as a vital transporter of HPV under physiological conditions (Chan et al, 1996; Yang et al, 2003). However, whether HPV can infect FTECs is unknown. The potential role of HPV in the malignant transformation of FTECs needs to be verified.

In the present study, we found that high-risk HPV was present in a portion of chronically inflamed human fallopian tube tissues and fallopian tube carcinomas. HPV is able to infect normal and cancerous FTECs. Hyperactivated YAP1 stimulated the expression of putative HPV receptors and suppressed innate immune signaling, which increased the acquisition of FTECs to HPV infection and facilitated the evasion of HPV from immune surveillance. The interaction between YAP1 and HPV greatly promoted the progression and metastasis of fallopian tube-derived high grade serous carcinoma. Our study shows that HPV may serve as an oncogenic pathogen in some ovarian cancers that are of fallopian tubal origin, and hence, HPV vaccination may reduce the incidence of HGSOC. Our data also demonstrate that YAP1 is a promising target for developing new prevention strategies and novel therapies to conquer HGSOC with tubal origin.

## Results

### HPV oncoproteins prevent FTECs from natural and oncogene-induced senescence

Using immortalized fallopian tube epithelia cell lines, our previous studies demonstrated that hyperactivated YAP1 plays a critical role in the malignant transformation of immortalized FTSECs (Hua et al, 2016). To examine whether YAP1 has the same effect on the cultured primary human fallopian tube epithelial cells (hFTECs), we transfected hFTECs with retrovirus vectors expressing wild-type

YAP1 (FTEC-YAP) or constitutively active YAP1 (FTEC-YAP1$^{S127A}$). hFTECs transfected with empty control retrovirus vectors (FTEC-MXIV or FTEC-MX) were used as control. Unexpectedly, we found that ectopic expression of YAP1 or YAP1$^{S127A}$ promoted cellular senescence in the primary FTECs, which was indicated by the significant increase in the percentage of cells that underwent cell cycle arrest, cellular hypertrophy, and expression of senescence-associated β-galactosidase (SA-β-gal) (Fig. 1A,B; Appendix Fig. S1). This observation suggests that YAP1, like many other known oncogenes, can induce cellular senescence. The differential response to YAP1 hyperactivation between the immortalized FTSECs and the primary hFTECs suggests that unknown factors (e.g., tumorigenic pathogens), via directly or indirectly preventing FTSECs from oncogene-induced senescence, aid YAP1 in inducing malignant transformation of FTECs. Interestingly, our screening studies showed that ectopic expression of HPV16 E6/E7 in primary hFTECs blocked these cells from YAP1-promoted cellular senescence (Fig. 1A,B; Appendix Fig. S1B). In addition, we found that ectopic expression of HPV16 E6/E7 also prevented natural replicative senescence. In the primary culture and lower-passaged hFTECs, ectopic expression of HPV16 E6/E7 moderately but significantly stimulated the proliferation of hFTECs (Fig. 1C,D). After several passages, the majority of hFTECs became senescent, stopped proliferation, and gradually underwent cell death (Fig. 1E). However, FTEC-E6/E7 cells (hFTECs with ectopic expression of HPV16 E6/E7) continued to divide and showed no evidence of senescence (Fig. 1E). These data clearly indicate that HPV16 is able to prevent primary hFTECs from natural, and YAP1-promoted cellular senescence, and facilitate their proliferation. Implantation of the 6th passage hFTECs with differential YAP1 activity in the presence or absence of HPV16 E6/E7 showed that, regardless of the expression/activation of YAP1 oncogene, these cells stop proliferation without HPV16 E6/E7. However, in the presence of HPV16 E6/E7, both YAP and YAP$^{S127A}$ promote hFTEC proliferation, with the most rapid proliferation observed in the FTEC-E6/E7-YAP1$^{S127A}$. These data indicate that HPV16 E6/E7 proteins can prevent fallopian tube cells from natural and YAP-promoted cellular senescence and drive cell proliferation (Fig. 1F). In the immortalized fallopian tube epithelial cell lines such as FT190 cells (fallopian tube secretory epithelial cells immortalized by hTERT and SV40 large T) and FNE1 cells (fallopian tube secretory epithelial cells immortalized with hTERT), expression of HPV16 E6/E7 significantly promoted cell proliferation (Fig. 1G,H).

### HPV is present in normal and neoplastic fallopian tube epithelium

Theoretically, HPV can reach fallopian tube epithelial cells. We hypothesize that HPV may be able to infect and transform the epithelia of the fallopian tube. As the first step to test our hypothesis, we used PCR to amplify a 450 bp conserved sequence in the HPV L1 gene in a broad spectrum of HPV types in genomic DNA isolated from 19 cell lines, ten fallopian tube tissues that have been diagnosed with chronic inflammation and 20 fallopian tube carcinoma using the well-verified HPV detection PCR primer set (PGMY09/11) (Winder et al, 2009). ME180 (HPV68 positive) and HeLa (HPV18) cervical cancer cell lines were used as positive controls. The HT3 cervical cancer cell line (HPV negative) was used

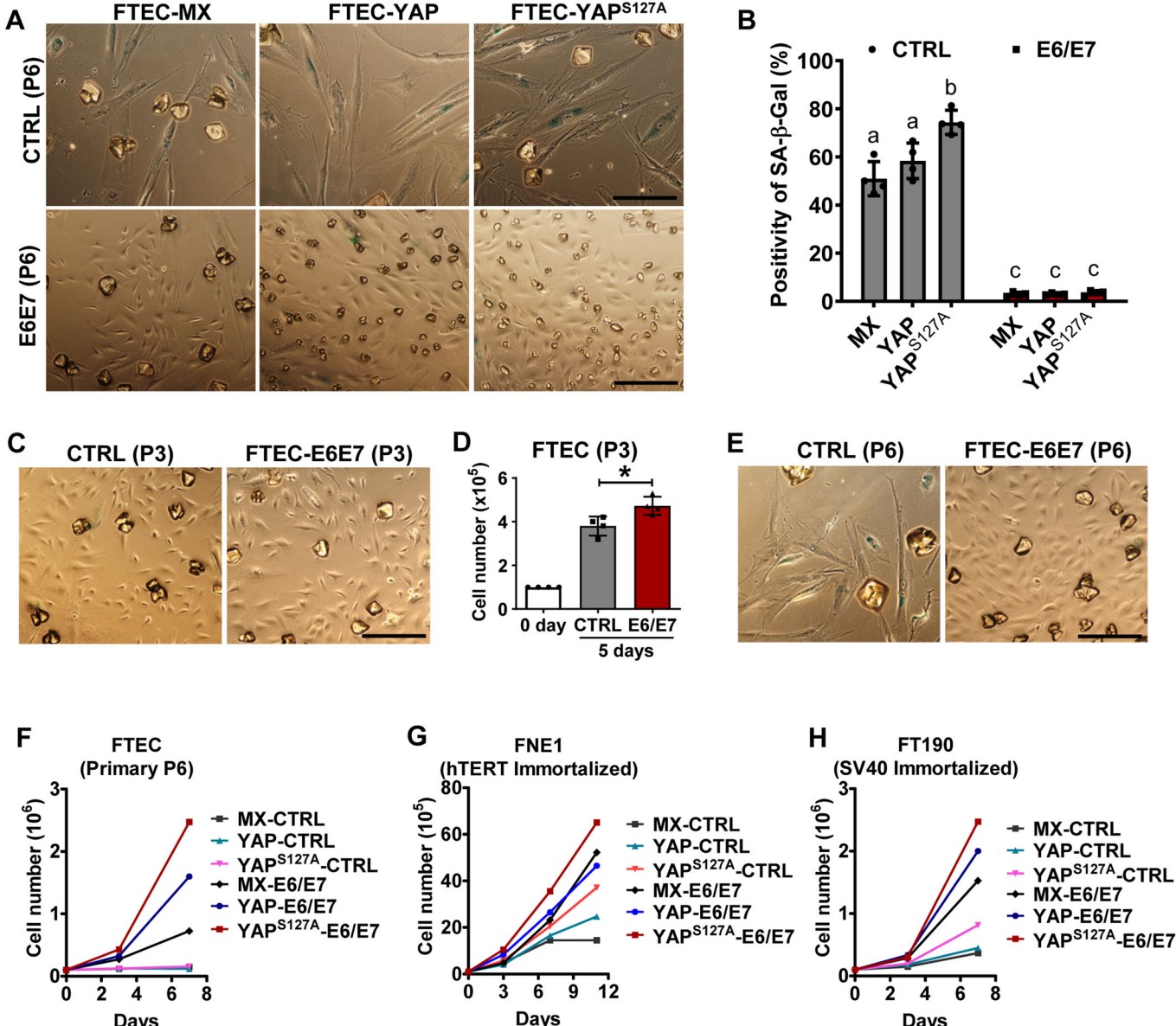

**Figure 1. HPV inhibits natural replicative and YAP1-promoted senescence.**

(A) Representative images showing the morphology and SA-β-galactosidase activity in FTECs (FTEC-MX, control), FTEC-YAP (FTECs with ectopic expression of wild-type YAP1), and FTEC-YAP^S127A cells (FTECs expressing YAP1^S127A) at their 6th passage in the presence and absence of E6/E7 gene expression. Scale bar: 200 μm.
(B) Quantitative data showing the percentage of SA-β-gal positive cells in FTEC-MX, FTEC-YAP, and FTEC-YAP^S127A cells at their 6th passage in the presence or absence of E6/E7 gene expression. Each bar represents the mean ± SEM ($n = 4$ technical replicates). Bars with different letters are significantly different from each other. Data were analyzed for significance using the two-way ANOVA followed by Tukey's multiple comparisons test. A value of $P < 0.05$ was considered statistically significant. Exact $P$ values for this analysis are presented in the source data, which is available online. (C) Representative images showing the morphology of control FTEC cells and FTEC cells expressing HPV16 E6/E7 oncoproteins (FTEC-E6/E7 cells) at their 3rd passage. Scale bar: 200 μm. (D) Quantitative data showing the cell number of control FTECs and FTEC-E6/E7 cells at the 3rd passage incubated in the growth media for five days. Each bar represents the mean ± SEM ($n = 4$ technical replicates). Data were analyzed for significance using one-way ANOVA (with the Tukey's post hoc test). A value of $P < 0.05$ was considered statistically significant. *$P < 0.05$. Exact $P$ value: CTRL vs. E6/E7, $P = 0.0112$. (E) Representative images showing the morphology and SA-β-gal staining in control FTECs and FTEC-E6/E7 cells at their 6th passage. Scale bar: 200 μm. (F) Growth curves of FTEC-MX (control), FTEC-YAP, FTEC-YAP^S127A, FTEC-E6/E7, FTEC-E6/E7-YAP, and FTEC-E6/E7-YAP^S127A cells at the 6th passage. Data in each time point represented the mean ± SEM ($n = 4$ technical replicates). (G) Growth curves of FNE1-MX (control), FNE1-YAP, FNE1-YAP^S127A, FNE1-E6/E7, FNE1-E6/E7-YAP, and FNE1-E6/E7-YAP^S127A cells. FNE1 cell is a hTERT immortalized FTEC cell line. Data in each time point represented the mean ± SEM ($n = 4$ technical replicates). (H) Growth curves of FT190-MX (control), FT190-YAP, FT190-YAP^S127A, FT190-E6/E7, FT190-E6/E7-YAP, and FT190-E6/E7-YAP^S127A cells. FT190 is an SV40 immortalized FTEC cell line. Data in each time point represented the mean ± SEM ($n = 4$ technical replicates). Source data are available online for this figure.

A

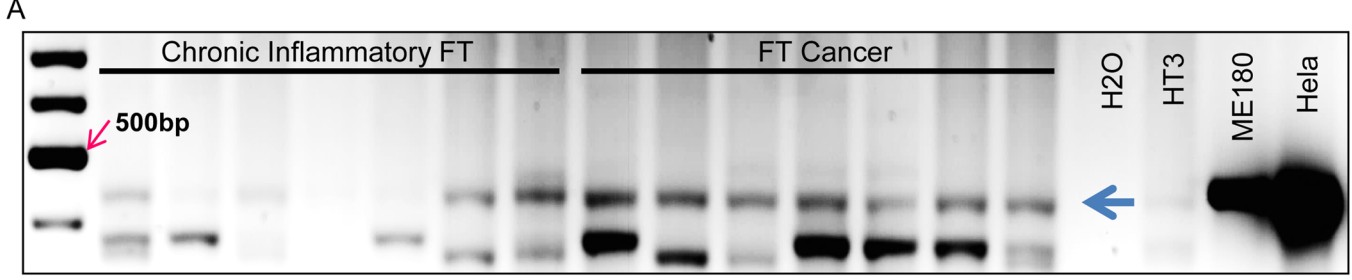

B

**Low Risk HPV Test**

(HPV6/11/42/43/44)

C

**High Risk HPV Test**

HPV16/18/31/33/35/39/45/51/52/56/58/59/68

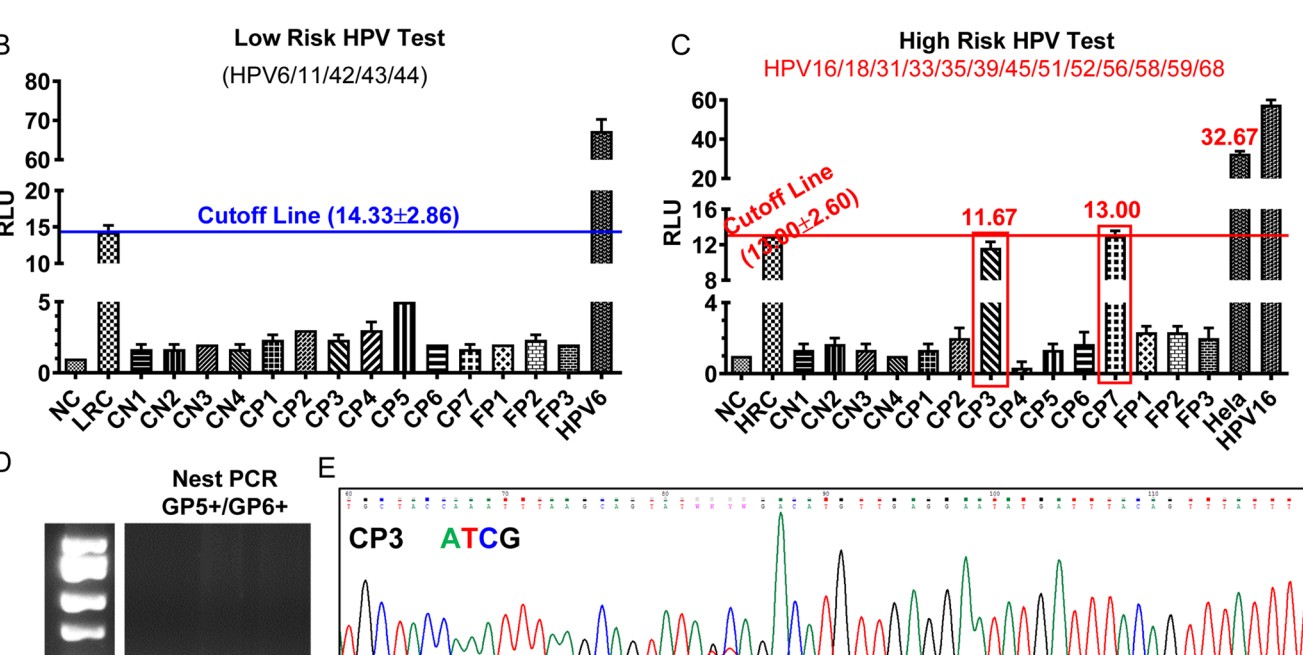

D

**Nest PCR GP5+/GP6+**

E

F

**Human papillomavirus type 18 isolate CC108 L1 protein (L1) gene, partial cds**

Sequence ID: MF066886.1   Length: 145   Number of Matches: 1

| Score | Expect | Identities | Gaps | Strand |
|---|---|---|---|---|
| 97.1 bits(52) | 3e-17 | 56/60(93%) | 0/60(0%) | Plus/Plus |

```
CP3      1   TGCTACCAAATTTAAGCAGTATWKYWGACATGTTGAGGAATATGATTTACAGTTTATTTT   60
             |||||||||||||||||||||||    | ||||||||||||||||||||||||||||||||
HPV18L1  84  TGCTACCAAATTTAAGCAGTATAGCAGACATGTTGAGGAATATGATTTACAGTTTATTTT   143
```

G

**Human papillomavirus type 16 isolate Achham_1, complete genome**

Sequence ID: KX947269.1   Length: 7906   Number of Matches: 1

| Score | Expect | Identities | Gaps | Strand |
|---|---|---|---|---|
| 93.5 bits(50) | 4e-16 | 56/61(92%) | 0/61(0%) | Plus/Plus |

```
CP7       1     AAAATACTAACTTTAAGGAGTACCTMCYWYWTGGGGAGGAATATGATTTACAGTTTATTT   60
                ||||||||||||||||||||||||| | |  | ||||||||||||||||||||||||||||
HPV16L1   6705  AAAATACTAACTTTAAGGAGTACCTACGACATGGGGAGGAATATGATTTACAGTTTATTT   6764
```

**Figure 2. HPV is present in normal and cancerous fallopian tube tissues.**

(A) Representative images showing PCR products of PGMY09/PGMY11 primer set in chronic inflammatory fallopian tube tissues and fallopian tube cancer tissues. PCR products were fragmented on a 2% agarose gel, stained with ethidium bromide, and visualized under a UV-transilluminator. The left lane (Lane 1) is the 100 bp nucleic acid molecular marker. Lanes 2 to 8 are chronic inflammatory fallopian tube tissues. Lanes 9 to 15 are fallopian tube cancer tissues. Note that ME180 and Hela cells (HPV-positive cell lines) are used as positive controls. HT3 (HPV negative cell line) and $H_2O$ (to replace the cDNA template in the reaction mixture) are negative controls. The blue arrow points to the expected HPV+ bands. (B, C) Results of HC2 HPV DNA Test in fallopian tube tissues for five low-risk types (6/11/42/43/44) (B) and 13 high-risk types (16/18/31/33/35/39/45/51/52/56/58/59/68) (C). Each bar represents the mean ± SEM ($n = 3$ technical replicates). Sample preparation, experimental operation, and result interpretation were performed according to a protocol included in QIAGEN's digene HC2 HPV DNA Test kit (# 5198-1220, QIAGEN Ltd, UK). NC (Negative Calibrator), LRC (Low-Risk HPV Calibrator), HRC (High-Risk HPV Calibrator), HPV16 DNA, and HPV6 DNA were from the QIAGEN Kit. CN: cancer samples with negative PGMY11/09 Result; CP: fallopian tube cancer samples with positive PGMY11/09 result; FP: fallopian tube samples with positive PGMY11/09 result. (D) Representative images showing nested PCR products amplified with the PGMY09/11-GP5 + /6+ system on fallopian tube cancer samples #3 (CP3) and #7 (CP7). These CP3 and CP7 Samples were negative for the GP5 + /6+ test in the primary PCR screen. (E) Representative images showing DNA sequences of PCR products of CP3 and CP7 amplified using PGMY09/11-GP5 + /6 + PCR system. (F, G) Representative images showing that the DNA sequence of nested PCR product of CP3 cDNA was aligned to HPV18 L1 gene (F), while the DNA sequence of nested PCR product of CP7 cDNA was aligned to HPV16 L1 gene (G). Source data are available online for this figure.

as a negative control. *β-actin* was used as a DNA loading control. We found that HeLa and ME180 cells were HPV positive, while the HT3 cell line was HPV negative, suggesting that the PCR system was valid (Appendix Fig. S2). Our results indicated that at least two (IGROV1 and OVCAR3) out of nine ovarian cancer cells and two (FT190 and FT240) out of five immortalized fallopian tube secretory epithelial cells were HPV positive (Appendix Fig. S2). Importantly, HPV was detected in seven out of twenty fallopian tube cancer tissues and three out of ten chronically inflammatory fallopian tube tissues (Fig. 2A). We then used an FDA-approved CE-IVD marked HPV test kit to confirm the presence of HPV in fallopian tube tissues. CE-IVD marked HPV assay is based on DNA-RNA hybridization and can detect 13 high-risk HPV types (16/18/31/33/35/39/45/51/52/56/58/59/68) and five low-risk types (6/11/42/43/44). The Hybrid capture results showed that all fallopian tube tissues examined are negative for low-risk HPV types (Fig. 2B). However, two fallopian tube cancer samples (CP3 & CP7) were positive for High-risk HPV probe (Fig. 2C). The nested PCR using PGMY09/11 and GP5 + /6+ primer sets, which has been proven to be able to detect several types of HPV with low HPV signal (Fuessel Haws et al, 2004), clearly indicated that sample CP3 & CP7 were high-risk HPV positive (Fig. 2D). We then sequenced the purified PCR products using GP5+ primer (Fig. 2E). The sequence BLAST analyses (using NCBI databases) indicated that sample CP3 and sample CP7 contained HPV18 and HPV16 DNA, respectively (Fig. 2F,G). To examine if the HPV DNA in ovarian cancer tissues is transcriptionally active, we examined HPV16 E6/E7 mRNA expression in 11 serous tubal intraepithelial carcinoma tissues using RNAscope assay (HGSOC early lesion, FFPE slides was a gift from Dr. Christopher Crum at Dana-Farber Cancer Center). The hybridization test indicated that out of eleven tested slides, only one sample was positive for HPV16 E6/E7 (Fig. EV1).

## HPV, mediated by putative cell surface HPV receptor molecules, can infect fallopian tube epithelial cells and ovarian cancer cell lines

The above results show that HPVs are present in at least a portion of normal and cancerous fallopian tube tissues, suggesting that HPVs can infect FT epithelial cells. To provide direct evidence that HPV can infect FTECs, we generated an HPV16 pseudovirion (HPV16 PsV) by co-transfection of L1 and L2-encoding expression

vector plasmids and a plasmid encoding a GFP reporter gene using a protocol described previously to mimic HPV infection process (Buck et al, 2006). The percentage of GFP-positive cells in HPV16-PsV treated FTECs indicates the infectivity of HPV. After incubated with 0.1 MOI (Multiplicity of Infection) HPV16-PsVs for 72 h, GFP signal was detected in primary human cervical epithelial cells (hCerEC), but not in primary FTECs (Fig. 3A,B). A detectable GFP signal was observed in primary FTECs when the concentration of PsV increased to 0.5 MOI. When the MOI increased to 5.0, around 13% of primary FTECs and 51% of hCerECs were infected by HPV16-PsV after incubating for 72 h (Fig. 3A,B). To compare the susceptibility to HPV infection in FTECs and HGSOC cells, hCerEC (positive control), primary FTECs, immortalized FTSECs (FT194 and FT246), and HGSOC cell lines (COV362 and OVSAHO) were incubated in growth medium with 2.0 MOI HPV16-PsVs for 72 h. Interestingly, immortalized FT194 FTSECs also had higher susceptibility to HPV infection compared to primary FTECs (Fig. 3C,D). The HPV infectivity in the slow-growing FT246 cell line only increased slightly compared to the FTECs, suggesting that HPV infectivity may be associated with cell proliferation rate. Consistently, two HGSOC cell lines had the highest ratios of GFP-positive cells, even much higher than hCerEC. Primary FTECs had the lowest susceptibility to HPV infection (Fig. 3C,D; Appendix Fig. S3). It seems that incubation time also affected HPV16-PsV infection of primary and immortalized FTECs. Compared to 6 h treatment, incubating cells with 2.0 MOI HPV16-PsVs for 72 h increased the infectivity of hCerEC, primary FTECs, and immortalized FTECs (FT194 and FT246) ($P < 0.001$, Appendix Fig. S3).

Previous studies indicated that cell surface molecules, such as ITGA6, SDC1, and EGFR, facilitate the binding and initial entry of HPV virions into cells and were identified as HPV receptor molecules (Day et al, 2003; Raff et al, 2013; Yoon et al, 2001). By exploring GTEx and TCGA ovarian cancer datasets with the TCGA TARGET GTEx study online platform (https://xenabrowser.net/), we compared the expression of genes encoding the putative HPV receptor molecules in normal ovarian tissues ($n = 88$), primary ovarian tumor ($n = 418$), and recurrent ovarian tumor ($n = 8$), and found that the expression of HPV receptor molecules such as *SDC1*, *ITGA6*, *HSPG2*, are significantly higher in primary and recurrent tumor tissues when compared to that in the normal control tissue ($P < 0.0001$) (Fig. EV2). We speculated that the differential susceptibility of cells to HPV infection observed above may be

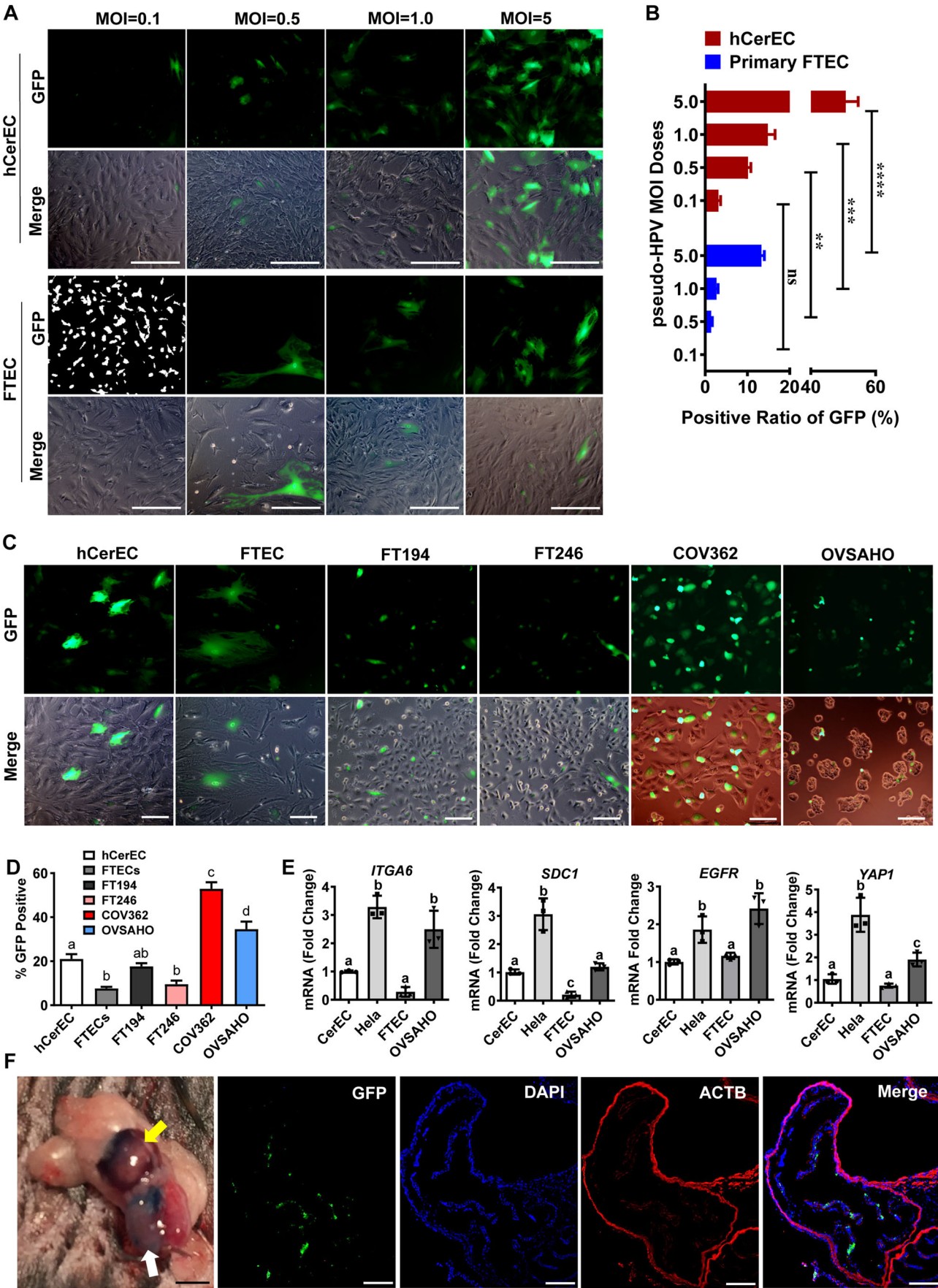

**Figure 3. HPV differentially infects fallopian tube epithelial and ovarian cancer cells.**

(A) Representative images showing HPV16 pseudovirions-derived GFP signal in primary human cervical epithelial cells (hCerEC) and primary fallopian tube epithelial cells (FTEC) incubated with different concentrations (MOI = 0.1, 0.5, 1, and 5) of HPV16 pseudovirions (PsV). The infection efficiencies of HPV16 PsV in hCerEC and FTEC cells are presented as the ratio of GFP-positive cells. Scale bar: 200 µm. (B) Quantitative data showing the ratio of GFP-positive cells in hCerEC and FTEC cells under different concentrations of HPV PsV. Each bar represented the mean ± SEM ($n = 4$ technical replicates). Data were analyzed for significance using the two-way ANOVA followed by the Tukey's multiple comparisons post hoc test. A value of $P < 0.05$ was considered statistically significant. ****$P < 0.0001$; ***$P < 0.001$; **$P < 0.01$; ns: not significant, between indicated two groups. Exact $P$ values for all comparisons are presented with the corresponding source data, which is available online. (C) Representative images showing the presence of HPV16 PsV-derived GFP signal in hCerEC, FTEC, FT194, FT246, COV362, and OVSAHO cells incubated with of HPV16-Psv (MOI = 2.0) for 72 h. Scale bar: 100 µm. (D) Quantitative results showing the ratio of HPV16 PsV positive cells in hCerEC, FTEC, FT194, FT246, COV362, and OVSAHO cells. Each bar represented the mean ± SEM ($n = 4$ technical replicates). Bars with different letters are significantly different from each other ($P < 0.05$). Data were analyzed for significance using the one-way ANOVA followed by the Tukey's post hoc test. A value of $P < 0.05$ was considered statistically significant. Exact $P$ values when compared to FTECs: $P = 0.0059$ for hCerEc; $P = 0.0526$ for FT194; $P = 0.9903$ for FT246; $P < 0.00011$ for COV326; $P < 0.0001$ for OVSAHO. (E) Quantitative data showing mRNA levels of *YAP1* and the putative HPV receptors (*ITGA6*, *SDC1*, and *EGFR*) in hCerEC, FTEC, Hela, and OVSAHO cells. Each bar represented the mean ± SEM ($n = 3$ technical replicates). Bars with different letters are significantly different from each other ($P < 0.05$). Data were analyzed for significance using the one-way ANOVA followed by the Tukey's multiple comparisons post hoc test. A value of $P < 0.05$ was considered statistically significant. Exact $P$ values for all comparisons are presented with corresponding source data, which is available online. (F) HPV16 PsV infected fallopian tube epithelial cells in vivo. HPV16 PsV ($1.0 \times 10^6$ pfu/µl in 30 µl saline with Evans blue dye) was injected into the right uterus (near the oviduct) of C57BL/6 mice ($n = 5$ technical replicates). The left uterus was injected with the same amount of saline with Evans blue dye and used as a negative control. The presence of HPV16 PsV in the oviducts and ovarian bursa was monitored by the blue color (left panel). Yellow arrow points to ovary and ovarian bursa. White arrow pints to uterus. Scale bar: 3.0 mm. Representative images showing the presence of HPV16 PsV-derived GFP signal in the epithelium of mouse oviduct. Nuclei were stained with DAPI (blue). Scale bar: 100 µm. Source data are available online for this figure.

attributed to the differential expression of these putative HPV receptor molecules. To verify this hypothesis, we examined the mRNA expression of these putative HPV receptors in hCerECs, primary FTECs, HeLa cells (HPV-positive cancer cell line), and OVSAHO ovarian cancer cell line. As shown in Fig. 3E, the expression of *ITGA6* and *SDC1* in hCerECs was much higher than in FTECs. Most importantly, *ITGA6* and *SDC1* were highly expressed in HeLa and OVSAHO cancer cells (Fig. 3E). As expected, YAP expression was significantly higher in Hela and OVSAHO cancer cells compared to the two primary cells. These data partially explain why the HPV16 PsV preferentially infected the hCerECs and ovarian tumor cells. Consistent with these in vitro findings, we observed uptake of HPV16-PsV into the oviduct epithelial cells 72 h after uterine HPV infusion (Fig. 3F).

## YAP, by regulating the expression of the putative cell surface HPV receptors, facilitates HPV infection of the fallopian tube and ovarian epithelial cells

Our previous results have shown that YAP1 transforms immortalized FTECs (Hua et al, 2016). Here, we observed that fallopian tube and ovarian HGSC cells, which generally have higher expression of YAP1 (He et al, 2015a; Hua et al, 2016), also have higher expression of HPV receptors and increased susceptibility to HPV infection (Fig. 3). These results indicated that YAP1 may be involved in the HPV infection of FTECs. To verify this speculation, we examined the susceptibility of FT246-MX (FT246 transfected with empty control vector MXIV), FT246-YAP (FT246 with ectopic expression of wild-type *YAP1*), and FT246-YAP$^{S127A}$ (FT246 expressing YAP$^{S127A}$, a constitutively active form of YAP1) cells to HPV infection. After incubation with 2.0 MOI HPV16-PsVs for 72 h, we found that compared to control FT246-MX cells, significantly more HPV16-PsV GFP signals were detected in FT246-YAP and FT246-YAP$^{S127A}$ cells (Fig. 4A; Appendix Fig. S4A). Similar changes in HPV infectivity were observed in FT194-MX (Control), FT194-YAP, and FT194-YAP$^{S127A}$ cells (Appendix Fig. S4B,C). These results demonstrate that YAP1 increases the susceptibility of FTECs to HPV infection. Mechanistic studies showed that ectopic

expression of YAP or YAP$^{S127A}$ significantly induced expression of the putative HPV16 receptors (*ITGA6*, *SDC1*, and *EGFR*) in FNE1, an hTERT immortalized fallopian tube epithelial cells (Fig. 4B), while knockdown of *YAP1* suppressed expression of the putative HPV16 receptors (*ITGA6*, *SDC1*, and *EGFR*) ($P < 0.001$, Fig. 4C). Consistent with these observations, we found that knockdown of *YAP1* in OVSAHO cells suppressed the infectivity of HPV16 PsV in these cells ($P < 0.001$, Fig. 4E,F). Similarly, the knockdown of *ITGA6* in OVSAHO resulted in an 80% decrease in HPV16 PsV-GFP signal in these cells ($P < 0.001$, Fig. 4D,G,H). ITGA6 has been reported to be an essential protein for HPV16 entry of cervical cells (Yoon et al, 2001). To confirm the role of ITGA6 in mediating YAP1 action on the HPV infection of FTECs, we knocked down *ITGA6* in FT246-MXIV, FT246-YAP1, and FT246-YAP$^{S127A}$ cells. Fluorescent microscopy showed that knockdown of *ITGA6* suppressed YAP- and YAP$^{S127A}$-induced increase of HPV16 PsV infection in FT246 cells (Fig. 4I; Appendix Fig. S5). These data demonstrate that YAP1-stimulated upregulation of the putative HPV receptor proteins such as ITGA6 are critical mediators of YAP1 action on HPV infection of FTECs.

## HPV oncoproteins aid YAP1 in malignant transformation of primary fallopian tube epithelial cells and drive massive metastasis of YAP-induced fallopian tube HGSC

The above results indicate that HPV can infect fallopian tube epithelial cells, and the hyperactivated YAP1 in FTECs facilitated HPV infection. We have shown that hyperactivation of YAP1 in immortalized fallopian tube epithelial cells induced high-grade serous carcinoma (Hua et al, 2016). Whether HPV plays a role in YAP1-induced FTEC malignant transformation is unknown. The soft agar assay showed that primary FTECs and two immortalized cell lines (FT190 and FNE1) could not form a colony in the soft agar system (Fig. 5A; Appendix Fig. S6). Expression of HPV16 E6/E7 or wild-type YAP1 alone in these cells also did not induce colony formation. Ectopic expression of YAP$^{S127A}$ caused some colonies in the FT190 and FNE1 cells but failed to do so in the primary FTEC cells. Combined expression of HPV16 E6/E7 and

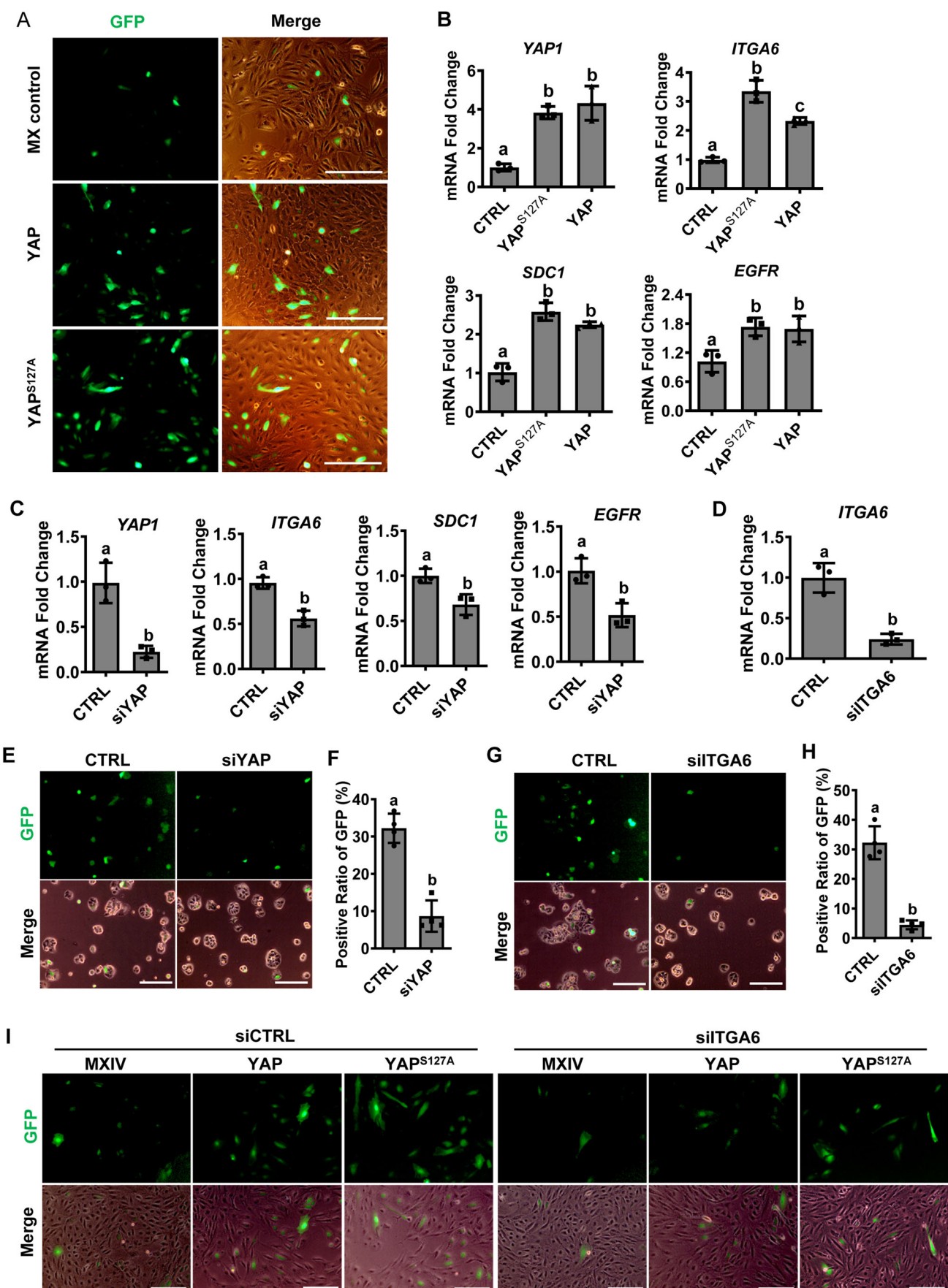

◀ **Figure 4. YAP regulates expressions of putative HPV receptors.**

(A) Representative images showing HPV16 PsV-derived GFP signal in FT246-MX (control), FT246-YAP, and FT246-YAP$^{S127A}$ cells. The quantitative data is presented in Appendix Fig. S4A. Scale bar: 100 µm. (B) Quantitative data showing mRNA levels of *YAP1* and the putative HPV receptors (*ITGA6*, *SDC1*, and *EGFR*) in FT246-MX (control), FT246-YAP, and FT246-YAP$^{S127A}$ cells. Each bar represented the mean ± SEM ($n = 3$ technical replicates). Bars with different letters are significantly different from each other. Data were analyzed for significance using the one-way ANOVA followed by the Tukey's post hoc test. A value of $P < 0.05$ was considered statistically significant. Exact *P* values for each gene are presented with the corresponding source data, which is available online. (C) Quantitative data showing mRNA levels of the putative HPV receptors (*ITGA6*, *SDC1*, and *EGFR*) in control (scramble siRNA, CTRL) and *YAP1*-knockdown (siYAP1) OVSAHO cells. Each bar represented the mean ± SEM ($n = 3$ technical replicates). Bars with different letters are significantly different from each other. Data were analyzed for significance using the unpaired t test. A value of $P < 0.05$ was considered statistically significant. Exact *P* values when compared to their corresponding control (CTRL): $P = 0.0048$ for *YAP1*; $P = 0.0031$ for *ITGA6*; $P = 0.0161$ for *SDC1*; $P = 0.0115$ for *EGFR*. (D) RT-PCR analyses showing successful knockdown of *ITGA6* in OVSAHO cells using *ITGA6* siRNAs (siITGA6). Each bar represented the mean ± SEM ($n = 3$ technical replicates). Bars with different letters are significantly different from each other ($P = 0.0025$). Data were analyzed for significance using the unpaired t test. A value of $P < 0.05$ was considered statistically significant. (E) Representative images showing HPV16 PsV-derived GFP signal in control (scramble siRNA, CTRL) and *YAP1*-knockdown (siYAP1) OVSAHO cells. GFP signal indicated the infection efficiency of HPV16 PsV in these cells. Scale bar: 100 µm. (F) Quantitative results of (E) to show the ratio of GFP-positive cells in OVSAHO cells with (siYAP1) or without (CTRL) *YAP1* knockdown. Each bar represented the mean ± SEM ($n = 4$ technical replicates). Bars with different letters are significantly different from each other ($P = 0.0002$). Data were analyzed for significance using the unpaired t test. A value of $P < 0.05$ was considered statistically significant. (G) Representative images showing HPV16 PsV-derived GFP signal in control (scramble siRNA, CTRL) and *ITGA6*-knockdown (siITGA6) OVSAHO cells. Scale bar: 100 µm. (H) Quantitative results of (G) to show the ratio of GFP-positive cells in OVSAHO cells with (siITGA6) or without (CTRL) knockdown of *ITGA6*. Each bar represented the mean ± SEM ($n = 4$ technical replicates). Bars with different letters are significantly different from each other ($P = 0.0002$). Data were analyzed for significance using the unpaired t test. A value of $P < 0.05$ was considered statistically significant. (I) Representative images showing HPV16 PsV-derived GFP signal in FT246-MXIV, FT246-YAP, and FT246-YAP$^{S127A}$ cells with (siITGA6) or without (siCTRL) knockdown of *ITGA6* using RNA interference technique. The quantitative data are presented in Appendix Fig. S5. Source data are available online for this figure.

wild-type YAP1 induced the formation of small colonies in all three cells (Fig. 5A; Appendix Fig. S6). However, the combined expression of HPV16 E6/E7 and YAP$^{S127A}$ induced many large colonies in all three cells (Fig. 5A; Appendix Fig. S6). These results indicate that the combination of hyper-activation of YAP and infection of high-risk HPV can cause the transformation of fallopian tube epithelial cells.

FNE1-MX, FNE1-YAP$^{S127A}$, FNE1-E6/E7, and FNE1-E6/E7-YAP$^{S127A}$ cells were then injected subcutaneously ($2 \times 10^6$ cells/mouse) to examine whether HPV16 E6/E7 plays a role in YAP1-induced tumorigenesis in vivo. As expected, tumors formed in all six mice injected with FNE1-YAP$^{S127A}$ cells within three months of injection, while no tumor (0/6) formed in mice injected with FNE1-MX cells (Fig. 5B). Consistent with our previous report (Hua et al, 2016), these tumors express a high level of nuclear YAP and are morphologically resemble high grade cancers with serous characters (Fig. 5C,D). Injection of FNE1-E6/E7 cells to athymic nude mice failed to form tumors nine months after cell injection, suggesting that HPV16 E6/E7 alone is not sufficient to induce malignant transformation of fallopian tube epithelial cells (Fig. 5B,E). Interestingly, we found that all mice (6/6) injected with FNE1-E6/E7-YAP$^{S127A}$ cells developed tumors approximately one month after cell injection (roughly two months earlier than mice injected with FNE1-YAP$^{S127A}$ cells alone). Compared to FNE1-YAP$^{S127A}$ cells, tumors derived from FNE1-E6/E7-YAP$^{S127A}$ cells initiated and progressed more rapidly. In addition, 67% (4/6) of mice developed signs of cancer-associated cachexia, indicated by evident weakness, loss of body weight, muscle atrophy, and reduced fat mass (Fig. 5E,F). Two-thirds of mice injected with FNE1-E6/E7-YAP$^{S127A}$ cells accumulated ascites fluid in their abdomen, which could be easily identified by the abdominal expansion (Fig. 5E,F). Cancer cells derived from FNE1-E6/E7-YAP$^{S127A}$ cells metastasized to multiple organs in the peritoneal cavity, including the gastrointestinal system, the mesentery and pancreas, the diaphragm and peritoneal membrane (Fig. 5G,F), although cancer cells were injected subcutaneously and the tumors were initially presented only under the skin. The massive pelvic metastasis was similar to that observed in human HGSOC patients at the advanced stages.

Histological analyses showed that these cancer cells had marked nuclear atypia and pleomorphism, showing more than a three-fold variation in size. These tumor cells had a high percentage of Ki-67 positive cells and frequent appearance of mitotic figures (Fig. 5H,I). Importantly, immunohistochemical studies showed that these cancer cells express known biomarkers for HGSOC, including KRT7 (Fig. 5J,K), PAX8 (Fig. 5L,M), WT1 (Fig. 5N,O), and nuclear TP53 (Fig. 5P,Q). These cancers had all the features of high-grade serous carcinoma.

## YAP suppresses innate immunity to protect HPV from immune attack

To understand the molecular mechanisms by which YAP1 and HPV interact with each other to drive HGSOC development, we performed genome-wide gene expression profiling in FNE1-MX, FNE1-YAP$^{S127A}$, FNE1-E6/E7, and FNE1-YAP$^{S127A}$-E6/E7 cells. Increased expression of *YAP1* and its target genes (*CYR61*, *KRT7*) in FNE1-YAP$^{S127A}$ and FNE1-YAP$^{S127A}$-E6/E7 cells and an increased expression of *TERT* (a known HPVE6/E7 target gene) in FNE1-E6/E7 and FNE1-YAP$^{S127A}$-E6/E7 cells indicated that ectopic expression of YAP$^{S127A}$ and E6/E7 in these cell lines were successful (Fig. 6A). Interestingly, HPV16 E6/E7, not YAP1, upregulated transcription of WT1, a known marker of HGSOC (Fig. 6A). Gene set enrichment analysis (GSEA) based on genome-wide gene expression data indicated that the top genes and pathways enriched in HPV E6/E7 cells are associated with chromatin remodeling and cell stemness (Fig. 6B), while the top genes enriched in the FNE1-YAP$^{S127A}$ cells are associated with increased cellular senescence and decreased innate immunity (Fig. 6C). The combined expression of E6/E7 and YAP$^{S127A}$ in FNE1-YAP$^{S127A}$-E6/E7 cells resulted in significant enrichment of genes and pathways involved in carcinogenesis, which is consistent with above observations. Notably, combined expression of E6/E7 with YAP$^{S127A}$ also results in drastic enrichment of genes and pathways involved in the suppression of innate immunity (Fig. 6D). Our subsequent Real-time PCR analyses showed that ectopic expression of YAP$^{S127A}$ in FNE1 cells suppressed expression of multiple TLRs, including

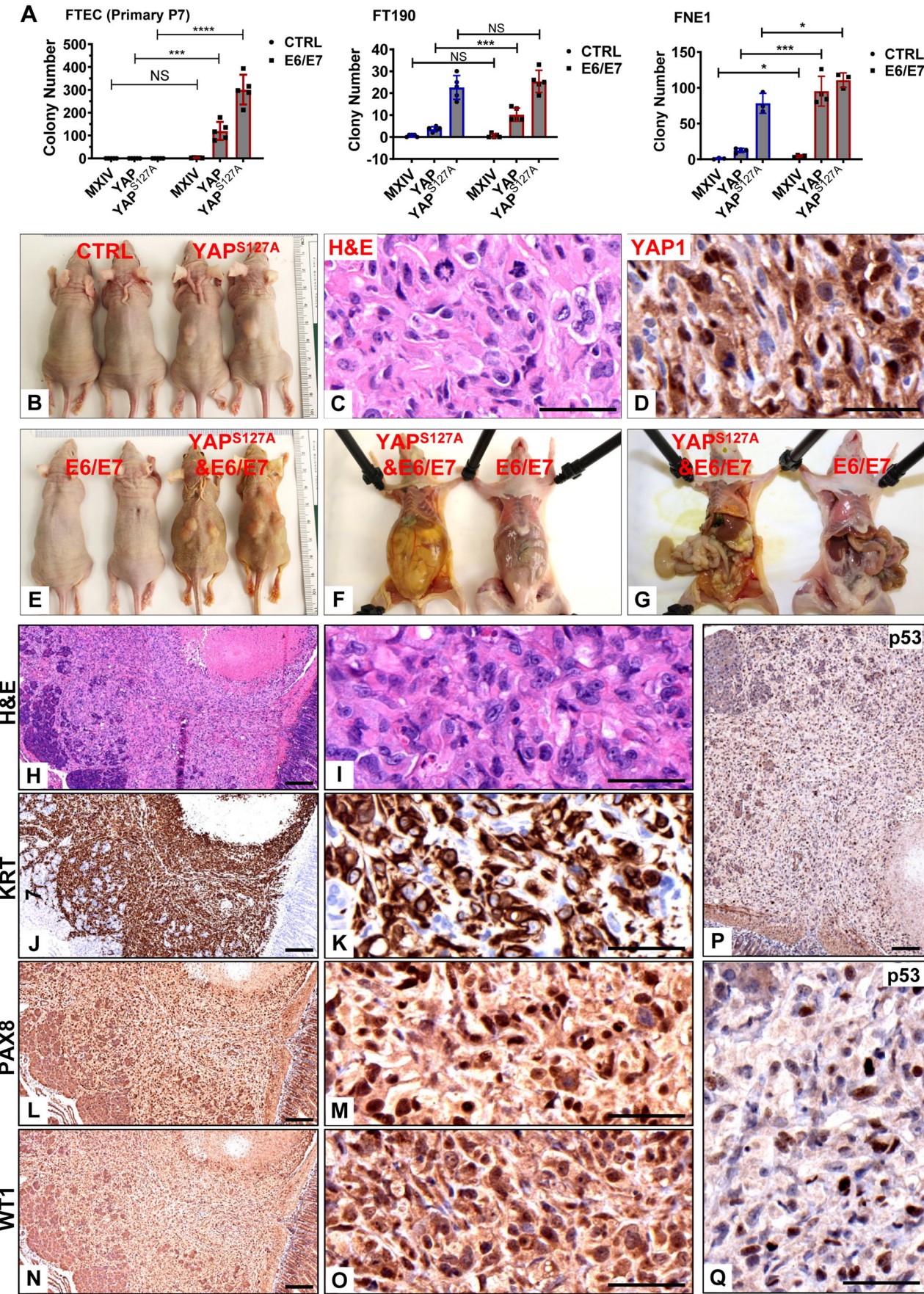

Figure 5. HPV E6/E7 contributes to the malignant transformation of FTECs and drives metastasis of YAP1-induced HGSOC.

(A) Quantitative data showing colonies formed by modified primary FTECs [FTEC-MX (control), FTEC-YAP, FTEC-YAP$^{S127A}$, FTEC-E6/E7, FTEC-E6/E7-YAP and FTEC-E6/E7-YAP$^{S127A}$ cells], FNE1-derived cells [FNE1-MX (control), FNE1-YAP, FNE1-YAP$^{S127A}$, FNE1-E6/E7, FNE1-E6/E7-YAP and FNE1-E6/E7-YAP$^{S127A}$ cells], and FT190-derived cells [FT190-MX (control), FT190-YAP, FT190-YAP$^{S127A}$, FT190-E6/E7, FT190-E6/E7-YAP and FT190-E6/E7-YAP$^{S127A}$ cells] at their 6th passage in the soft agar assay. Each bar represented the mean ± SEM ($n = 5$ technical replicates). Data were analyzed for significance using the two-way ANOVA followed by the Tukey's multiple comparisons post hoc test. A value of $P < 0.05$ was considered statistically significant. ****$P < 0.0001$; ***$P < 0.001$; *$P < 0.05$; ns: not significant, between indicated two groups. Exact $P$ values for the compared groups in each graph are presented with source data, which is available online. (B) Representative images showing tumorigenesis of FNE1-MX (control) and FNE1-YAP$^{S127A}$ cells. Please note that no tumor formed in the FNE1-MX control group. (C) Representative images showing H&E staining of FNE1-YAP$^{S127A}$ cell-derived tumors. Scale bar: 50 μm. (D) Representative IHC images showing YAP1 expression in FNE1-YAP$^{S127A}$ cell-derived tumors. Scale bar: 50 μm. (E, G) Representative images showing the tumorigenesis of FNE1-E6/E7 and FNE1-E6/E7-YAP$^{S127A}$ cells. Please note that no tumor formed in E6/E7 alone group (FNE1-E6/E7 cells). Tumors derived from FNE1-E6/E7-YAP$^{S127A}$ cells metastasized intraperitoneally to multiple organs and tissues. (H, I) Representative images showing the histology (H&E staining) of tumor tissues derived from FNE1-E6/E7-YAP$^{S127A}$ cells. (J–Q) Representative IHC images showing expressions of known biomarkers (KRT7, PAX8, WT1, and TP53) of high grade serous ovarian carcinoma (HGSOC) in tumor tissues derived from FNE1-E6/E7-YAP$^{S127A}$ cells. Scale bars in (H), (J), (L), (N), and (P): 200 μm; Scale bars in (I), (K), (M), (O), and (Q): 50 μm. Source data are available online for this figure.

*TLR1, TLR2, TLR3, TLR5,* and *TLR6* (Fig. 7A, $P < 0.001$, compared to FNE1-MX control). The expression of *TICAM1* and *MYD88*, which encode two key adapter proteins for TLRs, was also significantly suppressed by expressing YAP$^{S127A}$. Moreover, the expression of another TLRs' adapter gene, TIR-domain-containing adapter-inducing interferon-β (*TRIF*) was downregulated by YAP1 activation (Appendix Fig. S7). Importantly, expression of YAP$^{S127A}$ in FNE1 cells suppressed expression of *TBK1*, the critical kinase for activation of *NFκBs* and *IRFs* (Cui et al, 2014), and reduced mRNA levels of *NFκB1, NFκB2, RELA (NFκB3), IRF3, IRF7,* as well as other members of the IRF family, which are key transcription factors for the production of type I interferons (Fig. 7A; Appendix Fig. S8). Fluorescent immunohistochemistry showed that in FNE1 cells, IRF3 and NFκB1 were localized in nucleus and cytoplasm (Fig. 7B). Ectopic expression of YAP or YAP$^{S127A}$ resulted in decreased nuclear IRF3 and NFkB1 proteins (Fig. 7B). In response to the reduction of type I interferon signaling, we found that mRNA levels of the IRF3 and NFkB1 downstream genes, such as *IL6, IL8, RSAD2* (viperin), and *IRF1* was significantly reduced in FNE1-YAP$^{S127A}$ cells compared to that of FNE1-MX control cells. Consistently, the mRNA levels of type I interferons, such as *IFNA1, IFNA2, IFNB2,* and *IFNE*, were significantly reduced compared to the control cells (Fig. 7C, $P < 0.0001$). Western blot, at the protein level, confirmed that ectopic expression of YAP or YAP$^{S127A}$ suppresses the expression and/or activation of critical receptors, adapters, kinase, and transcription factors that are essential for the production of type I interferons in the presence or absence of the HPV-PsV (Fig. 7D).

Type I interferons induce the expression of a large spectrum of antiviral interferon-stimulated genes (ISGs) via the IFNαRs/JAKs/STATs signaling pathway (Ivashkiv and Donlin, 2014; Zhou et al, 2013). RT-PCR analyses indicated that ectopic expression of YAP$^{S127A}$ in FT246 cells suppressed expression of genes encoding major components of the IFNαRs/JAKs/STATs signaling pathway, including IFNα receptors (*IFNαR1*, $P < 0.05$; *IFNαR2*, $P < 0.01$, compared to that of FT246-MX control cells, Fig. 8A), JAK kinases (*JAK1* & *JAK2*, $P < 0.001$, compared to that of FT246-MX control cells, Fig. 8A), Signal Transducer and Activator of Transcription family members (*STAT1*, $P < 0.001$; *STAT2*, $P < 0.01$, *STAT4*, $P < 0.001$, Fig. 8A), and key transcriptional factor IRF9 ($P < 0.001$, Fig. 8A), which initiates the transcription of the IFN-stimulated genes (ISGs) to modulate cellular antiviral functions (Sun et al, 2010). Western blot results indicated that proteins of these signaling transducers are also suppressed in FNE1 cells expressing YAP$^{S127A}$ (Fig. 8B). Fluorescent immunohistochemistry showed that ectopic expression of YAP1 in FT246 cells not only reduced the intensity of STAT1 and IRF9 immunosignal in the nucleus, but also total intensity of STAT1 and IRF9 in these cells (Fig. 8C). Treatment of FT246 cells with high level of recombinant IFNα2b (50 IU, 24 h) rapidly increased nuclear STAT1 and IRF9 in FT246-MX cells, but failed to do so in the FT246-YAP$^{S127A}$ cells (Fig. 8C). The similar results were observed in FNE1 cells, which is a human-TERT immortalized Fallopian tube epithelial cells (Fig. EV3). These data indicate that hyperactivation of YAP1 blocks type I interferon action through inhibiting JAK/STAT signaling transduction. Importantly, results derived from analyzing human patient data extracted from GTEx and TCGA datasets demonstrated that expression of genes encoding key components of the innate immune signaling pathways, such as *DHX58, IRF3, STAT5A, STAT6,* and *JAK1*, are significantly downregulated in ovarian cancer tissues when compared to normal control (Fig. EV4), verifying the clinical relevance of our observations.

Consistent with these observations, pre-treatment of FT246-MX and FT246-YAP$^{S127A}$ cells with Ruxolitinib (a JAK1/2 inhibitor, 50 nM, 24 h) increased the ratio of GFP-positive cells in FT246-MX after infecting with 2.0 MOI HPV16-PsVs for 72 h ($P < 0.001$), indicating that the JAK/STAT signaling pathway plays an antiviral action in the fallopian tube epithelial cells (Fig. 8D). If we incubate FT246-MX and FT246-YAP$^{S127A}$ cells with 2.0 MOI HPV16-PsVs for 72 h after treatment with IFNα2b (50 IU, 24 h), IFNα2b greatly reduced the ratio of GFP-positive cells in FT246-Mx control cells, but not in FT246-YAP$^{S127A}$ cells, suggesting that hyper-activation of YAP1 indeed inhibited IFNα-induced antiviral effect in the fallopian tube epithelial cells (Fig. 8D; Appendix Fig. S9).

Type I Interferons induced binding of phosphorylated STAT1 and STAT2 with IRF9 to form a complex to initiate the transcription of several hundreds of so-called 'IFN-stimulated genes' (ISGs) (Seth et al, 2006; Sun et al, 2010). Since hyperactivated YAP1 suppressed the IFNαR/JAK/STAT signaling pathway in FTECs, we speculated that the expression of antiviral genes would be compromised by the hyper-activation of the YAP1 protein. Our analyses indicated that ectopic expression of YAP$^{S127A}$ in FNE1 cells downregulated antiviral ISGs examined, including *IRF1* (positive regulators of IFN signaling), *MX1, CH25H* (inhibitors of virus entry), and *IFIT1, OAS1, ISG15* (suppressors of virus translation and replication) (Appendix Fig. S10; Fig. EV5).

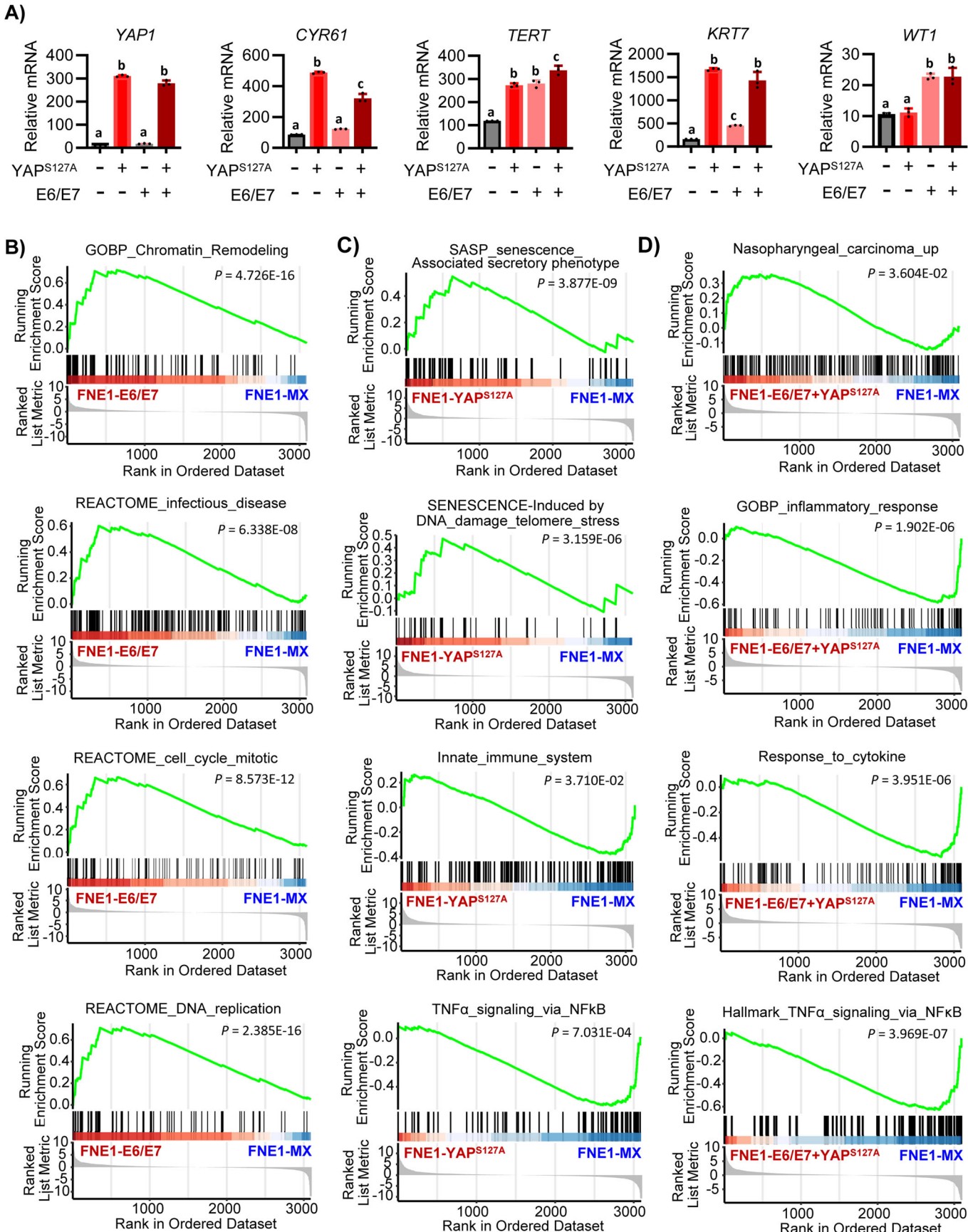

**Figure 6. Enrichment of genes and pathways in FNE1 cells with differential expression of YAP1 and HPV E6/E7 oncogenes.**

(A) Increased expression of YAP1 and HPV16 E6/E7 downstream genes provides evidence for the successful ectopic expression of YAP1 and HPV16 E6/E7 in FNE1 cells. Each bar represents the mean ± SEM ($n \geq 3$ technical replicates). Bars with different letters are significantly different from each other. Data were analyzed for significance using the two-way ANOVA followed by the Tukey's multiple comparisons post hoc test. A value of $P < 0.05$ was considered statistically significant. Exact $P$ values for the compared groups in each graph are presented with source data, which is available online. (B–D) Representative GSEA graphs showing the enriched genes and pathways in FNE1-E6/E7 (B), FNE1-YAP $^{S127A}$ (C), and FNE1-E6/E7-YAP$^{S127A}$ (D) cells. GSEA analyses were performed based on RNA-seq data from FNE1-eE6/E7 (B), FNE1-YAP$^{S127A}$, and FNE1-E6/E7-YAP$^{S127A}$ cells. Source data were uploaded to Gene Expression Omnibus (accession: GSE268836). The exact $P$ value for each assay is directly presented on the graph. Source data are available online for this figure.

Importantly, ectopic expression of YAP$^{S127A}$ also eliminated HPV16-PsVs-induced elevation of mRNA expression for *IRF1*, *MX1*, *CH25H*, *IFIT1*, *OAS1*, and *ISG15* in FNE1 cells (Fig. EV5A). Similarly, YAP$^{S127A}$ significantly compromised IFNa2-induced expression of ISGs, including *IFIT1*, *MX1*, *ISG15*, *IRF7*, *IRF3*, and *IRF1* (Fig. EV5B). These results prove that the constitutive activation of YAP1 suppresses IFNαRs/JAK/ STATs to reduce the production of antiviral ISGs.

## Discussion

The overall survival rate of ovarian cancer patients has improved only modestly in the past decades. This lack of progress can be attributed, in part, to the fact that the exact etiology and even the cell-of-origin of epithelial ovarian cancer is unclear. Accumulating evidence supports the concept that a significant proportion of ovarian HGSOC may originate from fallopian tube epithelial cells (Bowtell et al, 2015; Erickson et al, 2013; Kim et al, 2012; Perets et al, 2013). However, the molecular mechanisms underlying the transformation of fallopian tube epithelial cells and metastasis of fallopian tube-derived HGSOC to ovarian and other pelvic tissues are largely unknown. Using immortalized fallopian tube epithelial cell lines, we have shown that hyperactivation of YAP1, the major effector of the Hippo signaling pathway, induced malignant transformation of immortalized fallopian tube epithelial cells leading to the development of HGSOC (Hua et al, 2016). In contrast to the malignant transformation observed in the immortalized cells, we found that YAP1, similar to KRAS and BRAF oncogenes (Dimauro and David, 2010; Wajapeyee et al, 2008), induced cellular senescence in the cultured primary epithelial cells, including the primary culture of fallopian tube epithelial cells (FTECs) (Fig. 1). These findings hinted that under physiological conditions, cellular senescence may serve as a functional mechanism preventing fallopian tube epithelial cells from YAP1-induced malignant transformation. Obviously, other pathogenic factors could synergize with the disrupted Hippo pathway (and thereby hyperactivated YAP1 oncogene) to induce carcinogenesis of fallopian tube epithelium. In the present study, we found that HPV, a common sex-transmitted pathogenic virus and known causative agent of cervical cancer, prevented primary FTECs from natural replicative and YAP1-induced senescence and synergized with YAP1 to induce malignant transformation of primary fallopian tube epithelium cells. Moreover, HPV oncoproteins drove pelvic metastasis of YAP1-induced fallopian tube carcinoma. Evidence provided in the present study demonstrates that infection of high-risk HPV (hrHPV) may represent a

previously neglected pathogenic factor contributing to the initiation of HGSC from fallopian tube epithelial cells.

HPV infection of the female lower genital tract is common (Baseman and Koutsky, 2005). The most recent data from the National Center for Health Statistics showed that during 2013–2014, the prevalence of 37 any (high- and low-risk) and 14 high-risk genital HPVs for women aged 18–59 in the United States was 39.9% and 20.4%, respectively (McQuillan et al, 2017). HPV virions found in the lower portion of the female genital tract could reach the fallopian tube fimbriae via retrograde menstruation, sperm transmission, or surgical procedures. Retrograde menstruation is found in over 90% of menstruating patients during gynecological surgery (Heidarpour et al, 2017; Oppelt et al, 2010; Rocha et al, 2019; Vercellini et al, 2007). The presence of HPV in the semen has been frequently reported (Chan et al, 1996; Green et al, 1991; Ostrow et al, 1986). In the present study, using different detection methods, we confirmed the presence of HPV DNA in normal and cancerous fallopian tube tissues. Using a well-studied HPV pseudovirus and several fallopian tube and ovarian cancer cell, we found that HPV was capable of infecting normal and cancerous fallopian tube tissues. Our research results also indicated that HPV preferentially infects the cervical epithelial cells and cancerous fallopian tube epithelial cells. This is consistent with previous reports that HPV infection of epithelial cells requires cell cycle progression, and that cancer cells (usually highly proliferative) are more susceptive to HPV infection (Baseman and Koutsky, 2005; Doorbar, 2005; Pyeon et al, 2009).

HPV has been identified as a causative agent of cervical cancer (Viens et al, 2016; zur Hausen, 2002), but its impact on fallopian tube carcinogenesis remains unclear. Our recent study has shown that hyperactivated YAP1 could induce malignant transformation of immortalized fallopian tube secretory epithelial cells (Hua et al, 2016). Interestingly, we found that in the primary fallopian tube epithelial cells (FTECs), YAP1 promoted cellular senescence. This discrepancy indicated that under pathological conditions, unknown pathogenic factor(s) in the fallopian tube epithelium are capable of preventing the non-immortalized FTECs from YAP1-induced senescence. The results in the present study provide convincing evidence that HPV is a candidate pathogen that could help non-immortalized FTECs with hyperactivated YAP1 to circumvent YAP1-induced senescence. We found that in the absence of HPV, expression of YAP$^{S127A}$ promoted cellular senescence, while in the presence of HPV, hyper-activation of YAP1 not only stimulated FTEC proliferation but also drove their transformation. Intriguingly, results from the present study demonstrate that HPV also promotes the progression and drives metastasis of YAP-induced fallopian tube HGSC. For example, FNE1-E6/E7 cells couldn't form

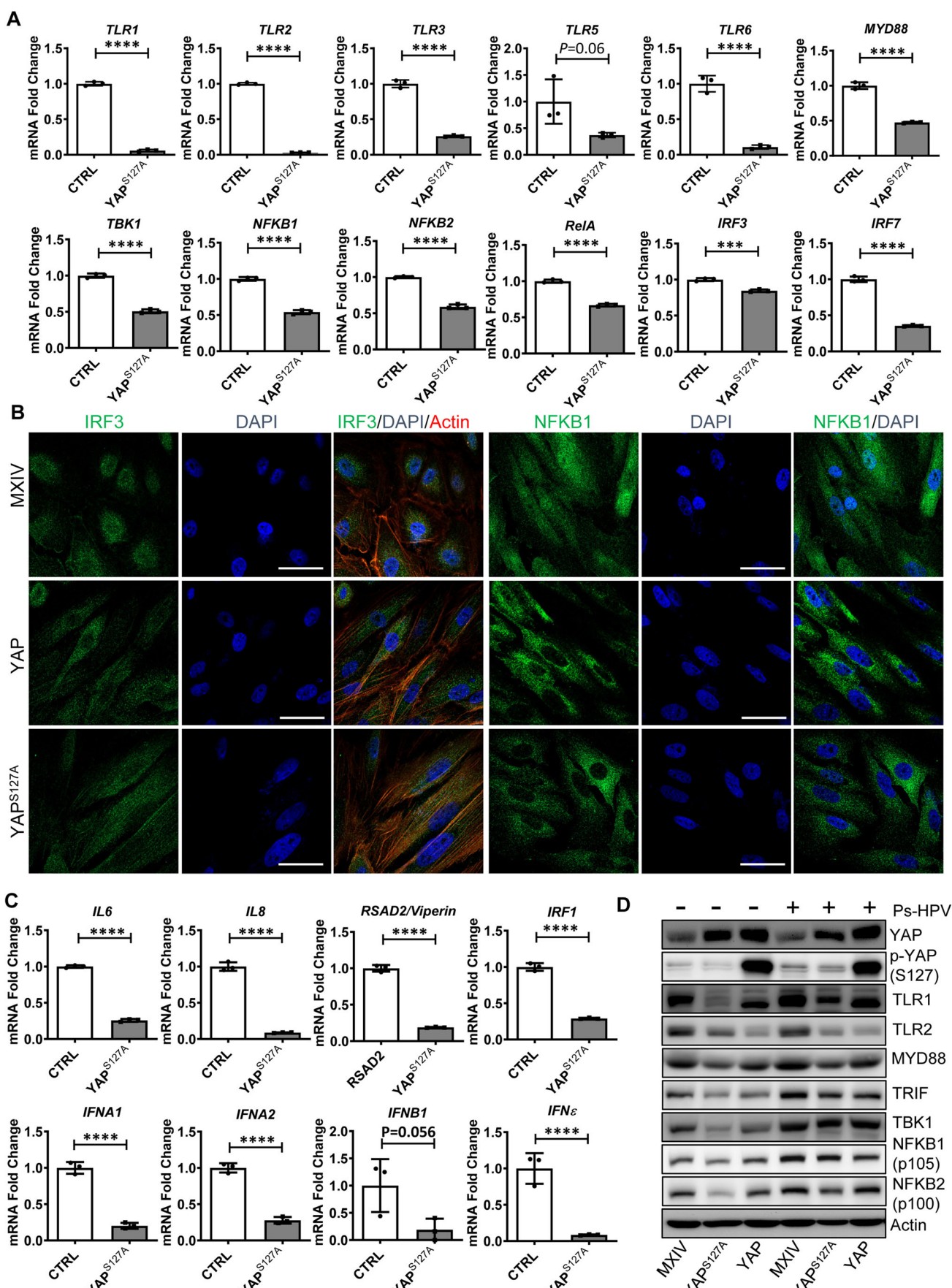

**Figure 7. Constitutive activation of YAP1 suppresses type I interferon production in FTECs.**

(A) Quantitative data showing mRNA levels of major components of viral recognition pathway in FNE1-MX (control or CTRL) and FNE1-YAP$^{S127A}$ cells. Each bar represented the mean + SEM ($n = 3$ technical replicates). Data were analyzed for significance using the unpaired t test. A value of $P < 0.05$ was considered statistically significant. ***$P < 0.001$; ****$P < 0.0001$, when compared to their control (CTRL). Exact p values for each gene: $P < 0.0001$ for *TLR1, TLR2, TLR3, TLR6, MYD88, TBK1, NFKB1, NFKB2, RelA,* and *IRF7*; $P = 0.0603$ for *TLR5*; $P = 0.006$ for *IRF3*. (B) Representative images selected from three biological replicates showing the expressions and locations of IRF3 and NFKB1 in FT246-MXIV, FT246-YAP- and FT246-YAP$^{S127A}$ cells. IRF3 and NFKB1 proteins were visualized using an Alexa-488 (green) conjugated secondary antibody. Actin filaments were stained with rhodamine-phalloidin (red). Nuclei were stained with DAPI (blue). Scale bar: 50 μm. (C) Quantitative data showing mRNA levels of downstream target genes of IRF3 and NFKB, including type I interferons, pro-inflammatory cytokines, and key antiviral factors in FNE1-MX (control) and FNE1-YAP$^{S127A}$ cells. Each bar represents the mean + SEM ($n = 3$ technical replicates). Data were analyzed for significance using the unpaired t test. A value of $P < 0.05$ was considered statistically significant. ****$P < 0.0001$, when compared to their control (CTRL). Exact p values for each gene: $P < 0.0001$ for *Il6, IL8, RSAD2, IRF1,* and *IFNA2*; $P = 0.0001$ for *IFNA1*; $P = 0.0561$ for *IFNB1*; $P = 0.0017$ for *IFNε*. (D) Representative western blots selected from three biological replicates showing protein levels of major components of the viral recognition and interferon production pathways in FNE1-MX (control), FNE1-YAP, and FNE1-YAP$^{S127A}$ cells with or without HPV16 pseudovirions treatment. Source data are available online for this figure.

tumors in athymic nude mice after subcutaneous implantation for nine months. FNE1-YAP$^{S127A}$ cells only form in situ carcinoma in athymic nude mice after subcutaneous implantation for over three months. However, the introduction of HPV E6/E7 into FNE1-YAP$^{S127A}$ cells not only promoted tumor formation and progression but also induced massive pelvic metastasis of cancer cells, leading to reduced survival rate of tumor-carrying mice. Cancer cells derived from FNE1-E6/E7-YAP$^{S127A}$ cells metastasized to the gastrointestinal system, the mesentery, pancreas, diaphragm, and peritoneal mesothelium. Peritoneal metastasis is a feature of human HGSOC (Kyriazi et al, 2010). Around 70% of human HGSOC are diagnosed at the advanced stage, a time when tumors have metastasized to many organs and tissues in the peritoneal cavity (Pradeep et al, 2014). Although extensive metastasis is associated with a poor survival rate, the mechanism(s) contributing to metastasis of HGSOC are yet to be fully described. Our observation that HPV drives massive pelvic metastasis (peritoneum mesothelium, omentum, diaphragm, and pancreas) of YAP1-induced fallopian tube HGSC provides a clue for our understanding of the metastasis of human HGSOC, and opens a window for developing new strategies to effectively prevent tumor metastasis by eradicating HPV infection and thereby improve the patient survival rate.

Humans develop a sophisticated immune system to control pathogenic infections. Upon infection, innate immunity serves as the host's first line of defense system to eradicate pathogens (Turvey and Broide, 2010). It is estimated that innate immunity accounts for 80–90% of all immune responses against pathogen invasion. Therefore, a common argument is that even if HPV reaches the fallopian tube fimbriae and invades the fallopian tube epithelium, these viruses will be eradicated by the host immune system. How HPV virion evades innate immune surveillance is an unanswered question. Previous studies showed that HPV developed a broad spectrum of immune evasion strategies to evade immune surveillance (Bhat et al, 2011; Steinbach and Riemer, 2018; Yang et al, 2005). However, the vast majority of virus and infected cells were eventually eliminated by the immune system in the female reproductive tract, even after the establishment of cervical intracellular neoplasia (CIN), suggesting that generally the immune system has the capability to clean up HPV and HPV-infected cells. Therefore, persistent HPV infection in certain individuals indicates that the host immunity in this group of people is incompetent. In the present study, we found that the hyperactivated YAP1 oncogene takes several measures to shut off the innate immune system to facilitate HPV evasion of innate immune surveillance in FTECs.

First, we found that hyperactivated YAP1 targets the viral recognition system of innate immunity. The recognition of a pathogen such as HPV by the innate immune system depends on the interaction between host cell germline-encoded pattern recognition receptors (PRRs) and pathogen-associated molecular patterns (PAMPs) of HPV. Toll-like receptors (TLRs) are the best-studied PRRs mediating HPV infection. The binding of host cell TLRs to viral PAMPs recruits and activates some adapter proteins, including MYD88 and TRIF (Di Paolo, 2014). However, hyper-activation of YAP in FETCs blocked expression of *TLR1, TLR2, TLR3, TLR5,* and *TLR6* genes. Moreover, the expression of YAP$^{S127A}$ in FTECs also significantly suppressed the expression of *MYD88* and *TRIF* (*TICAM1*) genes. Second, we found that hyper-activation of YAP1 suppressed the production of IFNs, key antiviral molecules involved in eradicating infected pathogens. Generally, recognition of foreign PAMPs initiates a series of signaling cascades that lead to the activation of TANK binding kinase 1 (TBK1) and subsequent stimulation of nucleus translocation of the transcription factors (NF-κB) and interferon regulatory factor 3 (IRF3). Ectopic expression of YAP$^{S127A}$ in FTECs suppressed the expression of almost all key components of the type I interferon production pathway, including *TBK1, NFkB1, NFkB2, RelA, IRF3,* and *IRF7*. Correspondingly, it significantly reduced transcription of antiviral type I interferon (*IFNA1, IFNA2, IFNB1,* and *IFNE*) and pro-inflammation cytokines such as *IL6* and *IL8*, which are critical factors for the eradication of invaded virus (e.g., HPVs) (Negishi et al, 2008; Stanley, 2012; Zhou et al, 2013). Finally, we found that hyper-activation of YAP in FTECs shut off the IFNARs-JAK/STAT-IRFs signaling pathway. Expression of YAP$^{S127A}$ in FTECs significantly suppressed expression of *IFAR1, IFAR2, JAK 1, JAK2, STAT1, STAT2,* and *IRF9*. The IFNARs-JAK/STAT-IRFs signaling pathway is critical for producing hundreds of interferon-stimulated genes (ISGs) and other antiviral proteins to protect host cells against invading pathogens (Seth et al, 2006; Sun et al, 2010). Clearly, hyperactivation of YAP1 suppressed the expressions of almost all key adapter proteins, kinase, and effectors critical for the activation of the innate immune in these cells, inhibited the innate antiviral signals from PRRs activation to ISGs production in FTECs, resulting in the increased susceptibility of FTECs to HPV infection, and suppressed immune response of FTECs to HPV invasion. Therefore, the hyperactivated YAP1 may represent a novel molecular mechanism by which high-risk HPV evades immune surveillance to promote HGSOC development in the fallopian tube epithelium.

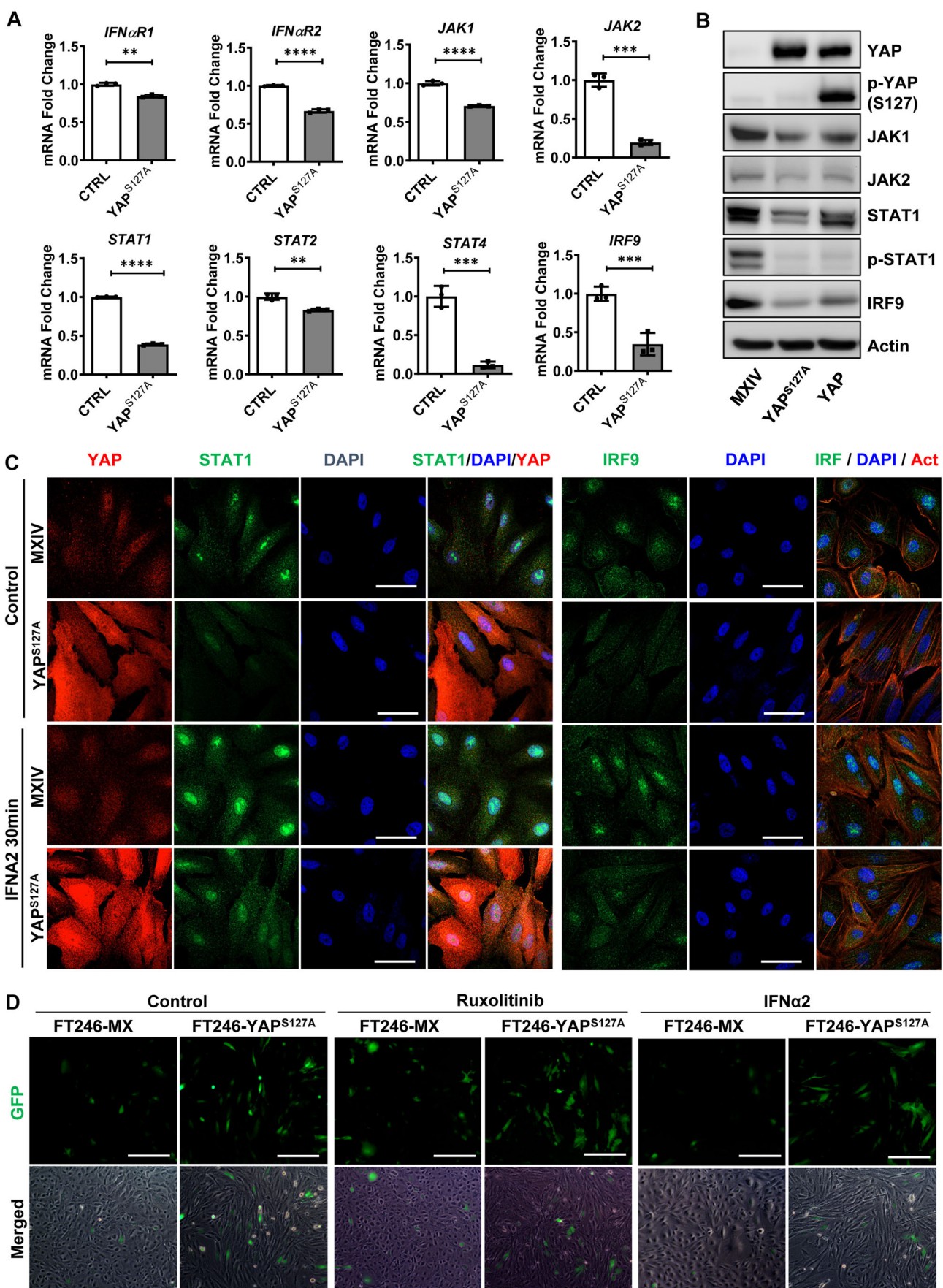

**Figure 8. Constitutive activation of YAP1 suppresses the JAK/STAT/IRF9 pathway in FTECs.**

(A) Quantitative data showing mRNA levels of major components of the IFNs/JAK/STAT/IRF9 pathway in control (FNE1-Mx) and FNE1-YAP$^{S127A}$ cells. Each bar represented the mean + SEM ($n = 3$ technical replicates). Data were analyzed for significance using the unpaired t test. A value of $P < 0.05$ was considered statistically significant. **$P < 0.01$; ***$P < 0.001$, when compared with control (CTRL). Exact p values for each gene: $P = 0.0006$ for *IFNaR1*; $P < 0.0001$ for *IFNaR2*; $P < 0.0001$ for *JAK1*; $P = 0.0001$ for *JAK2*; $P < 0.0001$ for *STAT1*; $P = 0.0024$ for *STAT2*; $P = 0.0004$ for *STAT4*; $P = 0.0028$ for *IRF9*. (B) Representative blots selected from three biological replicates showing expression and activation of key proteins and kinases in the JAK/STAT/IRF9 pathway in FNE1-MX (control), FNE1-YAP, and FNE1-YAP$^{S127A}$ cells. (C) Representative images selected from three biological replicates showing the expressions and locations of STAT1 and IRF9 in FT246-MX (control) and FT246-YAP$^{S127A}$ cells in the presence or absence of IFNα2b. STAT1 and IRF9 proteins were visualized using an Alexa-488 (green) conjugated secondary antibody. In the left panel, STAT1 and IRF9 proteins were visualized using an Alexa-488 (green) conjugated secondary antibody, while YAP was visualized using an Alexa-594 (red) conjugated secondary antibody. In the right panel, IRF9 proteins were visualized using an Alexa-488 (green) conjugated secondary antibody. Actin was stained with rhodamine-phalloidin (red). Nuclei were stained with DAPI. Scale bar: 30 μm. (D) Representative images selected from three biological replicates showing HPV16 PsV-derived GFP signal in control FT246-MX and FT246-YAP$^{S127A}$ cells in the presence or absence of Ruxolitinib (JAK inhibitor) or recombinant human interferon alpha 2b (IFNα2b). GFP signal indicates the infection efficiency of HPV16 PsV in these cells. Scale bar: 100 μm. Quantitative results of the GFP signal are presented in Appendix Figure S9. Source data are available online for this figure.

High-risk HPV is considered a causative agent of cervical cancer because >95% of cervical cancer tissues are HPV DNA positive. A significant concern on the role of HPV in HGSOC is that high-risk HPV DNA was not frequently detected in human ovarian tumor tissues, which is mirrored by the results presented in Fig. 2. However, evidence from early studies has already shown that HPV DNA integration into the host genome is not even required for the transformation of cervical epithelial cells. Hudelist et al found that in the high grade lesion (CIN3) and carcinoma in situ (CIS), around 50% HPV16 and 10% HPV18 were present in the non-integrated episomal form (Hudelist et al, 2004). Similar observations have been reported by many other research groups (Cullen et al, 1991; Fukushima et al, 1990; Hong et al, 2017; Li et al, 2008; Park et al, 1997). Moreover, a portion of invasive cervical cancer only had the purely episomal HPV (Cullen et al, 1991; Fukushima et al, 1990; Hong et al, 2017). These observations suggest that non-integrated episomal HPV is able to induce tumorigenesis. Since Fallopian tube only occasionally exposed to limited amount of HPV (e.g., via sperm transmission or retrograde menstruation), the chance of HPV DNA integrates into chromosomes of fallopian tube epithelial cells is very low, and HPV will be more likely exist as episomal form in these infected cells. Suppose HPV infection coincides with the disruption of the Hippo/YAP signaling in the fallopian tube epithelium. In that case, the episomal form of HPV can assist FTECs to overcome the hyperactivated YAP1-induced cellular senescence leading to the malignant transformation of these cells (the "hit" step). Previous studies has already demonstrated that impairment of the DNA-damage response (DDR) pathway and/or inactivation of TP53 could lead to the loss of HPV episomes in HPV immortalized cells (Edwards et al, 2013; Fisher, 2015; Templeton and Laimins, 2023). Since the impaired DDR signaling and mutated *TP53* are molecular features of HGSOC, the HPV episome in the transformed FTECs will gradually loss with tumor progression/evolution (the "run" step). Results in the present study support the concept that high-risk HPV may contribute to the development of at least a portion of HGSOC via a "hit & run" mechanism. We need to develop better in vitro and in vivo experimental models and design more ingenious experiments to verify this hypothesis.

In conclusion, our study indicates that HPV could be an important pathogen for HGSOC development. Results in the present study provide more in-depth insights into the etiology of HGSOC and have significant implications for the prevention, early

detection, and therapeutic intervention of these deadly malignancies. First of all, the identification of HPV as a tumorigenic pathogen of ovarian HGSC provides a potential explanation for many previously unexplained observations. For example, although previous data showed that salpingectomy and tube ligation could reduce ovarian cancer incidence (Duus et al, 2023), the underlying reasons for this phenomenon have not been resolved. Our study suggested that salpingectomy and tube ligation may block HPV infecting the fallopian tube epithelium, leading to a reduction of HPV-associated FTEC transformation. Another example is the ovulation theory of ovarian cancer development. Epidemiological studies showed that ovulation rate is strongly associated with ovarian cancer incidence. It should be considered that monthly ovulation, which can induce damage to the fallopian tube epithelium (by follicular fluid), may also cause a higher HPV infection rate. Moreover, the identification of HPV as a critical tumorigenic pathogen of fallopian tube-derived HGSOC may help us to prevent this disease by reducing HPV prevalence (e.g., sexual education) and promoting HPV vaccination. In addition, the identification of HPV as a critical tumorigenic pathogen for fallopian tube-derived HGSOC may facilitate the early detection of this disease. Finally, the carcinogenic interaction between YAP1 and HPV in the FTECs suggests that targeting HPV and YAP has the potential to improve the treatment of HGSOC with tubal origin.

## Methods

### Chemicals, cell lines, and human fallopian tissues

Interferon alpha 2b (IFNα2b) and Ruxolitinib were from R&D Systems Inc. (Minneapolis, MN). DMEM/F12 and other cell culture media were from Invitrogen (Carlsbad, CA). Fetal bovine serum (FBS) was from Atlanta Biologicals, inc. (Lawrenceville, GA). The FOMI medium was from Live Tumor Culture Core at Sylvester Comprehensive Cancer Center (Miami, FL). Ultroser™ Serum Substitute was purchased from Pall Corporation. iScript Reverse Transcription Supermix for RT-qPCR and iTaq™ Universal SYBR® Green Supermix were from Bio-Rad Laboratories, Inc. (Hercules, CA); RNeasy Mini Kit was purchased from QIAGEN Inc. (Valencia, CA). YAP siRNA (E-012200-00-0005) and ITGA6 siRNA (E-007214-00-0005) were from Dhamarcon/Thermo Scientific (Pittsburgh, PA). Lentivirus containing HPV16 E6/E7

gene (LV617) was from Applied Biological Materials (ABM) Inc. (Richmond, BC, Canada). PCR chemicals were from Invitrogen (Carlsbad, CA), QIAGEN (Carlsbad, CA), or Bio-Rad (Hercules, CA). Antibodies against YAP (#4912), phospho-YAP (Ser127) (#4911), TLR1 (# 2209), TLR2 (#1227), MYD88 (#4283), TRIF (#4596), TBK1 (#3504), NF-κB1 p105/p50 (#12540), NF-κB2 p100/p52 (#3017), JAK1 (#3344), JAK2 (#3230), STAT1 (#9172), STAT2 (#72604), IRF3 (#11904), IRF9 (#76684), phospho-STAT1 (#7649) and phospho-STAT2 (#88410) were from Cell Signaling Technology Inc. (Danvers, MA). β-actin antibody (A5441) was from Sigma-Aldrich (St. Louis, MO). Fluorescence-conjugated secondary antibodies for immunofluorescent analysis were from Jackson Immunoresearch Laboratories Inc. (West Grove, PA); The Super-Signal West Femto Chemiluminescent Substrate Kit for Western blotting was from Pierce/Thermo Scientific (Rockford, IL); Optitran nitrocellulose transfer membrane was from Schleicher & Schuell Bioscience (Dassel, Germany). PVDF Membrane for Western Blotting was from Sigma-Aldrich (St. Louis, MO). All other General Chemical Reagents were purchased from Sigma (St. Louis, MO), Fisher (Pittsburgh, PA), or United States Biochemical (Cleveland, OH).

ME180, HT3, Hela, TOV21G, SW626, SKOV3, OVCAR3, OVCAR5, OVCAR8, COV362.4, CAOV3, SK-UT-1, KLE, AN3-CA, MES-SA, and SK-LSM-1 cells were from ATCC (Manassas, VA). Immortalized Fallopian tube epithelial cell lines FT190 and FT246 were from Dr. Ronny Drapkin's Lab (University of Pennsylvania, Philadelphia, PA). These cells were established with a protocol described previously (George et al, 2016; Karst and Drapkin, 2012; Karst et al, 2011). Kurumochi and Ovsaho cell lines were from Dr. Adam Karpf's Lab (University of Nebraska Medical Center, Omaha, NE). All cell lines were recently authenticated and tested for free of mycoplasma contamination before the experiment. hTERT immortalized FNE1 cells were from the Live Tumor Culture Service Center at the Interdisciplinary Stem Cell Institute, University of Miami (Miami, FL). Primary FTEC cells (hFTEC) were either from Lifeline Cell Technology (Frederick, MD) or isolated from primary human fallopian tube tissue as described previously (George et al, 2016; Karst and Drapkin, 2012). Primary human cervical epithelial cells (hCerEC) were purchased from ScienCell Research Laboratories, inc. (Carlsbad, CA). The chronically inflammatory fallopian tube tissues and fallopian tube carcinoma tissues were purchased from the UNMC tissue bank (under an approved IRB). A total of 20 fallopian tube carcinoma samples and 10 chronically inflammatory fallopian tube tissues were used for HPV detection.

### Detection of HPV DNA in human fallopian tube tissues and cell lines by PCR

HPV DNA in human fallopian tube tissues and cell lines was detected by polymerase chain reaction (PCR) using HPVL1 degenerated primer pairs MY09/MY11 and consensus primer pairs GP5+/GP6+. Genome DNA was extracted from 20 μg fallopian tube tissues or $1 \times 10^7$ cultured cells. DNA concentrations were determined using NanoDrop2000 (Thermo Scientific). DNA from HPV-positive cervical cancer cell lines (ME180 and Hela) was used as positive control. DNA from HPV-negative cervical cancer cell line (HT3) and water were used as negative controls. Previous studies have verified the effectiveness of MY09/MY11 and GP5+/

GP6+ primers (Fuessel Haws et al, 2004). MY09/MY11 generated a ~450 bp PCR product. Nested PCR followed by primers GP5+/GP6+ amplified a ~150 bp PCR product. The PCR reaction mixture contained 1x PCR Master Mix (Promega, Madison, WI), 3 mM MgCl₂, 5U Taq DNA Polymerase, 300 nM of each primer, and 500 ng DNA template. Each sample was tested three times. Amplification cycling with MY09/MY11 primers: Starting at 94 °C for 5 min and then followed by 40 amplification cycles. Each cycle includes denaturation: 1 min at 95 °C; annealing: 1 min at 55 °C; and elongation: 1 min at 72 °C. The final extension followed the last cycle: 10 min at 72 °C. For nested PCR, the reaction mixture contained 1XPCR Master Mix, 5U Taq DNA Polymerase, 3 mM MgCl₂, 100 nM GP5+/GP6+ primers, and 5 μl PCR products derived from MY09/MY11 primers. The amplification cycling with GP5+/GP6+ primers is the same as MY09/MY11 primers except for the annealing step, which was performed at 40 °C for 2 min. PCR products were analyzed on a 2% agarose gel, stained with GelRed Nucleic Acid Stain (Phenix Research Products, Candler, NC), and visualized by UV trans-illumination (UVP, Upland, CA).

### The RNAscope HPV assay

The RNAscope HPV assay, which allowed direct visualization of E6/E7 mRNA in situ, was used to detect whether HPV16 DNA in the fallopian tube cancer is transcriptionally active. Eleven FPPE serous tubal intraepithelial carcinoma slides (HGSOC early lesion) were kindly provided by our collaborator Dr. Christopher Crum at Dana-Farber Cancer Institute. HPV16 E6/E7 were detected using an RNAscope™ 2.5 VS Probe-HPV16 kit (Advanced Cell Diagnostics, Inc., Newark, CA) following the protocol provided by the manufacturer. which can detect HPV16 E6/E7 mRNA. The E6/E7 mRNA signal was imaged using a PANNORAMIC 1000 high-end whole slide digitalization system (3DHISTECH, Budapest, Hungary). The Siha cell xenograft tumor tissue, which is HPV16 E6/E7 positive, was used as a positive control.

### Western blot analysis

Protein levels were determined using Western blot with a protocol described in previous reports (He et al, 2019; Huang et al, 2022; Lv et al, 2020). Briefly, control or treated cells were washed with pre-cold PBS, harvested on ice with ice-cold lysis buffer, briefly sonicated on ice, and centrifuged for 20 min at 12,000 rpm at 4 °C. The supernatant was collected into a new ice-cold tube, and the protein concentration of these samples was determined using a Pierce™ BCA protein assay kit. Samples for western blot were diluted with 6x Laemmli buffer and water to a final concentration of 1 μg/μl, boiled at 90 °C for 5 min, loaded (30 μl) onto a 10% SDS-PAGE gel, and fractioned with a Bio-Rad electrophoresis system. After transferring protein onto the nitrocellulose membranes, the membrane was blocked with 5% BSA at room temperature for 60 min before incubating with specific primary antibodies at 4 °C overnight and corresponding Horseradish peroxidase (HRP)-conjugated secondary antibodies at room temperature for 1 h. The immunosignal was generated using the Thermo Scientific SuperSignal West Femto Chemiluminescent Substrate Kit, and the images were captured and analyzed using a UVP gel documentation system (UVP, Upland, CA).

## Quantitative real-time PCR

Total RNA was extracted by combining the TRIzol protocol (Invitrogen; Carlsbad, CA) with the QIAGEN RNeasy mini kit (QIAGEN, Carlsbad, CA). Briefly, $1 \times 10^7$ cells were lysed with 0.65 ml TRIzol reagent for 10 min at room temperature before adding 300 μl chloroform. The mixture was vortexed for 15 s, kept static for 3 min, and then centrifuged samples at 12,000 rpm for 3 min at room temperature. The supernatant was carefully transferred into another DNase/RNase-free centrifuge tube, mixed well with an equal volume of 70% ethanol by pipetting, and loaded to the RNeasy Mini spin column provided in the QIAGEN RNeasy mini kit. The column with samples was kept static for 2 min before centrifuging for 15 s at 12,000 rpm. Discard the flow-through. After washing, the spin column was placed in a new 2 ml collection tube and dried by spinning at 15,000 rpm for 5 min. RNA on the membrane will be eluted with 40 μl RNase-free water (supplied in kit). Reverse transcription was performed using an iScript Reverse Transcription Supermix for RT-qPCR Kit (Bio-Rad Laboratories, Inc.). qT-PCR was performed in a Bio-Rad CFX96 real-time PCR system using iTaq™ Universal SYBR® Green Supermix Kit (Bio-Rad Laboratories, Inc.). GAPDH was used as a loading control. All primer sequences are presented in appendix (Appendix Table S1).

## Establishment of YAP overexpressing and knockdown cell lines

Primary hCerEC cells were cultured following the protocol provided by the vendor (Catalog #7060, ScienCell Research Laboratories, Inc.). Primary FTECs, FT194, FT246, and FT190 cells were cultured in DMEM/F12 medium with 2% Ultroser™ G serum substitute (Pall Corporation). FNE1 cells were cultured with FOMI medium as described in a previous report (Karst and Drapkin, 2012). For ectopic YAP1 expression, primary FTECs, FT190, FT194, FT246, and FNE1 cells were cultured to 40% confluent and then transfected with retrovirus-based empty control vector (MXIV), or vectors expressing wild type of YAP1 (YAP), or constitutively active YAP1 (YAP$^{S127A}$, a replacement of Serine at residue 127 with Alanine resulting in the constitutive activation of YAP1 protein (Dong et al, 2007). All transfected cells were selected with G418 (200–400 μg/ml). YAP1 expression in these cells was confirmed by RT-PCR and Western blot. For gene knockdown studies, siRNAs of *YAP1* (Accell YAP1 siRNA, E-012200-00-0005) and *ITGA6* (Accell ITGA6 siRNA, E-007214-00-0005) were synthesized by Dharmacon (Lafayette, CO). Primary or immortalized cells (60% confluence) were transfected with siNeg (Accell human non-targeting negative control siRNA, K-005000-R1), siYAP (*YAP1* siRNAs), or siITGA6 (ITGA6 siRNA) using a Lipofectamine 2000 protocol (Invitrogen, Carlsbad, CA). Knockdown of YAP1 and ITGA6 was confirmed by RT-PCR and Western blot.

## Cell senescence, proliferation, and colony formation assays

Cellular senescence was detected using an SA-β-Gal staining kit following the manufacturer's instructions (#9860, Cell Signaling Technology Inc.). After incubating with the β-galactosidase solution for 16 h on the plate, the senescent signal (senescence cells with blue color) was evaluated under a microscope (×20 magnification).

Cell proliferation was determined by counting cell numbers with an Invitrogen Countess® automated cell counter (Carlsbad, CA). Briefly, cells were detached using 5% trypsin. The suspended cell (10 μL) was mixed with 0.4% trypan blue stain (10 μL, supplied with the Countess™ cell counting chamber slides) and loaded onto the chamber of the sample slide (# C10315, Invitrogen) for automatic cell number counting. To ensure the counting accuracy, the cellular concentration was adjusted within a range between $1 \times 10^5$ cells/mL to $4 \times 10^6$ cells/ml.

Soft agar colony formation was used to examine the ability of anchorage-free growth, a feature of malignantly transformed cells. The assay was performed using a Cytoselect 96-well Cell Transformation assay kit (#CBA-130, Cell Biolabs, Inc., San Diego, CA). Briefly, the basal agar layer was prepared by mixing 1.2% agar solution with an equal volume of 4% Ultroser™ G/2x DMEM/F12 medium (1:1), distributing the mixture immediately to each well of a 96-well plate (50 μL/well), incubating at 4 °C for 30 min, and then warming up at 37 °C for 15 min. The cell suspension was prepared by mixing an equal volume of cells (adjusted the cell concentration to $1 \times 10^5$ cells/mL), 1.2% agar solution, and 4% Ultroser™ G/2x DMEM/F12 medium. A total of 75 μL of the cell suspension was transferred immediately to the 96-well plate containing the base agar layer and incubated at 4 °C for 20 min to make the cell and agar layer solid. Added 120 μL of cell growth medium into each well and cultured cells under standard culture conditions with a 72 h medium change interval. Colony formation was monitored daily under a microscope. After incubating for nine days, the yellow tetrazolium MTT (3-(4, 5-dimethylthiazolyl-2)-2,5-diphenyltetra-zolium bromide) (12 μL, #30-1010 K, ATCC, Manassas, VA) was added to the culture to stain cells. Images were taken under a microscope (4x) and analyzed with the ImageJ software (https://imagej.nih.gov/ij/).

## In vivo tumorigenicity

All animal handling and experimental procedures were approved by the Institutional Animal Care and Use Committee (IACUC) of the University at Nebraska Medical Center (UNMC) and Massachusetts General Hospital (MGH). FNE1-CTRL, FNE1-E6/E7, FNE1-cMyc, FNE1-E6/E7-cMyc, FNE1-MX, FNE1-YAP$^{S127A}$, and FNE1-E6/E7-YAP$^{S127A}$ cells ($6 \times 10^6$ cells suspended in 0.1 mL PBS + 0.1 mL Matrigel matrix (#354248, Corning Inc., Corning, NY)) were injected subcutaneously into the both dorsal flank of Athymic Nude Mouse (6-week-old, female, from charle river). All mice carrying injected cells were carefully monitored and euthanized 9 months (or when there was a wide range of metastases or ascites) after cell injections. Tumors were collected, weighed, and processed to prepare frozen and formalin-fixed paraffin sections.

## Preparation of HPV pseudovirions

HPV pseudovirions (HPV16-GFP) were prepared using a well-established protocol described previously (Buck and Thompson, 2007; Cardone et al, 2014). Briefly, 293FT cells (~60% confluent) were co-transfected with plasmids expressing HPV capsid proteins L1 and L2 (p16L1L2, #45291, Addgene), and GFP-expressing reporter plasmids (pCIneoEGFP, #46949, Addgene) using a

Lipofectamine 2000 (Invitrogen) protocol. The transfected cells were harvested 48 h after transfection, washed with DPBS-Mg solution (DPBS supplemented with 9.5 mM MgCl$_2$ and antibiotic-antimycotic mixture from Invitrogen) for three times, re-suspended in DPBS-Mg solution supplemented with 0.5% Brij58, 0.2% Benzonase (EMD Chemicals, Gibbstown, NJ), and 0.2% Plasmid Safe (Epicentre Biotechnologies, Madison, WI) at a concentration of $100 \times 10^6$ cells/ml, and incubated at 37 °C for another 24 h for capsid maturation. Cells were then lysed, and the cell lysate (the salt concentration was adjusted to 850 mM NaCl before ice incubation) was chilled on ice for 10 min. The chilled lysate was then clarified by centrifugation. The supernatant was layered onto an Optiprep gradient. The gradient was spun for 4.5 h at 16 °C at 40,000 rpm in a SW40 rotor (Beckman Coulter, Inc., Brea, CA). The HPV pseudovirions were collected, and the concentration of HPV pseudovirions was determined by viral plaque assays and analyzing GFP transduction efficiency in 293 cells.

### HPV16 pseudovirions infection of FTECs cells

Cells were plated in 12-well plates at a density of 10,000 cells per well and grown overnight before being infected with HPV16 pseudovirions (0.1 to 5.0 MOI) for 6–72 h. Infectivity was evaluated by examining the GFP signal, which was captured using a Zeiss 710 Meta Confocal Laser Scanning Microscope and analyzed using Zeiss Zen 2010 software (Carl Zeiss Microscopy, LLC, Thornwood, NY).

### Immunohistochemistry

YAP, MYC, KRT7, PAX8, WT1, and TP53 protein expression in tumors were detected using a peroxidase-based immunohistochemistry kit (VECTASTAIN ELITE ABC KITS, Vector Laboratories, Burlingame, Ca). Briefly, tumor tissues were deparaffinized with xylene and rehydrated with graded ethanol series. Antigens retrieval was performed in the Citrate-based unmasking solution (H-3300, Vector Laboratories, Burlingame, CA) with a pressure cooker. Endogenous peroxidase activity was quenched with 3% hydrogen peroxide for 30 min. After washing with PBS for 5 min, tissues were blocked with the blocking solution for 30 min at room temperature. Tissues were then incubated with primary antibodies at 4 °C for 16 h, washed with PBS for 3 × 5 min, and incubated with Biotinylated secondary antibody for 1 h at room temperature. The signal was visualized with VECTASTAIN ELITE ABC reagent (supplied in the kit) and peroxidase substrate solution (ImmPACT™ DAB kit, #SK4105, Vector Laboratories, Burlingame, CA), counterstained with Mayer's hematoxylin, scanned with an iSCAN Coreo Slide Scanner (Ventana Medical Systems, Inc., Oro Valley, AZ), and analyzed by ImageScope software (Leica Biosystems Imaging, Inc., Vista, CA).

Fluorescent immunohistochemistry was used to detect the expression and location of YAP1, IRF3, IRF9, NFKB1, and STAT1 in the established cell lines. Briefly, cells were seeded on coverslips (#12-545-80, Fisher Scientific) and incubated at 37 °C in 24-well plates. After reaching 40–60% confluent, cells were harvested, washed with pre-chilled PBS, and fixed with 4% formaldehyde in ice-cold 1X PBS for 20 min. Fixed cells were washed three times with ice-cold 1X PBST, blocked in 10% normal donkey serum at room temperature for 60 min, and incubated with diluted primary antibodies at 4 °C for 16 h before washing and incubating with diluted fluorochrome-conjugated secondary antibodies at room temperature (protected from light) for 1 h. After washing away the unbonded secondary antibody, cells on the slides were mounted with Fluoromount-G (#0100-01, SouthernBiotech, Birmingham, AL). Images were captured using a ZEISS Xradia 810 Ultra Confocal Laser Scanning Microscope and analyzed using Zeiss Zen 2012 software (Carl Zeiss Microscopy, LLC, Thornwood, NY).

### Next-generation RNA sequencing and gene set enrichment analysis (GSEA)

Ectopic expression of YAP1 and E6/E7 in FNE1-MX, FNE1-E6/E7, FNE1-YAP$^{S127A}$, and FNE1-E6/E7-YAP$^{S127A}$ cells was confirmed by Western blot. RNA was extracted from these cells and treated with DNase I. High-quality libraries were prepared and sequenced using Illumina HiSeq 4000 next-generation sequencer at the Next-Generation Core at the University of Nebraska Medical Center. Sequencing alignment and GSEA were performed as described previously (Huang et al, 2022). Reads were aligned with HISAT2 and assembled and quantified with Cufflinks. Differentially expressed genes (DEG) were revealed using edgeR 3.36.0 in R ver. 4.1.2. GeneSet Enrichment Analysis (GSEA) was used to identify key genes and pathways as described previously (Huang et al, 2022) and performed with fgsea 1.20.0 on MSigDB database 7.5.1, and visualized using clusterProfiler 4.2.2 and Complex-Heatmap 2.10.0 (Huang et al, 2022)

### Statistical analysis

All experiments were repeated at least four times unless otherwise noted. Data are presented as mean ± SEM of at least three technical replicates for each data point. Statistical analyses were conducted, and graphs were made with GraphPad Prism software (GraphPad Software, Inc., La Jolla, CA). Data were analyzed for significance using the Student's *t*-test (comparing one factor between two groups), one-way ANOVA with Tukey's post hoc tests (comparing one factor within multiple groups), or two-way ANOVA (comparing two factors within multiple groups). A value of $P < 0.05$ was considered statistically significant.

## Data availability

The datasets produced in this study are available in the following databases: RNA-Seq data: Gene Expression Omnibus accession GSE268836: (https://www.ncbi.nlm.nih.gov/geo/query/acc.cgi?acc=GSE268836).

The source data of this paper are collected in the following database record: biostudies:S-SCDT-10_1038-S44319-024-00233-3.

## Peer review information

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

## Acknowledgements

This work was supported by the National Cancer Institute/National Institute of Health (2R01CA197976, 1R01CA201500, 1R01CA279385), the Vincent Memorial Hospital Foundation/Vincent Center for Reproductive Biology, the Olson Center for Women's Health (no number), the Colleen's Dream Foundation (no number), the Ruggles Family Foundation, the Barbara Learned Bridge Funding Award from the Marsha Rivkin Center for Ovarian Cancer Research, and the Fred & Pamela Buffett Cancer Center (Lb595). JSD is the recipient of a Veterans Administration Senior Research Career Scientist Award. We thank the support from Next-Generation Core and Confocal Microscope Core at the University of Nebraska Medical Center for the deep sequencing and initial data analysis services.

## Author contributions

**Chunbo He**: Conceptualization; Data curation; Formal analysis; Investigation; Visualization; Methodology; Writing—original draft. **Xiangmin Lv**: Conceptualization; Data curation; Formal analysis; Validation; Investigation; Visualization; Methodology. **Jiyuan Liu**: Data curation; Formal analysis; Validation; Investigation; Visualization; Methodology; Writing—review and editing. **Peichao Chen**: Investigation; Writing—review and editing. **Jinpeng Ruan**: Validation; Investigation; Visualization; Methodology; Writing—review and editing. **Cong Huang**: Validation; Investigation. **Peter C Angeletti**: Resources; Investigation; Writing—review and editing. **Guohua Hua**: Validation; Investigation; Writing—review and editing. **Madelyn Leigh Moness**: Validation; Investigation. **Davie Shi**: Investigation. **Anjali Dhar**: Investigation. **Siyi Yang**: Data curation. **Savannah Murphy**: Investigation. **Isabelle Montoute**: Validation; Investigation. **Xingcheng Chen**: Investigation. **Kazi Nazrul Islam**: Validation; Investigation. **Sophia George**: Resources. **Tan A Ince**: Resources. **Ronny Drapkin**: Resources. **Chittibabu Guda**: Software; Investigation; Visualization. **John S Davis**: Resources; Writing—review and editing. **Cheng Wang**: Conceptualization; Resources; Data curation; Formal analysis; Supervision; Funding acquisition; Validation; Investigation; Visualization; Methodology; Writing—original draft; Project administration; Writing—review and editing.

Source data underlying figure panels in this paper may have individual authorship assigned. Where available, figure panel/source data authorship is listed in the following database record: biostudies:S-SCDT-10_1038-S44319-024-00233-3.

## Disclosure and competing interests statement

The authors declare no competing interests.

# Expanded View Figures

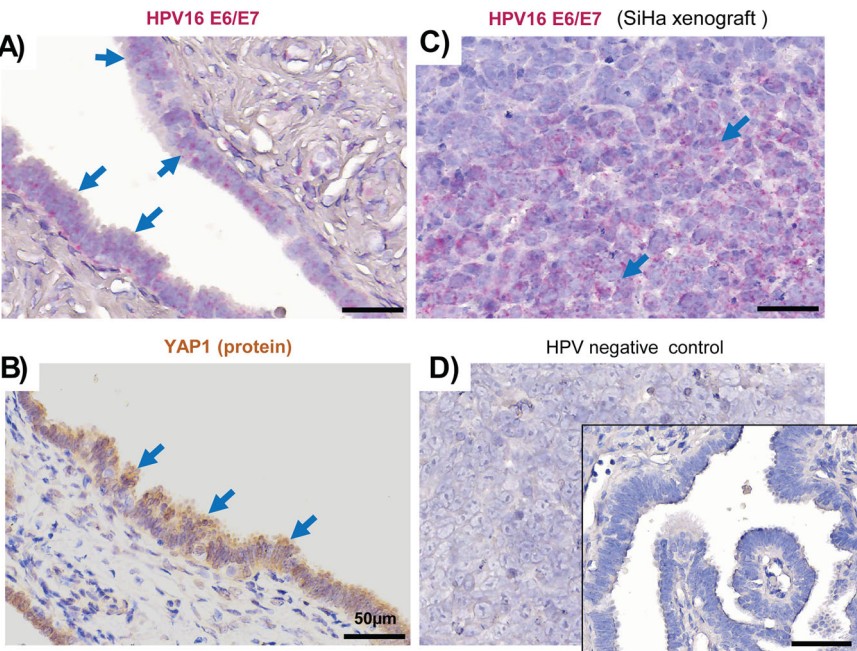

**Figure EV1.  Expression of HPV16 E6/E7 in the Fallopian tube STIC lesion (precursor of HGSOC) detected by RNA scope.**

(**A**) A representative image showing the expression of HPV16 E6/E7 mRNA (in pink) in fallopian tube STIC lesion (arrow) of a human patient (sample-GU980150-E10). E6/E7 were detected and visualized using the RNA scope technique. Scale bar: 50 μm. (**B**) A representative image showing the expression of YAP1 protein (in brown) in fallopian tube STIC lesion. YAP1 protein was detected and visualized by immunohistochemistry. Arrows point to neoplastic growth of epithelial cells with nuclear YAP1 protein. Scale bar: 50 μm. (**C**) A representative image showing the expression of HPV16 E6/E7 mRNA (in pink) in SiHa cell xenograft tumor tissues (positive control). Blue arrows point to the HPV16 E6/E7 positive cells (in pink). Scale bar: 50 μm. (**D**) Representative images showing negative staining (non-targeting probe) of HPV16 E6/E7 mRNA in SiHa cell xenograft tumor tissues and human STIC lesion (insert). Scale bar: 50 μm. Source data are available online for this figure.

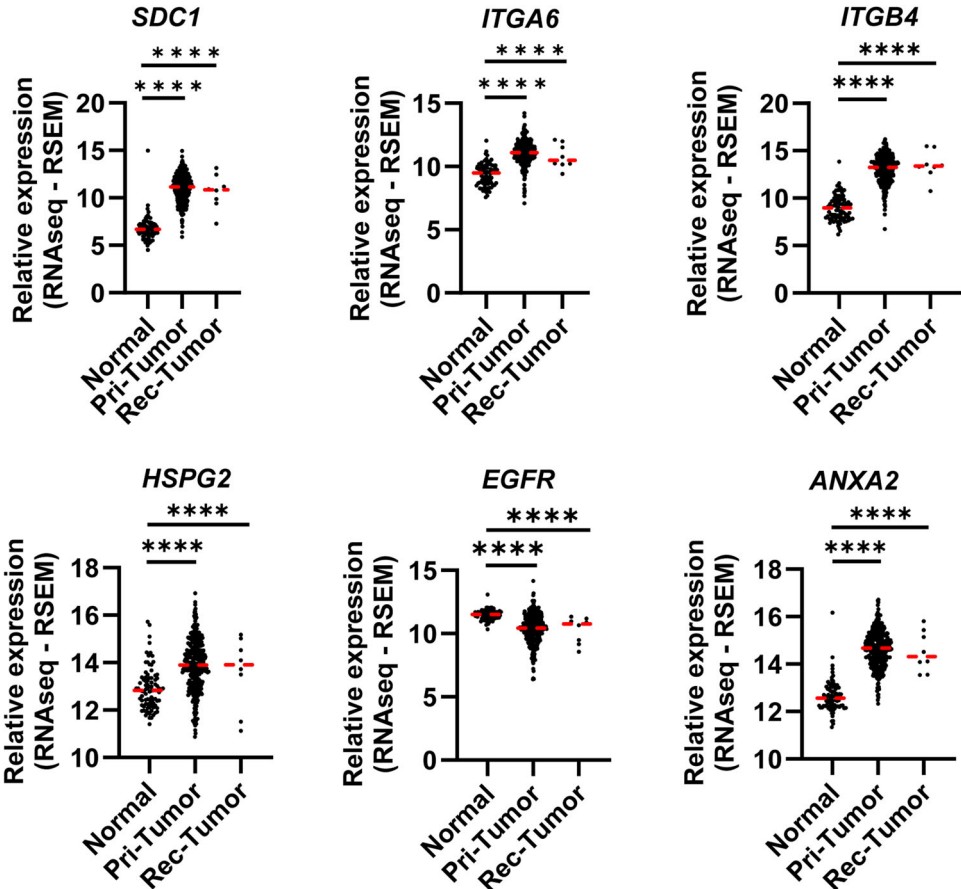

**Figure EV2. Expression of genes encoding the putative HPV receptor molecules in normal ovarian tissues, primary ovarian tumors, and recurrent ovarian tumor tissues.**

The TCGA TARGET GTEx study online tool (https://xenabrowser.net/) was used to compare the expression of genes encoding the putative HPV receptor molecules in normal ovarian tissues ($n = 88$ normal human samples), primary ovarian tumors ($n = 418$ patient samples), and recurrent ovarian tumor tissues ($n = 8$ patient samples). Data were taken from the UCSC RNA-seq Compendium, where TCGA, TARGET, and GTEx samples are re-analyzed using the same RNA-seq pipeline. Extracted data were analyzed for significance using the one-way ANOVA followed by the Tukey's post hoc test. A value of $P < 0.05$ was considered statistically significant. ****$P < 0.0001$, compared to the normal control group (Normal). Exact $P$ values for each gene are presented with the source data of this figure, which is available online. Source data are available online for this figure.

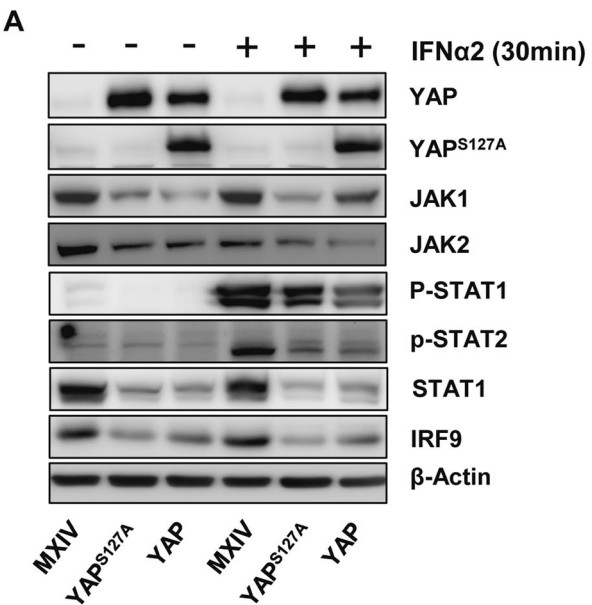

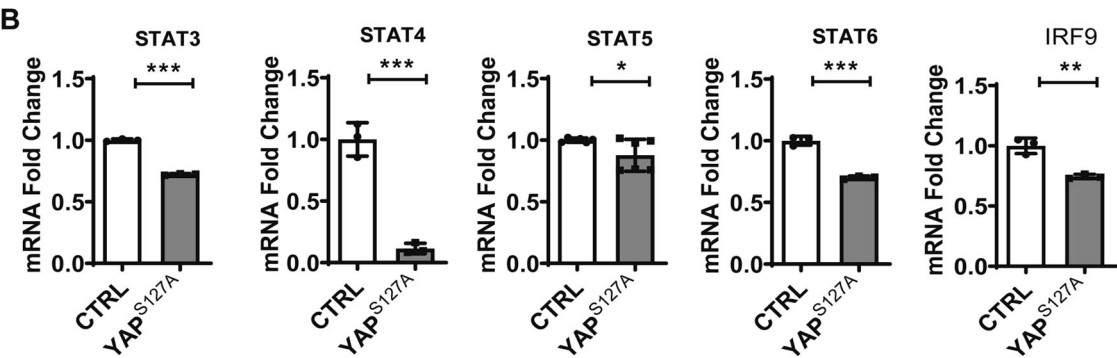

**Figure EV3. Constitutive activation of YAP inhibits the type I interferon (JAK/STAT) signaling pathway in fallopian tube secretory epithelial cells (FNE1).**

(A) Representative blots showing expression and activation of major components of the JAK/STAT/IRF9 pathway in FNE1-MX (control), FNE1-YAP, and FNE1-YAP$^{S127}$ cells with or without IFNα2b treatment for 30 min. Ectopic expression of YAP or YAP$^{S127A}$ in FNE1 cells suppressed IFNα2-induced phosphorylation of STAT1/2. (B) Quantitative data showing that transcription of STATs are suppressed by YAP$^{S127A}$ in FNE1-YAP$^{S127A}$ cells. Each bar represents the mean ± SEM ($n = 3$ technical replicates). Data were analyzed for significance using unpaired t test. A value of $P < 0.05$ was considered statistically significant. *$P < 0.05$, **$P < 0.01$; ***$P < 0.001$, when compared with MX control (CTRL). Exact P values for each gene: $P < 0.0001$ for *STAT3*; $P = 0.0004$ for *STAT4*; $P = 0.0448$ for *STAT5*; $P = 0.0002$ for *STAT6*; $P = 0.0027$ for *IRF9*. Source data are available online for this figure.

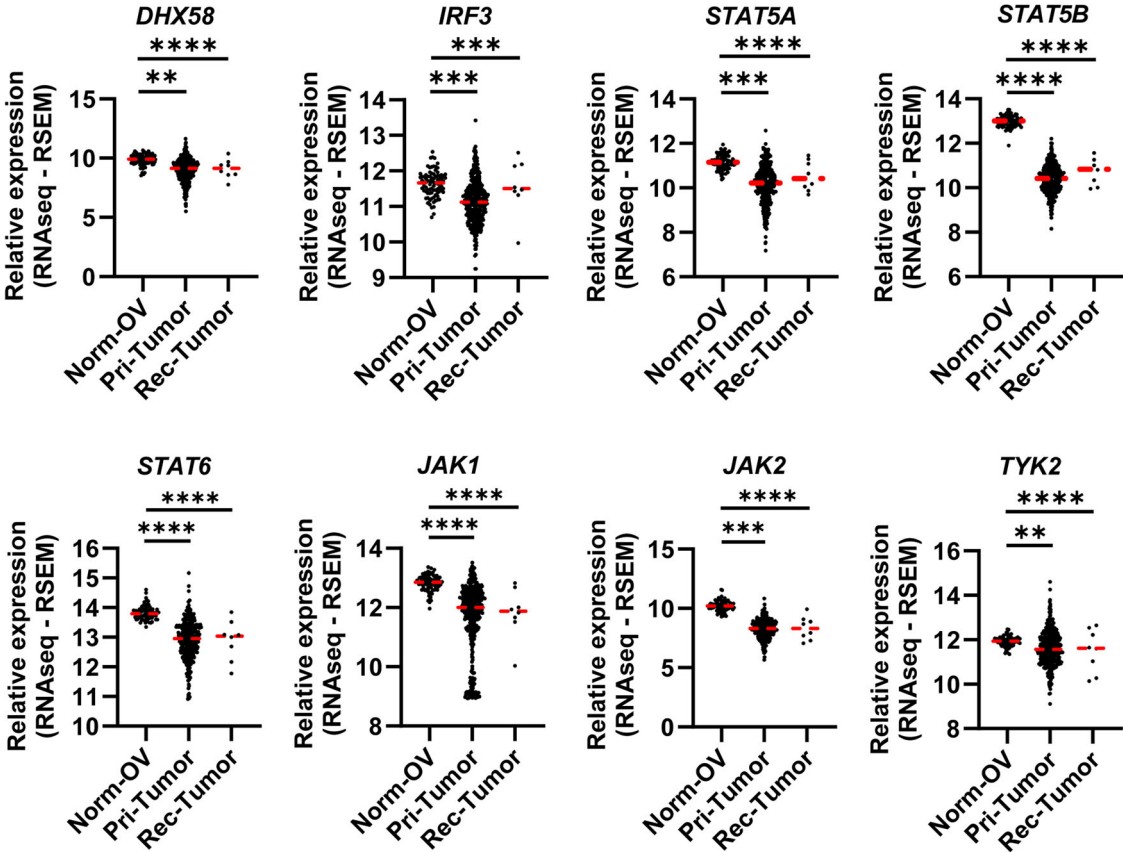

**Figure EV4. Expression of genes encoding the key molecules of the innate immune signaling pathway in normal ovarian tissues, primary ovarian tumors, and recurrent ovarian tumors.**

The TCGA TARGET GTEx study online tool (https://xenabrowser.net/) was used to compare the expression of genes encoding the putative HPV receptor molecules in normal ovarian tissues ($n = 88$ normal ovarian samples), primary ovarian tumor ($n = 418$ patient samples), and recurrent ovarian tumor tissues ($n = 8$ patient samples). Data were from the UCSC RNA-seq Compendium, where TCGA, TARGET, and GTEx samples are re-analyzed using the same RNA-seq pipeline. Extracted data were analyzed for significance using the one-way ANOVA followed by the Tukey's post hoc test. A value of $P < 0.05$ was considered statistically significant. ***$P < 0.001$; ****$P < 0.0001$, when compared to the normal control group (Norm-OV). Exact $P$ values for each gene are presented with the source data of this figure, which is available online. Source data are available online for this figure.

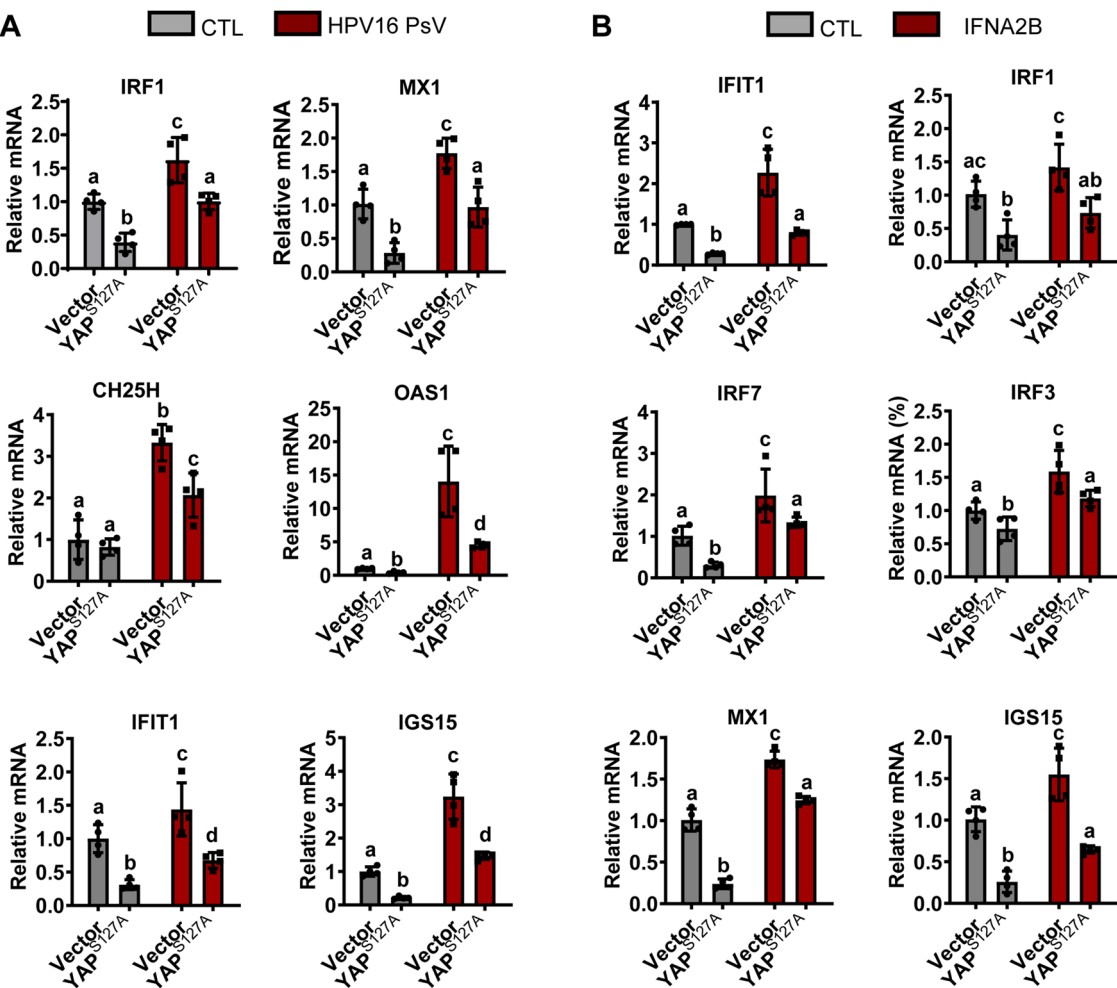

**Figure EV5. Constitutive activation of YAP1 blocks basal, pathogen-induced, or IFNα2b-induced production of antiviral molecules in FTECs.**

(A) Quantitative data showing mRNA levels of several major antiviral interferon-stimulated genes (ISGs) in control (FNE1-MX) and YAP[S127A]-expressing FNE1 (FNE1-YAP[S127A]) cells with or without HPV16 pseudovirions treatment. (B) Quantitative data showing mRNA levels of major components of the JAK/STAT/IRF9 pathway and some antiviral ISGs in FNE1-MX and FNE1-YAP[S127A] cells with or without IFNα2b treatment. Each bar represents the mean + SEM (*n* = 4 technical replicates). Bars with different letters are significantly different from each other. Data were analyzed for significance using the two-way ANOVA followed by the Tukey's multiple comparisons post hoc test. A value of $P < 0.05$ was considered statistically significant. Exact *P* values between the compared groups for each gene are presented with the source data of Fig. EV5A and Fig. EV5B, which are available online. Source data are available online for this figure.

