## [Peer Review File · EMBO Reports]

HPV-YAP1 oncogenic alliance drives malignant transformation of fallopian tube epithelial cells

Cheng Wang, Chunbo He, Xiangmin Lv, Jiyuan Liu, Peichao Chen, Jinpeng Ruan, Cong Huang, Peter Angeletti, Guohua Hua, Madelyn Moness, Davie Shi, Anjali Dhar, Siyi Yang, Savannah Murphy, Isabelle Montoute, Xingcheng Chen, Kazi Islam, sophia george, Tan Ince, Ronny Drapkin, Chittibabu Guda, and John Davis

Corresponding author(s): Cheng Wang (cwang34@mgh.harvard.edu)

Review Timeline:

Submission Date:	6th Dec 23
Editorial Decision:	17th Jan 24
Revision Received:	7th Jun 24
Editorial Decision:	18th Jul 24
Revision Received:	16th Aug 24
Accepted:	20th Aug 24

Editor: Achim Breiling

Transaction Report:

Dear Dr. Wang,

Thank you for the transfer of your research manuscript to EMBO reports. I have now received the reports from the three referees that were asked to evaluate your study, which can be found at the end of this email.

As you will see, the referees think that the findings are of interest. However, they have several comments, concerns, and suggestions, indicating that a major revision of the manuscript is necessary to allow publication of the study in EMBO reports. As the reports are below, and all the referee concerns need to be addressed, I will not detail them here.

Given the constructive referee comments, I would like to invite you to revise your manuscript with the understanding that all referee concerns must be addressed in the revised manuscript or in a detailed point-by-point response. Acceptance of your manuscript will depend on a positive outcome of a second round of review. It is EMBO reports policy to allow a single round of revision only and acceptance of the manuscript will therefore depend on the completeness of your responses included in the next, final version of the manuscript.

- 1) a .docx formatted version of the final manuscript text (including legends for main figures, EV figures and tables), but without the figures included. Figure legends should be compiled at the end of the manuscript text.
- 2) individual production quality figure files as .eps, .tif, .jpg (one file per figure), of main figures (up to 8) and EV figures. Please upload these as separate, individual files upon re-submission.

- 4) a complete author checklist, which you can download from our author guidelines (<https://www.embopress.org/page/journal/14693178/authorguide>). Please insert page numbers in the checklist to indicate where the requested information can be found in the manuscript. The completed author checklist will also be part of the RPF.

- 5) that primary datasets produced in this study (e.g. RNA-seq, ChIP-seq, structural and array data) are deposited in an

appropriate public database. If no primary datasets have been deposited, please also state this in a dedicated section (e.g. 'No primary datasets have been generated and deposited'), see below.

The accession numbers and database should be listed in a formal "Data Availability" section (placed after Materials & Methods) that follows the model below. This is now mandatory (like the COI statement). Please note that the Data Availability Section is restricted to new primary data that are part of this study. This section is mandatory. As indicated above, if no primary datasets have been deposited, please state this in this section

Data availability

8) Regarding data quantification and statistics, please make sure that the number "n" for how many independent experiments were performed, their nature (biological versus technical replicates), the bars and error bars (e.g. SEM, SD) and the test used to calculate p-values is indicated in the respective figure legends (also for potential EV figures and all those in the final Appendix). Please also check that all the p-values are explained in the legend, and that these fit to those shown in the figure. Please provide statistical testing where applicable. Please avoid the phrase 'independent experiment', but clearly state if these were biological or technical replicates. Please also indicate (e.g. with n.s.) if testing was performed, but the differences are not significant. In case n=2, please show the data as separate datapoints without error bars and statistics. See also: <http://www.embopress.org/page/journal/14693178/authorguide#statisticalanalysis>

9) Please add scale bars of similar style and thickness to all the microscopic images, using clearly visible black or white bars (depending on the background). Please place these in the lower right corner of the images themselves. Please do not write on or near the bars in the image but define the size in the respective figure legend.

10) Please also note our reference format:

12) We now use CRediT to specify the contributions of each author in the journal submission system. CRediT replaces the author contribution section. Please use the free text box to provide more detailed descriptions and do not provide your final manuscript text file with an author contributions section. See also our guide to authors: <https://www.embopress.org/page/journal/14693178/authorguide#authorshipguidelines>

13) We would encourage you to use 'Structured Methods', our new Materials and Methods format. According to this format, the

Materials and Methods section should include a Reagents and Tools Table (listing key reagents, experimental models, software and relevant equipment and including their sources and relevant identifiers), uploaded as separate file, followed by a Methods and Protocols section in which we encourage the authors to describe their methods using a step-by-step protocol format with bullet points, to facilitate the adoption of the methodologies across labs. More information on how to adhere to this format as well as downloadable templates (.doc or .xls) for the Reagents and Tools Table can be found in our author guidelines (section 'Structured Methods'):

14) Please order the manuscript sections like this, using these names:

Title page - Abstract - Keywords - Introduction - Results - Discussion - Materials and Methods - Data availability section - Acknowledgements - Disclosure and Competing Interests Statement - References - Figure legends - Expanded View Figure legends

I look forward to seeing a revised version of your manuscript when it is ready. Please let me know if you have questions or comments regarding the revision.

Yours sincerely,

Referee #1:

Authors beautifully introduced that occasional infection by high-risk HPVs in fallopian tube epithelial cells as one of the potential origins of some high grade serous ovarian cancers (HGSOC). Authors provided evidence for the essential role of activated YAP1 in oncogenic transformation of HPV-infected fallopian epithelial cells. These findings are novel about HGSOC and potentially useful for early detection, prevention and developing targeted therapy of this disease. Authors conducted well designed, controlled in vitro experiments to show presence of HPV DNA in patient derived specimens, susceptibility of fallopian tube epithelial cells to HPV infection, role of HPV to prevent replicative senescence, and role of YAP1 in transformation. Authors then conducted some in vivo experiments to support the in vitro data. Most of the experiments are well done with adequate controls. However there are following study gaps, which should be addressed.

Introduction:

(1) Previously published reports of function of HPV oncogenes in replicative senescence, proliferation, neoplastic transformation and relevance of Yap1 in HPV induced cancers should be acknowledged and briefly described to properly set the context of this study.

Result:

(1) Authors conducted convincing experiments to prove that HPV E6E7 induced proliferation in normal FTECs prevented natural or YAP driven replicative senescence. Considering the potential functions of E6 and E7 on p53 and pRB, this was expected. However, it is not clear which of the HPV genes prevented YAP-induced senescence in FTEC and how. How E6E7 elevated proliferation in immortalized cell lines which also expressed SV40 T-antigen and /or hTERT?

(2) Authors then performed standard DNA based detection of high-risk HPV DNA in almost 1/3 rd specimens of fallopian tube carcinoma, immortalized fallopian tube cell lines, chronically inflammatory fallopian tube tissues. However, RNA analysis would be necessary to prove that high risk HPV DNA is transcriptionally active and may be causally involved in these cells and patient specimens.

(3) Authors then performed HPV-infectivity assay to show that immortalized FTECs express cell surface molecules involved in HPV entry and infections at a level higher than normal cells. But infection of Ft246 as shown in Fig 3C, D is not very different than in normal FTEC, hence does not appear to be consistent with the text. Fig 3F experiment is very interesting, but how many viruses were perfused, and how this experiment will relate to retrograde menstruation? How common is retrograde menstruation?

(4) Authors provided convincing evidence that YAP1 regulated expression of ITGa6, SDC1 and EGFR receptor/co-receptors for HPV entry in FTEC (Fig 3 and 4). But why FNE1 was used for some mechanistic studies on the role of Yap1, instead of Ft194 or Ft246 described in earlier experiments? How did Yap1 regulate these cell surface receptors? In cervical cancers E6 was shown to stabilize/activate YAP1 (author's previous research, reference 12). Then why HPV16 E6/E7 did not elevate YAP1 in FTEC cells?

(5) Authors performed several transformation assays. But, I did not understand why the subtitle suggested use of primary fallopian tube cells, but text and Fig 5 used immortalized lines (FT190, and FNE1). What is the rationale of not using FT194 or FT246, which provided majority of in vitro data above? Text did not mention mice gender. Authors did not explain why there are

so much of p53 in E6E7 expressing cells (Fig 5P, Q). HPV16 E6 should have degraded it.

(6) The comparative GSEA of expressed genes is comprehensive and important contribution. It is interesting to note that combination of YAP and HPV16E6E7 drastically reduced innate immune pathways, compared to YAP1 alone. Authors may explain this phenotype.

Discussion:

The discussion is little too long. Authors discussed very well describing potential influence of HPV in HGSOc/fallopian tube cancers and critical role of YAP1 activation in HPV E6E7 mediated transformation. However, the discussion has significant redundancy and over speculations, especially regarding hit and run mechanism of oncogenic transformation in cervical cancers without proper evidence. The overwhelming consensus is that high risk HPV E6 and E7 expression is necessary for survival of cervical cancer cells. In 95% of cervical cancers hr HPV is detected either episomal or integrated and continued expression of E6/E7 is necessary for survival of cervical cancer cell lines. Authors did not present any evidence regarding role of HPV oncogenes in viability of HPV+ HGSOc/FTEC immortalized/cancer cells and effect of HPV oncogenes in HPV+ immortalized FTEC or FTEC cancers. Absence of evidence for HPV RNA in patient samples fails to convince the causative role of HPV in these HGSOc. Further, authors discussed possibility but presented no evidence for (a) infectivity by pseudo virus is same as wild type HPV in primary cells, and (b) HPV reinfection of cervical cancer actually happens in patients.

Referee #2:

He et al. describe in their manuscript the interaction of HPV E6/E7 and YAP1 and raise the hypothesis that this alliance can potentially contribute to the malignant transformation of fallopian tube cells resulting in HGSOc development. This work builds on earlier work of the authors regarding the Hippo-YAP pathway and HPV in cervical carcinogenesis and the YAP1 function in ovarian cancer. The manuscript is of high quality, used methods enable the analysis of the hypothesis in-vitro and partially in-vivo and the results support the interaction of HPV E6/E7 and YAP1. HPV E6/E7 can prevent the natural or YAP1-induced senescence, thus promote YAP1-induced malignant transformation and stimulate proliferation and metastasis. Moreover, YAP1 increases the HPV pseudovirion infectivity by upregulating "HPV receptor genes" and inhibits the expression of genes involved in the innate immune reaction. However, these data replicate findings from the research field of the Hippo-Yap1 pathway and the interaction of HPV and YAP1 in cervical or oropharyngeal cancer in another epithelial cell type and have limited novelty. The most important and novel point of the work would be a proof of the contribution of HPV to the development of tubal lesions and ovarian cancer but neither the presented experimental data nor the discussion strongly support this hypothesis.

Major remarks:

1. The authors used well-known PCR systems (MY09/11; GP5+/6+) for the detection of HPV-DNA in cell lines and clinical samples. However, these low stringency systems enabling a broad detection of different HPV types cannot be evaluated by pure agarose gel electrophoresis but need a specific detection (southern blot, sequencing, probe hybridization etc.) to verify the presence of HPV sequences. This requirement is already proven in the presented data. Only 2/11 MY09/11-positive samples could be validated by the HC2 test and many of the PCRs from the analyzed cell lines show a band at the expected size of the PCR product (Fig. S1).

In addition, a HPV DNA detection by PCR is not a strong proof of the presence of infected cells because such highly sensitive methods are prone to contaminations at any step from surgery to PCR setup and do not proof the presence of HPV in cells. Additional experiments are required to validate the detection of infected cells. These may consist of: (i) the detection of HPV transcripts and (ii) the in-situ detection of HPV-DNA by hybridization. It would also be important to exclude the presence of HPV induced tumors in the analyzed patients as several publications describe the metastasis of tumor cells to the fallopian tube mimicking STICs.

2. The discussion of the proposed hypothesis, that HPV can contribute to HGSOc must be improved and should include data pointing against this idea. (i) It is known that HPV can infect a variety of epithelial cells, but the replication is strongly dependent on the cell type and epithelial differentiation. (ii) The proposed "hit-and-run" mechanism is described for cutaneous beta-HPV types increasing the risk for squamous cell skin cancer after UV-exposure but not for genital alpha-HPV types. The overwhelming majority of cervical cancer contains HPV and strongly depends on the continuous expression of E6/E7. (iii) The majority of HGSOc precursor lesions (p53-signature and STIC) exhibit an overexpression of p53 that is unlikely to occur under expression of E6. The protective effect of tubal ligation seems to be subtype specific (endometrioid cases; Duus AH et al, GynOnc 2023) and thus may originate from the disruption of a retrograde menstruation.

3. The authors use a HPV pseudovirion infection assay for different cell lines/cultures and compare them directly (Fig. 3). Can they exclude an effect of different cell culture medium? How many cells were counted per experiment? Can they exemplarily confirm the quantification by flow cytometry?

4. The authors should improve the description of their GSEA analyses. It seems that only one of the analyses resulted in a significant enrichment albeit they used the fgsea algorithm.

Minor remarks:

1. Page 5: "Unexpectedly, we found that ectopic expression of YAP or YAP1S127A promoted cellular senescence in the primary FTECs, which is indicated by the significant increase in the percentage of cells that underwent cell cycle arrest, cellular hypertrophy, and expression of senescence-associated β -galactosidase (SA- β -gal) (Fig. 1A & B)." Data for cell cycle arrest, and cellular hypertrophy should be added.
2. Page 6: "(consensus primers PGMY09/MY11)" My09/11 primers are degenerated, the GP primers are consensus. Please correct in the material section
3. Page 9: "Here, we observed that fallopian tube and ovarian HGSC cells, which have higher expression of YAP1, also have higher expression of HPV receptors and increased susceptibility to HPV infection (Fig. 3)." Figure 3 shows this correlation only for CxCa and HGSC cells.
4. The authors include mouse experiments (Fig. 5). An graphical overview or KM-plot for tumor development would be an advantage. In Figure 5P, Q they show a strong p53 expression. Do the cells still express HPV E6/E7?
5. Page 16: "Generally, HPV preferentially infects the cervical epithelial cells and cancerous fallopian tube epithelial cells. This is consistent with previous reports that HPV infection of epithelial cells requires cell cycle progression, and that cancer cells (usually highly proliferative) are more susceptible to HPV infection [13, 41, 42]." The authors should differentiate between their infection assay and literature data for stable replication of HPV.

Referee #3:

In the manuscript HPV-YAP1 oncogenic alliance drives malignant transformation of fallopian tube epithelial cells , authors investigated the role of high-risk HPV infection together with hyperactivation of YAP1 in the onset of ovarian carcinoma, through the infection of fallopian tube epithelium. YAP1 activation fosters HPV infection by promoting cell intake of HPV virions and suppressing innate immunity signals.

Authors provide many in vivo and in vitro results to validate their hypotheses and data are clearly explained. From a mechanistic point of view, authors clearly explained the role of YAP1 in facilitating HPV infection and their coordinated activity in promoting tumour formation and metastasis.

Software quantifications need to be added to the immunofluorescence images in figure 3 and 4.

I have concerns regarding the real impact of HPV infection on the onset of ovarian carcinoma. In particular, the origin of the samples examined in figure 2B and C is not clear. Were they the same samples used in figure 2A? In that case, why results are discrepant? Were they all fallopian tube cancer samples?

Moreover, it would be interesting to investigate whether those two HPV high-risk positive fallopian tube cancer samples (CP3 and CP7) present higher expression of YAP1 and HPV related receptors, compared to non-infected samples, as well as the downregulation of innate immunity signalling.

Dear Dr. Breiling,

Thank you very much for the opportunity to revise our manuscript. We also appreciate the reviewers for their insightful and constructive suggestions. In the past several months, we performed new experiments to address reviewers' concerns point-by-point. This revision not only addressed concerns raised by the reviewers but also improved the quality of our manuscript significantly. Importantly, research results derived from these new experiments further support the role of the YAP1-HPV oncogenic alliance in HGSOE development and provide new evidence for the proposed "hit and run" mechanisms by which HPV interacts with YAP1 oncogene to drive the initiation of HGSOE from fallopian tube epithelial cells.

Referee #1:

Introduction:

(1) Previously published reports of function of HPV oncogenes in replicative senescence, proliferation, neoplastic transformation and relevance of Yap1 in HPV induced cancers should be acknowledged and briefly described to properly set the context of this study.

Response: We described and cited several previous findings in the introduction in the revised manuscript. To better formulate our manuscript, we described and cited additional previous findings associated with YAP and HPV in the discussion.

Result:

(1) Authors conducted convincing experiments to prove that HPV E6/E7 induced proliferation in normal FTECs prevented natural or YAP driven replicative senescence. Considering the potential functions of E6 and E7 on p53 and pRB, this was expected. However, it is not clear which of the HPV genes prevented YAP-induced senescence in FTEC and how. How E6/E7 elevated proliferation in immortalized cell lines which also expressed SV40 T-antigen and /or hTERT?

Response: It is well-known that high-risk HPV E6 and E7 proteins target TP53 and RB1 proteins, respectively, to induce cell proliferation and transformation [PMID: 2556261; PMID: 1657399; PMID: 2175676; PMID: 1316611]. Nevertheless, high-risk HPV E6 was reported to drive cell proliferation via TP53-independent pathways [reviewed by PMID: 29890655]. Consistently, our recent data demonstrated that the YAP1 oncogene induces senescence via a YAP1-LATS2 negative feedback loop in YAP1-hyperactivated ovarian and cervical epithelial cells [PMID: 35761037]. The role of HPV E6/E7 in ovarian and fallopian tube cancer development has not been confirmed, mainly because HPV DNA is not frequently detected in fallopian tube/ovarian tumor tissues. The evidence provided in this manuscript suggested that: 1) high-risk HPVs are able to infect fallopian tube epithelial cells (the cell-of-origin of the majority of HGSOE); 2) high-risk HPV E6/E7 aid YAP1 to induce overgrowth and transformation of FTECs by overriding YAP1 oncogene-induced senescence; 3) hyperactivated YAP1 facilitate HPV infection (and tumorigenesis) by suppressing innate immunity; These observation indicated that high-risk HPV E6/E7 oncoproteins synergize with the hyperactivated YAP1 to induce HGSC development. Ongoing studies in our lab focus on testing our hypothesis that high-risk HPVs play a role in HGSC initiation via a "hit & run" mechanism. Since E6 and E7 are always co-expressed, they may function as a group to exert tumorigenic function in different cells. It seems that E6 and E7 work jointly to prevent YAP1-induced senescence by targeting TP53 and RB1, modulating telomere, and disrupting YAP1-LATS2 feedback loop.

The exact mechanism by which E6/E7 elevated the proliferation of cell lines expressing SV40 or hTERT is still unclear. Although previous studies demonstrated that both high-risk HPVs and SV40 target TP53 and RB1 to induce transformation of infected cells, accumulating evidence demonstrated that both viruses employ TP53- and RB1-independent pathways to induce cell transformation. For example, high-risk HPV E6 can function by interacting with many other proteins (e.g., TP73, BRCA1-associated RING domain 1, etc.) to achieve oncogenic action [reviewed by: PMID: 29890655; PMID: 17678435]. Our

previous studies demonstrated that in cervical cancer cells, HPV16 E6 stabilized YAP1 oncoprotein, leading to hyperactivation of YAP1 and overgrowth of cervical epithelial cells [PMID: 26417066]. Our recent data also suggest that HPV E6/E7 could disrupt the YAP1-LATS2 negative feedback loop to facilitate ovarian and cervical epithelial cell transformation [PMID: 30755404; PMID: 35761037, Fig. R1A]. Similarly, previous studies demonstrated that SV40 large T antigen binds transcriptional co-activators such as p300 and CBP to drive cell transformation [PMID: 11243895]. In the FNE1 cells (immortalized by hTERT overexpression), our recent preliminary studies demonstrated that HPV16 E6/E7 could upregulate genes encoding histones, which plays a central role in transcription regulation, DNA repair, DNA replication, and chromosomal stability [Fig. R1B], and activate signaling pathways associated with cell cycle progression (Fig. R1C). It is clear that HPV E6/E7 could employ many different signaling pathways (not only TP53, RB1, and TERT) to drive cell proliferation. More studies are guaranteed to put in-depth insight into the potential synergistic effect of high-risk HPV and SV40 on cell proliferation and transformation.

Figure R1B with unpublished data and its description has been removed upon request by the authors.

Figure R1. Evidence for HPV16 E6/E7 function in Fallopian tube epithelial cells expressing SV40 T-antigen and/or human TERT. A) Representative blots showing that the presence of HPV16 E6/E7 protein blocked YAP1-induced upregulation of LATS2 in cultured FT190 cells. **B)** [REDACTED: Figure R1B's description.]. **C)** GSEA analysis showing that the presence of HPV16 E6/E7 induced enrichment of genes promoting cell cycle progression.

(2) Authors then performed standard DNA based detection of high-risk HPV DNA in almost 1/3rd specimens of fallopian tube carcinoma, immortalized fallopian tube cell lines, chronically inflammatory fallopian tube tissues. However, RNA analysis would be necessary to prove that high risk HPV DNA is transcriptionally active and may be causally involved in these cells and patient specimens.

Response: We did not include E6/E7 transcript data because we observed a discrepancy between HPV DNA positivity and HPV E6/E7 expression. Using RNA scope, we detected HPV16 E6 expression in only one case of eleven detected human tumor samples (Fig. R2). Our HPV DNA-positive samples were frequently found to be high-risk HPV E6/E7 RNA negative. This observation is explained by several reports showing that a portion of HPV DNA in the HPV DNA-positive cancer tissues is transcriptionally inactive [PMID: 28384274; PMID: 28077784]. Interestingly, a more recent study assessed the

Figure R2. Expression of HPV16 E6/E7 in the Fallopian tube STIC lesion (precursor of HGSOc) detected by RNA scope. **A)** Representative image showing the expression of HPV16 E6/E7 mRNA (in pink) in fallopian tube STIC lesion (arrow) of a human patient (sample-GU980150-E10). HPV16 E6/E7 were detected and visualized using the RNA scope kit from ADC (. **B)** Representative image showing the expression of YAP1 protein (in brown) in fallopian tube STIC lesion (arrow). YAP1 protein was detected using immunohistochemistry (arrow). **C)** Representative image showing the expression of HPV16 E6/E7 mRNA (in pink) in SiHa cell xenograft tumor tissues (positive control). **D)** Representative images showing negative staining (non-targeting probe) of HPV16 E6/E7 mRNA in SiHa cell xenograft tumor tissues(arrow). A representative image in insert is a negatively stained STIC tissue. Scale bar: 50µm.

consequences of HPV genomic integration using multi-omics and single-cell data in 106 cervical cancer patients and demonstrated that only ~20% of the HPV integrated breakpoints are productive integration (integrated HPV DNA is transcriptionally active), suggesting that even in the cervical cancer, the majority of integrated HPV DNA are silent integration (non-transcriptional integration) [PMID:36777180]. In

addition, the transcription-independent effects of HPV integration, such as altering the 3D genome architecture or transcriptional regulatory element activity around integration sites, can also exert tumorigenic activity. The presence of HPV DNA in the fallopian tube and tumor samples suggested that these samples had been infected by HPV. Our in vitro data demonstrated that if high-risk HPVs infect FTECs with hyperactivation of YAP1, HPV can facilitate the transformation of these cells by enabling these cells to overcome hyperactivated YAP1-induced senescence. We also found that hyperactivated YAP1 can suppress innate immunity to help the transformed cells evade immune surveillance. Therefore, the discrepancy between HPV DNA and E6/E7 expression doesn't exclude the possibility that HPV may be involved in the tumor initiation in at least a portion of HGSOc.

(3) Authors then performed HPV-infectivity assay to show that immortalized FTECs express cell surface molecules involved in HPV entry and infections at a level higher than normal cells. But infection of Ft246 as shown in Fig 3C, D is not very different than in normal FTEC, hence does not appear to be consistent with the text. Fig 3F experiment is very interesting, but how many viruses were perfused, and how this experiment will relate to retrograde menstruation? How common is retrograde menstruation?

Response: Our previous studies indicated that HPV infectivity is associated with cell proliferation rate. Compared to the cultured primary cells (e.g., FTECs), immortalized cells and cancer cell lines generally have higher proliferation rates and, thereby, a higher HPV infectivity. Among all immortalized cell lines (FT237, FT246, FT190, FT194, etc.), FT246 has the lowest growth rate. Consistently, we found that the PsV susceptibility of FT246 cells is lower (equivalent to the primary FTEC) than that of other immortalized cell lines and cancer cells. To confirm the impact of YAP1 on HPV infectivity, we repeated our study with FT246 cells and quantified the PsV-positive cells with FCM. We use a relatively lower MOI to ensure the specificity of PsV infection in this confirmative study. The results showed that hyperactivation of YAP1 in fallopian tube epithelial cells increased HPV infectivity, which is indicated by the significantly increased PsV-positive cells in the FT246-YAP^{S127A} group when compared to the FT246-MX group (Fig. R3A & 3B).

Figure R3. Hyperactivation of YAP1 increases the infectivity of HPV pseudovirus (PsV) in fallopian tube epithelial cells. A) representative images showing the PsV-positive cells in FT246-MX and FT246-YAP^{S127A} cells incubated in the presence or absence of HPV16 pseudovirus (PsV). HEK293-T cells were used as positive control. Scale bar: 20 μ m. B) Flow cytometric quantification of PsV-positive cells post PsV incubation. FT246-MX and FT246-YAP^{S127A} cells were incubated with PsV (MOI= 0.5) for 7 days before fluorescent microscopy or FCM quantification.

The concentration of HPV16 Pseudovirus (PsV) was 1.0×10^6 pfu/ μ l for *in vivo* infection studies. A total of 30 μ l of PsV (in saline with Evans blue) were injected into the upper uterine tube (near the oviduct) with a protocol established by Dr. John Schiller [Nat. Med. 2007, PMID: 17603495].

Previous reports demonstrated that retrograde menstruation is found in over 90% of menstruating patients during gynecological surgery [PMID: 6234483, PMID: 17636274]. Earlier studies also indicated that the abnormal endometrium derived from retrograde menstruation is the foremost predisposing factor for endometriosis [PMID: 17636274; PMID: 15541453]. Interestingly, previous reports have already demonstrated a significantly higher rate of high-risk HPV positivity among patients with endometriosis via ascending from the lower genital tract to the pelvis and upper genital tract [PMID: 27928852, PMID: 30940433, PMID: 19200955]. These observations strongly support our hypothesis that HPV could reach the fallopian tube epithelium by retrograde menstruation.

(4) Authors provided convincing evidence that YAP1 regulated expression of ITGA6, SDC1 and EGFR receptor/co-receptors for HPV entry in FTEC (Fig 3 and 4). But why FNE1 was used for some mechanistic studies on the role of Yap1, instead of Ft194 or Ft246 described in earlier experiments? How did Yap1 regulate these cell surface receptors? In cervical cancers E6 was shown to stabilize/activate YAP1 (author's previous research, reference 12). Then why HPV16 E6/E7 did not elevate YAP1 in FTEC cells?

Response: FTECs are primary fallopian epithelial cells and can only be cultured in a traditional culture system for a limited number of passages before senescence. For the mechanistic study, we need a large number of fast-growing cells with a relatively “clean” genetic background. FNE1 cells are fallopian tube epithelial cells immortalized by expressing human TERT. These cells grow significantly faster than primary FTECs and can provide sufficient cells for our mechanistic studies. We avoid using FT246, FT190, and FT194 cells for mechanistic studies because FT246 was immortalized by human TERT overexpression, shRNA-mediated TP53 inactivation, and CDK4 mutation (R24C). We also avoid FT190 and 194 cells because these cells express SV40 T antigen.

Yes, YAP1 also upregulated the expression of HPV cell surface receptor molecules, including *ITGA6* ($p < 0.0001$), *SDC1* ($p < 0.0001$), and *HSPG2* ($p < 0.0001$), and *ANXA2* ($p < 0.0001$) in FNE1 cells (Fig.R4).

Figure R4. Effect of hyperactivated YAP1 on the expression of the putative HPV receptor molecules in FNE1 cells. FNE1-MX and FNE1-YAP^{S127A} cells were incubated in the growth medium for 72 hours, and the expression of YAP1, SDC1, ITGA6, ITGB4, HSPG2, EGFR, LAMA3, and AnxA2 were examined by Next Generation Sequencing technology. **: P < 0.01 when compared to the control group (CTRL); ****: P < 0.0001 when compared to the control group (CTRL).

Our data indicated that, consistent with previous reports, HPV16 E6/E7 also upregulates YAP1 expression in FTECs. For example, in FNE1 cells, which were immortalized by hTERT (without SV40), HPV16 E6/E7 significantly upregulate transcription of *YAP1* ($p=0.00335$), and *YAP1* target genes such as *CTGF* ($p < 0.0001$), *CYR61* ($p < 0.0001$), *ANKRD1* ($p < 0.01$), *BIRC5* ($p < 0.01$), and *KRT7* ($p < 0.0001$), (Fig. R5).

Figure R5. Effect of HPV16 E6/E7 on the expression of YAP1 and YAP1 downstream genes in FNE1 cells. FNE1-MX and FNE1-YAP^{S127A} cells were incubated in the growth medium for 72 hours, and the expression of YAP1, and its downstream genes such as *CTGF*, *CYR61*, *ANKRD1*, *BIRC5*, and *KRT7* were examined by Next Generation Sequencing technology. **: $P < 0.01$ when compared to the control group (CTRL); ****: $P < 0.0001$ when compared to the control group (CTRL).

(5) Authors performed several transformation assays. But, I did not understand why the subtitle suggested use of primary fallopian tube cells, but text and Fig 5 used immortalized lines (FT190, and FNE1). What is the rationale of not using FT194 or FT246, which provided majority of in vitro data above? Text did not mention mice gender. Authors did not explain why there are so much of p53 in E6E7 expressing cells (Fig 5P, Q). HPV16 E6 should have degraded it.

Response: We apologize for the confusing information in the figure legend. The "FTECs" in the subtitle has been replaced with "immortalized FTECs" in the revised version.

We generally use different cell lines in our studies to make sure that our findings are not limited to a specific cell line. However, we tried to avoid FT190 and 194 cells in mechanistic and functional studies because they have ectopic expression of hTERT and SV40 TAg. We also avoid using FT246 cells in mechanistic and functional studies because these cells have multiple genetic modifications (TERT overexpression, TP53 inactivation, and CDK4 constitutively active mutation). FNE1 cells were human fallopian tube epithelial cells immortalized by expressing TERT and their genetic background is relatively clean. Importantly, these cells are not tumorigenic, providing a perfect cellular model for examining the oncogenic ability of the YAP1-HPV oncogenic alliance. Therefore, the most mechanistic and functional studies were designed based on the FNE1 model.

Our study focuses on the fallopian tube/ovarian cancer, and therefore, we only use female mice. The mouse gender is highlighted in the Materials and Methods.

Accumulation of TP53 in the tumor cells derived from FNE1-E6/E7-YAP^{S127A} cells is unexpected because these cells have HPV16 E6 protein. However, the presence of TP53 in the nuclei of tumor cells derived from FNE1-E6/E7-YAP^{S127A} cells suggests that these cells are appropriate for modeling HGSOc development since the accumulation of TP53 in the nucleus is the most critical molecular signature of HGSOc. Conceivably, the fast-growing FNE1-E6/E7-YAP^{S127A} cells result in TP53 gene mutation, and FNE1-E6/E7-YAP^{S127A} cells with mutated TP53 are selected and become dominant cells in the tumor tissues during tumor evolution.

Although how TP53 escapes from E6/E6AP-mediated ubiquitination and degradation in HGSOC is still unclear, the mutation-associated TP53 nuclear translocation may contribute significantly to the observed phenomenon. Indeed, previous studies have shown p53^{R175H} inhibited E6-mediated p53 degradation [PMID:31749782]. Recent studies demonstrated that mutant misfolded p53 proteins can aggregate with each other (and also with wild-type P53) to form prion-like structures that improperly accumulate in the nucleus (and cytoplasm) to exert tumor-promoting (gain-of-function) activities in many types of cancers, including ovarian cancer [PMID: 27549118; PMID: 26748848]. The mutation-associated conformation change in TP53 protein may also contribute to the observed evasion of TP53 from E6/E6AP-mediated ubiquitination and degradation. A new project is necessary to completely uncover the cellular and molecular mechanism(s) by which HPV16 interacts with YAP1 oncogene to modulate TP53 activity (e.g., gain-of-function mutation) during HGSOC development.

(6) The comparative GSEA of expressed genes is comprehensive and important contribution. It is interesting to note that combination of YAP and HPV16E6E7 drastically reduced innate immune pathways, compared to YAP1 alone. Authors may explain this phenotype.

Response: Previous reports have shown that high-risk HPV E6/E7 oncoproteins empower high-risk HPV to evade host anti-viral immunity by dysregulating gene expression, protein-protein interactions, posttranslational modifications, and cellular trafficking of critical immune modulators in host cells [PMID: 34628358; PMID: 27890631]. The results in the present study demonstrate that hyperactivation of YAP1 in the fallopian tube epithelial cells results in drastic suppression of innate immunity. A combination of HPV oncoproteins and hyperactivated YAP1 might generate a synergistic immunosuppressing effect and create an immunosuppressive microenvironment in the fallopian tube epithelium that is critical for transformed epithelial cells to overcome host immune surveillance and anti-tumor immune response.

Discussion:

The discussion is little too long. Authors discussed very well describing potential influence of HPV in HGSOC/fallopian tube cancers and critical role of YAP1 activation in HPV E6E7 mediated transformation. However, the discussion has significant redundancy and over speculations, especially regarding "hit and run" mechanism of oncogenic transformation in cervical cancers without proper evidence.

Response: We thank the reviewer for the constructive comments. We agree with the reviewer that the discussion is over-speculative. Our preliminary studies indicate the existence of a "hit and run" mechanism, but it may take years to confirm this hypothesis completely. Following the reviewer's suggestion, we revised our discussion and deleted redundant speculations.

The overwhelming consensus is that high risk HPV E6 and E7 expression is necessary for survival of cervical cancer cells. In 95% of cervical cancers hr-HPV is detected either episomal or integrated and continued expression of E6/E7 is necessary for survival of cervical cancer cell lines. Authors did not present any evidence regarding role of HPV oncogenes in viability of HPV+ HGSOC/FTEC immortalized/cancer cells and effect of HPV oncogenes in HPV+ immortalized FTEC or FTEC cancers.

Response: As mentioned by the reviewer, previous studies have shown that a large portion of cervical cancer cells are addicted to the HPV E6/E7 oncogene during cancer progression. However, HPV-negative cervical cancer and cervical cancers with silent fossil HPV DNA have been frequently reported [PMID: 11451553; PMID: 26175888; PMID: 20473886; PMID: 12556961; PMID: 22323075; PMID: 24477171; PMID: 18628412; PMID: 29179875]. Interestingly, it seems that wild-type TP53 plays a critical role in cervical cancer cell E6/E7 addicts. Loss of p53 predisposes HPV16 transformed cells to lose dependence on the continuous expression of HPV E6/E7 oncogenes for proliferation [PMID: 33321328]. TP53 gene is mutated in almost all high-grade serous ovarian carcinoma (primarily originating from

fallopian tube epithelial cells). We thank the reviewer for this constructive suggestion and plan to design more experiments to study the relationship between gain-of-function TP53 mutation and potential loss of E6/E7 addiction in FTECs transformed by the YAP1-HPV oncogenic alliance during HGSOc initiation and evolution/progression in the following studies. We may uncover a previously unprecedented mechanism by which HPV positive cells lose addiction to HPV oncoproteins.

Absence of evidence for HPV RNA in patient samples fails to convince the causative role of HPV in these HGSOc.

Response: Using the RNAscope technique, we indeed observed that a small fragment of HGSOc STIC lesion has expression of HPV E6/E7 mRNA (Figure R2). However, our findings in this manuscript indicate that HPV may facilitate FTECs with hyperactivated YAP1 to override the oncogene-induced senescence during the initiation of a portion of HGSOc. The mechanisms by which HGSOc loses the HPV genome during cancer progression and evolution need further investigation. Interestingly, recent studies indicated that a portion of cervical cancer also silenced their HPV DNA during cancer evolution [PMID: 11451553; PMID: 26175888; PMID: 20473886; PMID: 12556961; PMID: 22323075; PMID: 24477171; PMID: 18628412; PMID: 29179875].

Further, authors discussed possibility but presented no evidence for (a) infectivity by pseudovirus is same as wild type HPV in primary cells, and (b) HPV reinfection of cervical cancer actually happens in patients.

Response: Several recent studies can answer two questions mentioned by the reviewer.

First, more than 200 genetically distinct HPV subtypes have been identified. Recent studies have shown that multiple HPV infections in the cervix are common (PMID: 37577444). Importantly, studies found that multiple infections, mainly HPV16, are closely interrelated to cervical carcinoma (PMID: PMID: 37577444; PMID: 32429930). Recent longitudinal studies strongly supported the idea that hosts could be infected repeatedly [PMID: 26112742]. Interestingly, recent studies have further demonstrated that infection with one HPV type strongly increases the risk of infection with that type for years afterward. For HPV16, the type responsible for most HPV-related cancers, an initial infection increases the one-year probability of reinfection by 20-fold, and the likelihood of reinfection remains 14-fold higher two years later [PMID: 29208707].

The pseudovirus (PsV) was from the self-assembly of HPV L1 and L2 capsid proteins. The conformational structure of the surface proteins of PsV is similar to that of the native capsid. The similarities in the surface protein structure allow PsV to remain effective in their interaction with ECM, apical cellular proteins, and receptors leading to their internalization [PMID: 27190090; PMID: 32968082]. Therefore, PsV is among the most valuable and reliable tools for studying HPV infection and biology.

Referee #2:

He et al. describe in their manuscript the interaction of HPV E6/E7 and YAP1 and raise the hypothesis that this alliance can potentially contribute to the malignant transformation of fallopian tube cells resulting in HGSOc development. This work builds on earlier work of the authors regarding the Hippo-YAP pathway and HPV in cervical carcinogenesis and the YAP1 function in ovarian cancer. The manuscript is of high quality, used methods enable the analysis of the hypothesis in-vitro and partially in-vivo and the results support the interaction of HPV E6/E7 and YAP1. HPV E6/E7 can prevent the natural or YAP1-induced senescence, thus promotes YAP1-induced malignant transformation and stimulate proliferation and metastasis. Moreover, YAP1 increases the HPV pseudovirion infectivity by upregulating "HPV receptor genes" and inhibits the expression of genes involved in the innate immune reaction. However, these data replicate findings from the research field of the Hippo-Yap1 pathway and the interaction of HPV and YAP1 in cervical or oropharyngeal cancer in another epithelial cell type and

have limited novelty. The most important and novel point of the work would be a proof of the contribution of HPV to the development of tubal lesions and ovarian cancer but neither the presented experimental data nor the discussion strongly support this hypothesis.

Response: We thank the reviewer for pointing out the importance of this study. The importance of HPV infection in the development of cervical and oropharyngeal cancers has been confirmed. However, the role of HPV in ovarian cancer development is still under debate. This could be attributed to the fact that HPV DNA is not frequently detected in ovarian cancer tissues. We hypothesize that HPV may contribute to HGSC development in a "hit and run" mechanism. As an initial step to test this hypothesis, results presented in this study suggest that hyperactivated YAP1 and HPV formed a YAP1-HPV oncogenic alliance to drive the malignant transformation of Fallopian tube epithelial cells. We also provide preliminary cellular and molecular mechanisms by which the YAP1-HPV oncogenic alliance contributes to HGSC development. These findings provide the first clue for the role of the YAP1-HPV oncogenic alliance in HGSC development and suggest that the HPV vaccine may protect women from at least a portion of HGSOC. We agree with the reviewer that more investigations are needed to uncover the cellular and molecular mechanisms by which the YAP1-HPV alliance initiates HGSOC from ovarian and fallopian tube epithelial cells.

Major remarks:

1. The authors used well-known PCR systems (MY09/11; GP5+/6+) for the detection of HPV-DNA in cell lines and clinical samples. However, these low stringency systems enabling a broad detection of different HPV types cannot be evaluated by pure agarose gel electrophoresis but need a specific detection (southern blot, sequencing, probe hybridization etc.) to verify the presence of HPV sequences. This requirement is already proven in the presented data. Only 2/11 MY09/11-positive samples could be validated by the HC2 test and many of the PCRs from the analyzed cell lines show a band at the expected size of the PCR product (Fig. S1). In addition, an HPV DNA detection by PCR is not a strong proof of the presence of infected cells because such highly sensitive methods are prone to contaminations at any step from surgery to PCR setup and do not proof the presence of HPV in cells. Additional experiments are required to validate the detection of infected cells. These may consist of: (i) the detection of HPV transcripts and (ii) the in-situ detection of HPV-DNA by hybridization.

Response: We agree with the reviewer that the MY09/11;GP5+/6+ PCR system may detect many different HPV types, which is consistent with data presented in Figure 2A. Therefore, we used the FDA-approved and CE-IVD marked HC2 HPV test kit to examine whether some of these HPV positive samples had the oncogenic high-risk HPV types. We then sequenced DNAs from two high-risk HPV positive samples and confirmed that one sample had HPV16 DNA and another had HPV18 DNA (Figure 2C, Figure 2E-2G).

Our human samples were purchased from the UNMC Tissue Bank. The existing documents indicated that their strict sampling procedure avoided potential secondary contaminations of the collected tissues. Despite this, we followed the reviewer's suggestion and examined nine new fallopian tube early lesion (serous tubal intraepithelial carcinoma or STIC) samples with RNAscope analysis. The results showed the presence of HPV16 E6/E7 mRNA in tumor cells of one examined STIC lesion (Fig. R2), supporting our observation that at least a portion of STIC lesions of HGSOC are high-risk HPV positive.

It would also be important to exclude the presence of HPV induced tumors in the analyzed patients as several publications describe the metastasis of tumor cells to the fallopian tube mimicking STICs.

Response: The patient sample record demonstrated that these are primary fallopian tube tumors. The HPV16 E6/E7 mRNA positive STIC lesion (see nuclear TP53) has high PAX8 (fallopian tube epithelial cell marker) but negative KRT14 (cervical epithelial marker) (Fig. R6), suggesting that these are primary STIC lesions, not metastasis.

Figure R6. Expression of cervical and ovarian cancer biomarkers in an HPV E6/E7 positive STIC lesions. The high expression of PAX8 and nuclear TP53, as well as the negative staining of the KRT14, demonstrated that these tumor cells are derived from the fallopian tube epithelial cells, not metastasis of HPV-positive cervical cancer cells. The insert is the kRT14 positive control tissue (skin keratinocytes).

2. The discussion of the proposed hypothesis, that HPV can contribute to HGSC must be improved and should include data pointing against this idea. (i) It is known that HPV can infect a variety of epithelial cells, but the replication is strongly dependent on the cell type and epithelial differentiation. (ii) The proposed "hit-and-run" mechanism is described for cutaneous beta-HPV types increasing the risk for squamous cell skin cancer after UV-exposure but not for genital alpha-HPV types. The overwhelming majority of cervical cancer contains HPV and strongly depends on the continuous expression of E6/E7. (iii) The majority of HGSOC precursor lesions (p53-signature and STIC) exhibit an overexpression of p53 that is unlikely to occur under expression of E6. The protective effect of tubal ligation seems to be subtype specific (endometrioid cases; Duus AH et al, GynOnc 2023) and thus may originate from the disruption of a retrograde menstruation.

Response: We thank the reviewer for the constructive suggestions. We included the points mentioned by the reviewers in the discussion of our revised manuscript.

(i) It is known that HPV can infect a variety of epithelial cells, but the replication is strongly dependent on the cell type and epithelial differentiation.

Response: As mentioned by the reviewer, previous studies demonstrated that HPV replication depends on the cell type and epithelial differentiation. Replication of HPV in the stratified cervical epithelial cells maintains a persistent HPV infection that is required for the accumulation of genetic alterations (e.g., mutations, deletions, gain, etc.) to induce the transformation of normal cervical epithelium. However, our recent findings demonstrated that if the epithelial cells already have intrinsic genome alterations (e.g., YAP1 oncogene hyperactivation), HPV is only required temporally for aiding cells with hyperactivated YAP1 to override oncogene-induced senescence barrier (This is a critical defensive mechanism). It seems that replication and persistent infection of HPV in these cells are not necessarily required for the malignant transformation of these abnormal cells.

(ii) The proposed "hit-and-run" mechanism is described for cutaneous beta-HPV types increasing the risk for squamous cell skin cancer after UV-exposure but not for genital alpha-HPV types.

Response: it is very challenging to prove the presence of a "hit & run" mechanism in the tissues and organs that are frequently exposed to HPV (e.g., vaginal and cervical epithelium). Therefore, we cannot exclude the possibility that a portion of cervical cancer developed via the co-occurrence of YAP1 hyperactivation and temporal HPV infection. Because transformed cells are generally highly proliferative, they are more susceptible to the second HPV infection compared to regular cells. Since these cells have hyperactivated YAP1, which suppresses innate immunity, they are able to evade anti-tumor immune

surveillance, which facilitates the establishment of persistent HPV infection and HPV DNA integration in cells and tissue (e.g., Cervix) that are frequently exposed to high titer HPV virion.

Fallopian tube epithelium is only occasionally exposed to HPV (e.g., carried by sperm or retrograde menstruation). Compared to keratinocytes, FTECs transformed by the YAP-HPV oncogenic alliance have less opportunity to establish persistent HPV infection and HPV DNA integration. Tumors derived from these cells (HPV only hit during cell transformation/tumor initiation) are generally HPV DNA negative. Therefore, HPV "hit & run" may happen more frequently than we have acknowledged, especially to the cells and tissues that have less frequency exposing to a high titer of HPV virion.

The overwhelming majority of cervical cancer contains HPV and strongly depends on the continuous expression of E6/E7.

Response: We agree with the reviewer that a large portion of cervical cancer cells are addicted to the HPV E6/E7 oncogene during cancer progression. However, HPV-negative cervical cancer and cervical cancers with silent fossil HPV DNA have been frequently reported [PMID: 11451553; 26175888; 20473886; 12556961; 22323075; 24477171; 18628412; 29179875]. Interestingly, it seems that TP53 also plays a critical role in cervical cancer cell E6/E7 addicts. Loss of p53 predisposes HPV16 transformed cells to lose dependence on the continuous expression of HPV E6/E7 oncogenes for proliferation [PMID: 33321328].

(iii) The majority of HGSOc precursor lesions (p53-signature and STIC) exhibit an overexpression of p53 that is unlikely to occur under expression of E6.

Response: Although the vast majority of HGSOc precursor lesions exhibit overexpression of p53, these TP53 proteins are encoded by mutated *TP53* genes, leading to their nuclear accumulation (p53-signature). The rapid replication of FNE1-E6/E7;YAP^{S127A} cell in tumor xenograft may result in the accumulation of survival-prone *TP53* mutation and tumor evolution. Previous reports have already demonstrated that the TP53 protein mutant was able to inhibit E6-mediated p53 degradation [PMID: 31749782]. More recent studies also revealed that mutant misfolded p53 proteins can aggregate with each other (and also with wild-type P53) to form prion-like structures that improperly accumulate in the nucleus to exert tumor-promoting (gain-of-function) activities in many types of cancers, including ovarian cancer [PMID: 27549118; PMID: 26748848]. The mutation-associated conformation change may explain the observed evasion of TP53 protein from E6/E6AP-mediated degradation. These pieces of information have been added to the discussion of the revised manuscript. (future direction?).

The protective effect of tubal ligation seems to be subtype specific (endometrioid cases; Duus AH et al, GynOnc 2023) and thus may originate from the disruption of a retrograde menstruation.

Response: As mentioned above, retrograde menstruation represents one of several ways by which HPV reaches the upper female reproductive tract. This report strongly supports our hypothesis and further suggests that the endometrioid subtype of ovarian cancer may also be associated with high-risk HPV infection. Duus's finding has been cited and discussed in the revised manuscript.

3. The authors use an HPV Pseudovirion infection assay for different cell lines/cultures and compare them directly (Fig. 3). Can they exclude an effect of different cell culture medium? How many cells were counted per experiment? Can they exemplarily confirm the quantification by flow cytometry?

Response: We noticed cell doubling time is negatively associated with PsV infectivity, meaning highly proliferative cells generally have higher HPV infectivity. It is easy to examine the impact of serum or growth factor on PsV infectivity, but it is challenging to compare basic culture medium because cells have their specific culture medium, and changing the basic culture medium may immediately impact cell growth (and thereby PsV infectivity).

Following the reviewer's suggestion, we prepared a new batch of HPV16 PsV and confirmed the PsV infectivity by FCM in FT246 cells. A relatively lower MOI was used to ensure the specificity of PsV infection (Figure R3B).

4. The authors should improve the description of their GSEA analyses. It seems that only one of the analyses resulted in a significant enrichment albeit they used the fgsea algorithm.

Response: We tended to mark the one with a P value > 0.01 (but < 0.05)., To clarify, the P value for each analysis has been added to the related graph in the revised figure.

Minor remarks:

1. Page 5: "Unexpectedly, we found that ectopic expression of YAP or YAP1S127A promoted cellular senescence in the primary FTECs, which is indicated by the significant increase in the percentage of cells that underwent cell cycle arrest, cellular hypertrophy, and expression of senescence-associated β -galactosidase (SA- β -gal) (Fig. 1A & B)." Data for cell cycle arrest, and cellular hypertrophy should be added.

Response: We thank the reviewer for the constructive suggestion. New data showing cellular hypertrophy (Fig. R7A) and supporting cell cycle arrest (Fig. R7A & 7B) have been added to the supplemental information of the revised manuscript.

Figure R7. Ectopic expression of YAP protein in promoted cellular senescence in the cultured primary FTECs.

A) Representative images showing that overexpression of YAP1 in cultured primary FTECs induced cellular hypertrophy and cell cycle arrest. Scale bar: 20um. B) Hyperactivation of YAP1 in cultured primary FTECs induced upregulated expression of factors associated with cell cycle arrest (P21 and P16). Please note that the presence of HPV16 E6/E7 eradicated YAP^{S127A}-induced upregulation of P21 and P16. **: $P < 0.01$ when compared to the control group (CTRL); ***: $P < 0.001$ when compared to the control group (CTRL); ****: $P < 0.0001$ when compared to the control group (CTRL).

2. Page 6: "(consensus primers PGMY09/MY11)" My09/11 primers are degenerated, the GP primers are consensus. Please correct in the material section

Response: Thanks a lot. This error has been corrected in the revised manuscript.

3. Page 9: "Here, we observed that fallopian tube and ovarian HGSC cells, which have higher expression of YAP1, also have higher expression of HPV receptors and increased susceptibility to HPV infection (Fig. 3)." Figure 3 shows this correlation only for CxCa and HGSC cells.

Response: The information has been updated. Thanks a lot.

4. The authors include mouse experiments (Fig. 5). A graphical overview or KM-plot for tumor development would be an advantage. In Figure 5P, Q they show a strong p53 expression. Do the cells still express HPV E6/E7?

Response: We did not make a Kaplan-Meier survival plot because FNE1-MX and FNE1-E6/E7 cells couldn't form tumors in the athymic nude mice. In addition, tumors derived from the FNE1-E6E7;YAP^{S127A} cells grew very fast in athymic nude mice, leading to severe health issues and an IACUC-associated mandatory euthanasia. Following the reviewer's suggestion, we added a graphical overview in the supplementary information to clarify the experimental procedure.

Nuclear accumulation of TP53 protein in tumors derived from FNE1-E6E7;YAP^{S127A} cells verified the validity of our animal models because nuclear TP53 is a molecular signature of human HGSOV. It is possible that the rapid replication of FNE1-E6/E7;YAP^{S127A} cell resulted in *TP53* gene mutation in some cells and the FNE1-E6/E7;YAP^{S127A} cells with mutated TP53 were selectively enriched and became dominant cells in the xenograft tumor. Previous reports have already demonstrated that mutated TP53 inhibited E6-mediated p53 degradation [PMID: 31749782]. Interestingly, recent studies indicated that mutant misfolded p53 proteins can aggregate with each other (and also with wild-type P53) to form prion-like structures that improperly accumulate in the nucleus to exert tumor-promoting (gain-of-function) activities in many types of cancers, including ovarian cancer [PMID: 27549118; PMID: 26748848].

5. Page 16: "Generally, HPV preferentially infects the cervical epithelial cells and cancerous fallopian tube epithelial cells. This is consistent with previous reports that HPV infection of epithelial cells requires cell cycle progression, and that cancer cells (usually highly proliferative) are more susceptible to HPV infection [13, 41, 42]." The authors should differentiate between their infection assay and literature data for stable replication of HPV.

Response: Thanks for the constructive suggestion. We revised this section to differentiate the infection assay with literature data for stable replication of HPV.

Referee #3:

In the manuscript HPV-YAP1 oncogenic alliance drives malignant transformation of fallopian tube epithelial cells, authors investigated the role of high-risk HPV infection together with hyperactivation of YAP1 in the onset of ovarian carcinoma, through the infection of fallopian tube epithelium. YAP1 activation fosters HPV infection by promoting cell intake of HPV virions and suppressing innate immunity signals.

Authors provide many *in vivo* and *in vitro* results to validate their hypotheses and data are clearly explained. From a mechanistic point of view, authors clearly explained the role of YAP1 in facilitating HPV infection and their coordinated activity in promoting tumour formation and metastasis.

Response: We thank the reviewer for acknowledging the significance of the HPV-YAP1 oncogenic alliance in HGSOV development. The results in this manuscript provide *in vitro* evidence that this oncogenic can induce FTEC transformation. We are currently generating *in vivo* transgenic models to uncover the detailed mechanisms underlying its pathogenic activity.

Software quantifications need to be added to the immunofluorescence images in figure 3 and 4.

Response: The immunosignals in Figure 3 and Figure 4 have been qualified, and the quantification data was presented in figures and supplemental figures (Appendix) in the revised manuscript.

I have concerns regarding the real impact of HPV infection on the onset of ovarian carcinoma.

Response: Although the role of high-risk HPV in cervical cancer has been confirmed, its role in HGSC is still under debate, mainly because HPV DNA is not frequently detected in HGSC. Continuous exposure to HPV increases the chance of HPV DNA integration into the host genome, which commonly occurs in vaginal and cervical tissue. Fallopian tube tissues are only occasionally exposed to high-risk HPVs. Consistently, we rarely identify HPV DNA integration in the fallopian tube and ovarian cancer samples.

This study is inspired by the observation that although hyperactivated YAP1 induces malignant transformation of immortalized fallopian tube epithelial cells, it induces senescence in the primary fallopian tube epithelial cells. We screened many intrinsic and extrinsic factors in the past several years and found that the temporal presence of high-risk HPV E6/E7 proteins successfully switched primary fallopian tube epithelial cells from YAP^{S127A}-induced senescence to transformation. This observation demonstrated that high-risk HPV may contribute to HGSC initiation, possibly via a "hit & run" mechanism.

Although it may take years to verify our hypothesis completely, the results presented in this manuscript already indicate that: 1) high-risk HPV can infect fallopian tube epithelial cells; 2) HPV may contribute to the initiation of at least a portion of HGSC by facilitating cancer-initiating cells to override oncogene-induced senescence and/or aiding the transformed cells to evade innate immunity. It is the first solid step toward a complete understanding of the role of HPV in HGSC development.

The contribution of HPV to HGSOC is also supported by recent ovarian cancer genomic studies [PMID: 32214244; PMID: 28410234]. Since more than 95% of cervical squamous cell carcinoma (CESC) tissues are tested HPV DNA positive, we extracted and compared the HPV signature data from ovarian and CESC data sets in the TCGA database using cBioPortal online tools (<https://www.cbioportal.org>). These data demonstrated that at least a portion of HGSOC samples is potentially HPV positive (Fig. R8).

Figure R8. HPV signature in ovarian and cervical cancer tissues.

HPV signature data were extracted and downloaded from the TCGA database using the cBioportal online tool (<https://www.cbioportal.org>). Based on the HPV signature score and the fact that more than 95% of cervical squamous cell carcinoma (CESC) are HPV positive, it is reasonable to conclude that at least a portion of ovarian cancer tissues (red arrow) are HPV positive. Normal tissues: n=15; Ovarian tumor: n=913; CESC tumor: n=372. [Data source: TCGA database; associated publication: Cancer Genome Atlas Research Network, *et al.*, *Nature Genetics*, 2013. PMID:24071849. Gregory Poore, *et al.*, *Nature*, 2020, PMID: 32214244]

In particular, the origin of the samples examined in figure 2B and C is not clear. Were they the same samples used in figure 2A? In that case, why results are discrepant? Were they all fallopian tube cancer samples?

Response: The same DNA samples were used in these experiments to examine the presence of HPV DNA in non-cancerous and cancerous fallopian tube tissues. The discrepancy is derived from the detection system. Data in Figure 2A are from the standard PCR system (MY09/11;GP5+/6+), which can detect many different HPV types (known high-risk, known low-risk, and potentially unknown subtypes).

However, the HC2 assay (the FDA-approved and CE-IVD marked HPV test kit) followed by DNA sequencing confirmed the presence of high-risk HPV (HPV16 and HPV18) in two samples (figure 2B-2G). All samples are of fallopian tube origin. It seems that samples with chronic infection have a higher frequency of HPV positivity (with unclear subtypes). Only two cancer samples are high-risk HPV positive.

Moreover, it would be interesting to investigate whether those two HPV high-risk positive fallopian tube cancer samples (CP3 and CP7) present higher expression of YAP1 and HPV related receptors, compared to non-infected samples, as well as the downregulation of innate immunity signaling.

Response: The original human samples were purchased from UNMC Tissue Bank and used for HPV DNA detection several years ago. Therefore, we do not have the exact same sample to examine the expression of HPV related receptors and the major components of the innate immune signaling pathway. However, our previous study has shown that compared to normal ovarian and fallopian tube tissues, YAP1 is overexpressed in high grade serous ovarian cancer [PMID: 25798835; PMID: 26364602]. Consistently, we found that, when compared to that of normal control tissues, genes encoding the putative HPV receptor molecules such as *SDC1*, *ITGA6*, *ITGB4*, and *HSPG2* are significantly upregulated in the ovarian tumor tissues (Fig. R9), while genes encoding key components of the innate immune signaling pathways such as *DHX58*, *IRF3*, *STAT5A*, *STAT6*, and *JAK1*, are significantly downregulated in ovarian cancer tissues (Fig. R10, next page).

Figure R9. Expression of genes encoding the putative HPV receptor molecules in normal ovarian tissues, primary ovarian tumor, and recurrent ovarian tumor tissues. The TCGA TARGET GTEx study online tool (<https://xenabrowser.net/>) was used to compare expression of genes encoding the putative HPV receptor molecules in normal ovarian tissues (n=88), primary ovarian tumor (n=418), and recurrent ovarian tumor tissues (n=8). Data from the study is from the UCSC RNA-seq Compendium, where TCGA, TARGET, and GTEx samples are re-analyzed using the same RNA-seq pipeline. ****: P < 0.0001, compared to the normal control group (Normal).

Figure R10. Expression of genes encoding the key molecules of the innate immune signaling pathway in normal ovarian tissues, primary ovarian tumor, and recurrent ovarian tumor tissues. The TCGA TARGET GTEx study online tool (<https://xenabrowser.net/>) was used to compare expression of genes encoding the putative HPV receptor molecules in normal ovarian tissues (n=88), primary ovarian tumor (n=418), and recurrent ovarian tumor tissues (n=8). Data from the study is from the UCSC RNA-seq Compendium, where TCGA, TARGET, and GTEx samples are re-analyzed using the same RNA-seq pipeline. **: P < 0.01; ***: P < 0.001; ****: P < 0.0001, when compared to the normal control group (Norm-OV).

Dear Dr. Wang,

Thank you for the submission of your revised manuscript to our editorial offices. I have now received the reports from the three referees that I asked to re-evaluate the study, you will find below. As you will see, the referees now support the publication of the study in EMBO reports. Referees #1 and #2 have some remaining concerns and suggestions to improve the manuscript, I ask you to address in a final revised manuscript. Please also provide a final p-b-p-response regarding the remaining points of the referees.

- There are author name discrepancies: It is Madelyn Moness in the manuscript text file and Madelyn Leigh Moness in the submission system; Kazi Islam in the manuscript text file and Kazi Nazrul Islam in the submission system; Sophia HL George in the manuscript text file and Sophia George in the submission system; Ronny I Drapkin in the manuscript text file and Ronny Drapkin in the submission system; Babu Guda in the manuscript text file and Chittibabu Guda in the submission system. Please check and make sure the authors names are identical in the submission system and the manuscript text file.

- Please order the manuscript sections like this, using these names:

Abstract - Keywords - Introduction - Results - Discussion - Methods - Data availability section - Acknowledgements - Disclosure and Competing Interests Statement - References - Figure legends - Expanded View Figure legends

- Please remove the referee token from the "Data Availability section" and make sure the dataset is public latest upon online publication of the study.

- Please make sure that the number "n" for how many independent experiments were performed, their nature (biological versus technical replicates), the bars and error bars (e.g. SEM, SD) and the test used to calculate p-values is indicated in the respective figure legends (also for potential EV figures and all those in the final Appendix). Please also check that all the p-values are explained in the legend, and that these fit to those shown in the figure. Please provide statistical testing where applicable. Please avoid the phrase 'independent experiment', but clearly state if these were biological or technical replicates. Please also indicate (e.g. with n.s.) if testing was performed, but the differences are not significant. In case n=2, please show the data as separate datapoints without error bars and statistics. See also:

<http://www.embopress.org/page/journal/14693178/authorguide#statisticalanalysis>

If n<5, please show single datapoints for diagrams. Presently, some diagrams seem to miss statistics or the 'n.s.'. Please check. Moreover:

- Please note that the exact p values are not provided in the legends of figures 1b, d; 3b, d-e; 4b-d, f, h; 5a; 6a; 7a, c; 8a; EV 2; EV 3b; EV 4; EV 5a-b. Please provide exact p-values.

- Please indicate the statistical test used for data analysis in the legends of figures 1b, d; 3b, d-e; 4b-d, f, h; 5a; 6a-d; 7a, c; 8a; EV 2; EV 3b; EV 4; EV 5a-b.

- Please provide information related to n in the legends of figures 1d; 2b-c; 4d.

- Although 'n' is provided, please describe the nature of entity for 'n' in the legends of figures 1b, f-h; 3b, d-e; 4b-c, f, h; 5a; 6a; 7a, c; 8; EV 3b; EV 5a-b.

- Please note that the error bars are not defined in the legends of figures 1d; 2b-c; 4d."

- Please note that the blue arrows are not defined in the legends of figures 2a, EV 1c. This needs to be rectified.

- Please note that the white and yellow arrows are not defined in the legend of figure 3f. This needs to be rectified."

- Please add to each legend (main, and EV figures, where applicable) a 'Data Information' section explaining the statistics used or providing information regarding replicates and scales. See:

- Please add scale bars of similar style and thickness to microscopic images, using clearly visible black or white bars (depending on the background - NOT red). Please place these in the lower right corner of the images themselves. Please do not write on or near the bars in the image but define the size in the respective figure legend. Presently, scale bars are missing for panels 5h, j, l, n, p, the scale bar needs to be defined for figures 3c; 5c-d and a scale bar and its definition are missing for panel 3f.

- Please make sure that all the funding information is also entered into the online submission system and that it is complete and similar to the one in the acknowledgement section of the manuscript text file. Presently the Vincent Memorial Hospital Foundation/Vincent Center for Reproductive Biology, the Olson Center for Women's Health; the Ruggles Family Foundation, the Barbara Learned Bridge Funding Award from the Marsha Rivkin Center for Ovarian Cancer Research, and the Fred & Pamela Buffett Cancer Center (Lb595); Veterans Administration Senior Research Career Scientist Award are missing in the submission system.

- Please make sure that all figure panels are called out separately and sequentially. Moreover, the callout for "Table S1" needs to be corrected to "Appendix Table S1" and the one to "Fig. S8" needs to be corrected to "Appendix Fig. S8". Please check.
- In the Appendix file, there are labels to 'Supplementary figure Sx' (at the bottom of some pages). Please remove these. Moreover, Please call this file 'Appendix' on its title page.
- It seems that the figure panel EV 1c is not labelled in the figure, although there is a corresponding legend for EV 1c. Please check.
- The images shown in panel 4i, silTGAG6, YAP and YAPS127A (Merge), the last two at the bottom of Fig. 4, are partly identical (which should not be the case), also in the source data provided. Please check.
- Thanks you for depositing the source data (SD) for this study. But please provide a filled in source data checklist with your final submission (attached again) and make sure all SD requested is provided.

In addition, I would need from you:

Best,

Referee #1:

I thank the authors for answering my questions about the manuscript. I agree with most of the explanations provided. I will point few of my concerns.

(1) the authors inaccurately interpreted the silencing of integrated HPV copies in cervical cancers. They cited papers to suggest that only 20% of integrated HPV DNA is productive (I assume they meant transcribing HPV E6/E7 mRNAs). But the cited PMID 28384274 clearly presented the consensus that in cases of type 2 integration, when multiple tandem copies of HPV are integrated only the 3' terminal copy transcribe HPV E6/E7 encoding mRNA. This paper also informed that most of the HPV+ cervical cancers require continued expression of E6 and E7 to overcome pRB and p53 mediated cell cycle control and through additional oncogenic effects.

(2) Authors did not provide any evidence that the p53 in FNE1-E6/E7-YAPS217A was indeed mutated. Hence their argument of potential p53 mutation and effects is pure speculation. A sequencing of p53 in that cell line could have solved the confusion. Alternatively, there could be another explanation of reduced E6 expression.

(3) Sorry for not noticing before, the authors did not provide any evidence for HPV16 E6 or E7 protein in FTEC or FNE1 cells expressing ectopic viral genes. There is no evidence for E6/E7 protein or RNA expression, E6 mediated degradation of p53 or pRB, elevation of p16 Ink4a, a hall mark of HPV16 E7 activity. In fact, figure S1B shows reduction of p16 in the presence of HPV16 E6/E7. This is confusing.

(4) The fact that HC2 test confirmed presence of high-risk HPV in only two specimens and RNA analysis confirmed HPV E6/E7 RNA only one specimen out of many cases analyzed suggest rare incidence of HPV infection in HGSOcs.

(5) If the authors are correct, then after immortalization, FNE1-E6/E7-YAPS127A cell should remain transformation competent if E6/E7 is silenced. This experiment would have validated hit & run theory. The authors probably did not perform this experiment.

However, after over all reading, the hit and run mechanism of high-risk HPV induced carcinogenesis in HGSOc or fallopian tube cancers is an interesting possibility in the light of presented evidences from in vitro and xenograft experiments. Importantly, authors presented significant data on the effect of activated YAP1 on immune pathways in FNE1 fallopian tube cells in the

presence and absence of HPV oncogenes, which could be informative to readers. Hence this is worth of sufficient scientific merit after minor corrections/explanations and limiting overspeculation.

Referee #2:

He et al. describe in their revised manuscript the interaction of HPV E6/E7 and YAP1 and raise the hypothesis that this alliance can potentially contribute to the malignant transformation of fallopian tube cells resulting in HGSOE development. The manuscript is improved by the revision and some rather speculative parts were removed. However, I still see some, likely erroneous parts in this manuscript and the authors did not fully address all comments and did not include all changes mentioned in the rebuttal into the manuscript text.

1. The authors used well-known PCR systems (MY09/11; GP5+/6+) for the detection of HPV-DNA in cell lines and clinical samples. However, these low stringency systems enabling a broad detection of different HPV types cannot be evaluated by pure agarose gel electrophoresis but need a specific detection (southern blot, sequencing, probe hybridization etc.) to verify the presence of HPV sequences. This requirement is already proven in the presented data. Only 2/11 MY09/11-positive samples could be validated by the HC2 test and many of the PCRs from the analyzed cell lines show a band at the expected size of the PCR product (Fig. S1). The authors discuss in their rebuttal letter, that PCR bands (Fig. 2A) being not validated by the HC2 test may originate from unknown HPV. Albeit possible, the amplification of unspecific PCR products is not excluded. Thus Fig 2A may point to a higher frequency of HPV than truly present.

To support the hypothesis of an active contribution of HPV in vivo the authors added RNAScope experiments to prove the presence of HPV-derived transcripts. One out of 11 tested STIC tissues showed positive hybridization signals. The frequency resembles more likely HC2 test results than MY09/11 PCR data. The authors try to explain the lower frequency of HPV transcripts with the presence of silent HPV integrates in CxCa. However, albeit such integrates are present in CxCa tissue, the majority of cells still express HPV oncogenes from active integrates or episomes.

2. The discussion of the proposed hypothesis, that HPV can contribute to HGSOE must be improved and should include data pointing against this idea. This criticism was not fully addressed in the revised manuscript. E.g. the authors still include the argument of the protective effect of tubal ligation to support their hypothesis albeit data point to a subtype specific effect for endometrioid cases but not HGSOE (Duus AH et al, GynOnc 2023). Moreover, they argue that episomal HPV in CxCa will gradually disappear during cell division and is recovered by recurring infections. This is not correct. Episomal HPV genomes are stably inherited and CxCa tissue will not produce viral particles, because of the missing keratinocyte differentiation (being a prerequisite for expression of capsid proteins). Also the stated frequency of 95% integration in CxCa is not correct.

3. The authors include in their rebuttal TCGA data to show a certain fraction of HGSOE with a HPV signature (R8). However, the mentioned reference Poore 2020 was actually retracted. Thus the accuracy of their analysis should be checked.

Referee #3:

Authors have critically addressed all issues. Manuscript can be accepted. Thank you

Dear Dr. Breiling,

Thank you for the opportunity to improve our manuscript further. We appreciate the reviewers for recognizing the scientific merits of our findings and supporting the publication of our research results in *EMBO Reports*. We addressed the reviewer's remaining concerns point-by-point and carefully revised our manuscript based on the reviewer's suggestions. We also carefully formatted the final version of our manuscript to ensure it complies with all editorial requirements. We believe this revision has significantly improved our manuscript.

Thank you again for all your kind support.

Cheng Wang, Ph.D.

Department of Obstetrics and Gynecology
Massachusetts General Hospital
Harvard Medical School

The editorial requests:

- There are author name discrepancies: It is Madelyn Moness in the manuscript text file and Madelyn Leigh Moness in the submission system; Kazi Islam in the manuscript text file and Kazi Nazrul Islam in the submission system; Sophia HL George in the manuscript text file and Sophia George in the submission system; Ronny I Drapkin in the manuscript text file and Ronny Drapkin in the submission system; Babu Guda in the manuscript text file and Chittibabu Guda in the submission system. Please check and make sure the authors' names are identical in the submission system and the manuscript text file.

Authors' response: The author's name has been carefully checked in the manuscript and submission system, and they are identical now.

- Please order the manuscript sections like this, using these names:
Abstract - Keywords - Introduction - Results - Discussion - Methods - Data availability section - Acknowledgements - Disclosure and Competing Interests Statement - References - Figure legends - Expanded View Figure legends

Authors' response: we re-ordered the manuscript sections based on the above suggestion.

- Please remove the referee token from the "Data Availability section" and make sure the dataset is public latest upon online publication of the study.

Author's response: The referee token has been removed.

- Please make sure that the number "n" for how many independent experiments were performed, their nature (biological versus technical replicates), the bars and error bars (e.g. SEM, SD) and the test used to calculate p-values is indicated in the respective figure legends (also for potential EV figures and all those in the final Appendix). Please also check that all the p-values are explained in the legend, and that these fit to those shown in the figure. Please provide statistical testing where applicable. Please avoid the phrase 'independent experiment', but clearly state if these were biological or technical replicates. Please also indicate (e.g. with n.s.) if testing was performed, but the differences are not significant. In case n=2, please show the data as separate datapoints without error bars and statistics. See also:

<http://www.embopress.org/page/journal/14693178/authorguide#statisticalanalysis>

If $n < 5$, please show single datapoints for diagrams. Presently, some diagrams seem to miss statistics or the 'n.s.'. Please check.

Authors' response: We carefully checked each diagram and made necessary revisions based on the above instructions.

- Please note that the exact p values are not provided in the legends of figures 1b, d; 3b, d-e; 4b-d, f, h; 5a; 6a; 7a, c; 8a; EV 2; EV 3b; EV 4; EV 5a-b. Please provide exact p-values.

Authors' response: For the relatively simple comparisons, the exact p values for each comparison are presented in the figure legend of the revised manuscript. For the complicated multiple comparisons, the exact P values are included in the Excel form of source data.

- Please indicate the statistical test used for data analysis in the legends of figures 1b, d; 3b, d-e; 4b-d, f, h; 5a; 6a-d; 7a, c; 8a; EV 2; EV 3b; EV 4; EV 5a-b.

Author's response: The statistical test information for the figures mentioned above has been carefully reviewed and updated in the figure legends.

- Please provide information related to n in the legends of figures 1d; 2b-c; 4d.

Authors' response: The number and nature of "n" in figures 1d, 2b-2c, and 4d have been updated in the figure legends of the revised manuscript.

- Although 'n' is provided, please describe the nature of entity for 'n' in the legends of figures 1b, f-h; 3b, d-e; 4b-c, f, h; 5a; 6a; 7a, c; 8; EV 3b; EV 5a-b.

Authors' response: The number and nature of "n" in the figures mentioned above have been carefully reviewed and updated in the figure legends of the revised manuscript.

- Please note that the error bars are not defined in the legends of figures 1d; 2b-c; 4d."

Authors' response: The error bars in figures 1d, 2b-2c, and 4d have been defined in the figure legends of the revised manuscript.

- Please note that the blue arrows are not defined in the legends of figures 2a, EV 1c. This needs to be rectified.

Authors' response: The missed information has been added to the figure legends.

- Please note that the white and yellow arrows are not defined in the legend of figure 3f. This needs to be rectified."

Authors' response: The information about arrows has been added to the legend of Figure 3f.

- Please add to each legend (main, and EV figures, where applicable) a 'Data Information' section explaining the statistics used or providing information regarding replicates and scales. See:

Authors' response: We included a "data information" section to the legend of each figure in the revised manuscript.

- Please add scale bars of similar style and thickness to microscopic images, using clearly visible black or white bars (depending on the background - NOT red). Please place these in the lower right corner of the images themselves. Please do not write on or near the bars in the image but define the size in the respective figure legend. Presently, scale bars are missing for panels 5h, j, l, n, p, the scale bar needs to be defined for figures 3c; 5c-d and a scale bar and its definition are missing for panel 3f.

Authors' response: We carefully reviewed and updated all scale bars and associated information in the revised manuscript.

- Please make sure that all the funding information is also entered into the online submission system and that it is complete and similar to the one in the acknowledgement section of the manuscript text file. Presently the Vincent Memorial Hospital Foundation/Vincent Center for Reproductive Biology, the Olson Center for Women's Health; the Ruggles Family Foundation, the Barbara Learned Bridge Funding Award from the Marsha Rivkin Center for Ovarian Cancer Research, and the Fred & Pamela Buffett Cancer Center (Lb595); Veterans Administration Senior Research Career Scientist Award are missing in the submission system.

Authors' response: The funding information in the submission system has been updated.

- Please make sure that all figure panels are called out separately and sequentially. Moreover, the callout for "Table S1" needs to be corrected to "Appendix Table S1" and the one to "Fig. S8" needs to be corrected to "Appendix Fig. S8". Please check.

Authors' response: We carefully reviewed the manuscript to ensure all figure panels are called out in the revised manuscript. The appendix tables and figures have also been carefully reviewed and updated.

- In the Appendix file, there are labels to 'Supplementary figure Sx' (at the bottom of some pages). Please remove these. Moreover, Please call this file 'Appendix' on its title page.

Authors' response: The appendix file has been updated.

- It seems that the figure panel EV 1c is not labelled in the figure, although there is a corresponding legend for EV 1c. Please check.

Authors' response: The label for EV1C has been added.

- The images shown in panel 4i, siITGAG6, YAP and YAPS127A (Merge), the last two at the bottom of Fig. 4, are partly identical (which should not be the case), also in the source data provided. Please check.

Authors' response: We are sorry for this copy/paste error. The mis-copied image has been replaced.

- Thank you for depositing the source data (SD) for this study. But please provide a filled in source data checklist with your final submission (attached again) and make sure all SD requested is provided.

Authors' response: An updated source data checklist has been submitted.

In addition, I would need from you:

- a short, two-sentence summary of the manuscript (not more than 35 words).

Authors' response: A new file with a two-sentence summary page, highlights, and a schematic abstract page is uploaded to the submission system.

- two to four short (!) bullet points highlighting the key findings of your study (two lines each).

Authors' response: A new file with a two-sentence summary page, highlights, and a schematic abstract page is uploaded to the submission system.

- a schematic summary figure as separate file that provides a sketch of the major findings

(not a data image) in jpeg or tiff format (with the exact width of 550 pixels and a height of not more than 400 pixels) that can be used as a visual synopsis on our website.

Authors' response: A new file with a two-sentence summary page, highlights, and a schematic abstract page is uploaded to the submission system.

Response to reviewers' comments:

Referee #1:

I thank the authors for answering my questions about the manuscript. I agree with most of the explanations provided. I will point few of my concerns.

(1) the authors inaccurately interpreted the silencing of integrated HPV copies in cervical cancers. They cited papers to suggest that only 20% of integrated HPV DNA is productive (I assume they meant transcribing HPV E6/E7 mRNAs). But the cited PMID 28384274 clearly presented the consensus that in cases of type 2 integration, when multiple tandem copies of HPV are integrated only the 3' terminal copy transcribe HPV E6/E7 encoding mRNA. This paper also informed that most of the HPV+ cervical cancers require continued expression of E6 and E7 to overcome pRB and p53 mediated cell cycle control and through additional oncogenic effects.

Authors' Response: We thank the reviewer for his comments. We hope the reviewer appreciates that this study focuses on the potential role of HPV in high-grade serous ovarian cancer initiation. Our results showed that high-risk HPV facilitates cancer initiation by assisting the FTECs to override hyperactivated YAP1-induced senescence. We also provide evidence showing that HPV may contribute to HGSOc development via a "hit & run" mechanism. Therefore, the biological events associated with HPV+ cancer progression, such as cancer cell HPV DNA integration and E6/E7 addiction, are out of the scope of this study. As mentioned in the first response letter, due to its specific anatomical location, the Fallopian tube is only occasionally exposed to a small amount of HPV. In contrast, the cervical epithelium is frequently exposed to large amounts of HPV. Therefore, the cellular and molecular mechanisms by which HPV drives tumorigenesis in these tissues have a lot of differences. Hence, we believe that molecular mechanisms underlying cervical cancer progression (e.g., continued expression of E6 and E7 to overcome pRB and p53) should not be directly implanted to that of Fallopian tube cancer initiation.

HPV-negative cervical cancer and cervical cancers with silent fossil HPV DNA have been frequently reported [PMID: 11451553; 26175888; 20473886; 12556961; 22323075; 24477171; 18628412; 29179875, 28384274]. PMID 28384274 was cited to answer a CxCa-associated question raised by the reviewer in the first rebuttal letter. Since the silencing of integrated HPV copies in cervical cancers is out of the scope of the present study, the relevant information was not included in our manuscript.

(2) Authors did not provide any evidence that the p53 in FNE1-E6/E7-YAPS217A was indeed mutated. Hence their argument of potential p53 mutation and effects is pure speculation.

A sequencing of p53 in that cell line could have solved the confusion. Alternatively, there could be another explanation of reduced E6 expression.

Authors' Response: Under physiological conditions, TP53 protein is undetectable because its expression level is tightly controlled by MDM2-dependent ubiquitination and degradation [PMID:20940128]. Nuclear TP53 is a well-known molecular signature of high-grade serous ovarian carcinoma (HGSOC). The nuclear TP53 in tumors derived from FNE1-E6/7-YAP^{S127A} cells provides evidence that these tumors are HGSOC.

Since >95% of HGSOC has *TP53* mutation and mutant TP53 is predominantly accumulated in the nucleus [PMID:9765199; PMID:1360088; PMID:28035683], we speculate that *TP53* gene in these transformed cells may acquire mutations during tumor progression/evolution. However, other factors may also impact TP53 expression and subcellular location in cancer cells. As mentioned by the reviewer, increased TP53 protein may reflect reduced E6 protein expression (which actually supports our "hit and run" hypothesis). Increased TP53 protein may also derived from the dysregulation of the *MDM2* gene in these tumor cells [PMID:20940128]. More recent studies have demonstrated that in tumors with wild-type *TP53*, *TP53* could accumulate in the nucleus and cytoplasm of these cancer cells by forming amyloid-like aggregates [PMID:36741442; PMID:9765199; PMID:1360088; PMID:28035683]. Therefore, we agree with the reviewer that the observed phenotype could be explained by mechanisms beyond *TP53* mutation. We modified our discussion section to eliminate the potential confusion. I believe that reviewers also agree that a separate project is required to examine the exact mechanisms underlying TP53 protein accumulation in the nucleus of these cancer cells.

(3) Sorry for not noticing before, the authors did not provide any evidence for HPV16 E6 or E7 protein in FTEC or FNE1 cells expressing ectopic viral genes. There is no evidence for E6/E7 protein or RNA expression, E6 mediated degradation of p53 or pRB, elevation of p16 Ink4a, a hall mark of HPV16 E7 activity. In fact, figure S1B shows reduction of p16 in the presence of HPV16 E6E7. This is confusing.

Authors' Response: We thank the reviewer for the suggestions to improve our manuscript. The functionality and efficiency of our lentiviral E6/E7 expression system have been validated by previous publications [*EMBO Rep.* 2019, PMID:30755404; *Oncogene*, 2022, PMID:35761037].

As mentioned above, this study focuses on the role of HPV E6/E7 in HGSOC initiation (assisting FTECs to override hyperactivated YAP1-induced senescence). The molecular mechanisms underlying cervical cancer progression (e.g., continued expression of E6 and E7 to overcome pRB and p53) should not be directly implanted to that of Fallopian tube cancer initiation. Importantly, high-risk HPVs can function in a TP53/RB1-independent manner [PMID:7624774; PMID:9858596; PMID:9129664; PMID:8617242]. Our previous studies have demonstrated that HPV16 E6 could disrupt the YAP1-LATS2 feedback loop to facilitate cervical tumorigenesis (PMID:35761037). While this manuscript is under revision, a new preprint posted in *BioRxiv* by Dr. Liz White at the University of Pennsylvania demonstrated that HPV18 E7 inhibits LATS1 kinase and activates YAP1 by degrading PTPN14 (<https://doi.org/10.1101/2024.03.07.583953>). These findings strongly confirm our previous finding that high-risk HPV can interact with the Hippo/YAP signaling pathway in a TP53/RB1-independent manner. Nevertheless, the interaction between the Hippo signaling and HPV E6/E7 oncoproteins, as well as their interactions with the traditional HPV targets (e.g., TP53, RB1, and CDKN1B), need to be systematically revisited in future studies.

(4) The fact that HC2 test confirmed presence of high-risk HPV in only two specimens and RNA analysis confirmed HPV E6E7 RNA only one specimen out of many cases analyzed suggest rare incidence of HPV infection in HGSOCs.

Authors' Response: Yes. This observation is also consistent with TCGA sequencing results showing that the presence of HPV DNA in HGSOC cancer tissue is rare (~5%). Importantly, these observations support our hypothesis that high-risk HPVs are able to infect fallopian tube epithelial cells, and they may contribute to the initiation of at least a portion of HGSOC via a "hit & run" mechanism.

(5) If the authors are correct, then after immortalization, FNE1-E6E7-YAPS127A cell should remain transformation competent if E6E7 is silenced. This experiment would have validated hit & run theory. The authors probably did not perform this experiment.

Authors' Response: We thank the reviewer for the constructive suggestion. We have made progress in this direction by employing a different strategy. We infected FTEC-YAP^{S127A} cells with non-integrative adenovirus expressing HPV16-E6/E7 and successfully blocked YAP^{S127A}-induced senescence. Using a single-cell cloning protocol, we established several HPV-negative cell lines from these transformed cells. Our recent preliminary *in vivo* studies demonstrated that these selected HPV16 E6/E7 negative cells are tumorigenic in nude mice.

However, after over all reading, the hit and run mechanism of high-risk HPV induced carcinogenesis in HGSOC or fallopian tube cancers is an interesting possibility in the light of presented evidence from *in vitro* and xenograft experiments. Importantly, authors presented significant data on the effect of activated YAP1 on immune pathways in FNE1 fallopian tube cells in the presence and absence of HPV oncogenes, which could be informative to readers. Hence this is worth of sufficient scientific merit after minor corrections/explanations and limiting overspeculation.

Authors' Response: We thank the reviewer for recognizing the scientific merit of our findings and agreeing to publish our data after minor correction. As mentioned, we are using our experimental tools, including transgenic mice and MmuPV1 (mouse papilloma virus) to provide direct evidence for a role of HPV in HGSOC development.

Referee #2:

He et al. describe in their revised manuscript the interaction of HPV E6/E7 and YAP1 and raise the hypothesis that this alliance can potentially contribute to the malignant transformation of fallopian tube cells resulting in HGSOC development. The manuscript is improved by the revision and some rather speculative parts were removed. However, I still see some, likely erroneous parts in this manuscript and the authors did not fully address all comments and did not include all changes mentioned in the rebuttal into the manuscript text.

1. The authors used well-known PCR systems (MY09/11; GP5+/6+) for the detection of HPV-DNA in cell lines and clinical samples. However, these low stringency systems enabling a broad detection of different HPV types cannot be evaluated by pure agarose gel electrophoresis but need a specific detection (southern blot, sequencing, probe hybridization etc.) to verify the presence of HPV sequences. This requirement is already proven in the presented data. Only 2/11 MY09/11-positive samples could be validated by the HC2 test and many of the PCRs from the analyzed cell lines show a band at the expected size of the PCR product (Fig. S1). The authors discuss in their rebuttal letter, that PCR bands (Fig. 2A) being not validated by the HC2 test may originate from unknown HPV. Albeit possible, the amplification of unspecific PCR products is not excluded. Thus Fig 2A may point to a higher frequency of HPV than truly present.

To support the hypothesis of an active contribution of HPV in vivo the authors added RNAScope experiments to prove the presence of HPV-derived transcripts. One out of 11 tested STIC tissues showed positive hybridization signals. The frequency resembles more likely HC2 test results than MY09/11 PCR data. The authors try to explain the lower frequency of HPV transcripts with the presence of silent HPV integrates in CxCa. However, albeit such integrates are present in CxCa tissue, the majority of cells still express HPV oncogenes from active integrates or episomes.

Authors' response: We thank the reviewer for the constructive comments. We agree with the reviewer that the low stringency MY09/11;GP5+/6+ PCR system could detect unknown HPV strains or even unspecific signals. However, the HC2 test results, RNAScope data, and DNA sequencing data provide convincing evidence that high-risk HPVs can infect fallopian tube epithelial cells (FTECs). The low frequency could be explained by the potential "hit & run" mechanism. Nevertheless, we carefully revised our discussion to ensure that the possibilities mentioned by the reviewer are included in the discussion section of the revised manuscript.

2. The discussion of the proposed hypothesis, that HPV can contribute to HGSOC must be improved and should include data pointing against this idea. This criticism was not fully addressed in the revised manuscript. E.g. the authors still include the argument of the protective effect of tubal ligation to support their hypothesis albeit data point to a subtype specific effect for endometrioid cases but not HGSOC (Duus AH et al, GynOnc 2023). Moreover, they argue that episomal HPV in CxCa will gradually disappear during cell division and is recovered by recurring infections. This is not correct. Episomal HPV genomes are stably inherited and CxCa tissue will not produce viral particles, because of the missing keratinocyte differentiation (being a prerequisite for expression of capsid proteins). Also the stated frequency of 95% integration in CxCa is not correct.

Authors' response: Duuns AH and colleagues observed a significant reduction of epithelial ovarian cancer (EOC) risk after unilateral salpingectomy, bilateral salpingectomy, and tubal ligation in almost all histological types (Duus AH *et al.*, *Gyn. Onc.* 2023, PMID: 37683548). Since salpingectomy and tubal ligation block HPV infection of the fallopian tube epithelium, the results presented in PMID:37683548 support our hypothesis that HPV contributes to HGSOC development. Interestingly, they also found that the risk reduction post salpingectomy was only present among women younger than 50 years at the surgery. This observation provided even

stronger support to our hypothesis because this population (women younger than 50 years) is more sexually active. We cited Duus's work in the revised version of our manuscript.

The results in our manuscript demonstrate that high-risk HPVs are able to infect Fallopian tube epithelial cells and that HPV E6/E7 can assist FTECs to override hyperactivated YAP1-induced senescence, leading to their malignant transformation. Therefore, the focus of this manuscript is HGSOC initiation. We think the CxCa progression-associated HPV functional mechanisms should not be directly implanted to that of HGSOC initiation. Thus, we removed information about CxCa progression-associated HPV functional mechanisms from our manuscript.

3. The authors include in their rebuttal TCGA data to show a certain fraction of HGSOC with an HPV signature (R8). However, the mentioned reference Poore 2020 was actually retracted. Thus the accuracy of their analysis should be checked.

Authors' response: We want to emphasize that the present study focuses on the contribution of the HPV-YAP1 oncogenic alliance to the malignant transformation of FTECs and HGSOC initiation. Due to the potential "hit & run" mechanism, the presence or absence of HPV DNA in HGSOC cancer tissue has no direct relevance to HGSOC initiation. Several published papers, including Poore's paper, were mentioned in the previous rebuttal letter to discuss the integration of HPV DNA in advanced-stage ovarian cancer tissues. Therefore, these reference papers were not cited in our manuscript.

Referee #3:

Authors have critically addressed all issues. Manuscript can be accepted. Thank you.

Author's Response: We thank the reviewer again for the constructive suggestions.

Dr. Cheng Wang
Massachusetts General Hospital/Harvard Medical School
Vincent Department of Obstetrics and Gynecology
Their Building, Floor 9
55 Fruit Street
Boston, MA 02114
United States

Dear Dr. Wang,

I am very pleased to accept your manuscript for publication in the next available issue of EMBO reports. Thank you for your contribution to our journal.

Yours sincerely,
